# Adversarial Attacks and Robust Training for Hypergraph Neural Networks

**Naheed Anjum Arafat** [1 2]  **Debabrota Basu** [3]  **Yulia R. Gel** [4]  **Danda B. Rawat** [1 2]

## Abstract

Recent studies show that Hypergraph Neural Networks (HGNNs) are vulnerable to adversarial attacks, while adversarial learning in the context of hypergraphs remains substantially under-investigated. In particular, all existing attacks on HGNNs are white-box and customized for either structural or feature perturbation. But in reality, the attacker might not have access to the target model parameters. Motivated by this knowledge gap, we propose a generic meta-objective-based learning framework, `MeLA`, that leverages the hypergraph Laplacian to conduct gray-box, structural, and feature perturbations under explicit perturbation budgets. In contrast to the attack literature, there is no adversarial training mechanism for HGNNs to defend against such attacks. Hence, we propose a novel adversarial training mechanism for HGNNs to obtain a robust classifier. We further prove the convergence of our robust training. Extensive experiments across various HGNN models and datasets show that (a) our proposed attack is effective in poisoning and evasion settings, and (b) our adversarial training enhances defense against adversarial attacks.

## 1. Introduction

Many real-world applications and their datasets consist of polyadic entities representing multimodal and heterogeneous data such as relations among individuals in collaborative co-authorship (Han et al., 2009), legislators in parliamentary voting (Benson et al., 2018), items in e-shopping carts (Xia et al., 2021), proteins in protein complexes, and metabolites in a metabolic process (Gaudelet et al., 2018). Along with the popularity of hypergraphs in modeling various types of multimodal data, there has been a growing demand to make predictions on them: predicting node labels (node classification) (Yadati et al., 2019; Duta et al., 2023), predicting hyperlinks (Yadati et al., 2020). *Hypergraph Neural Networks* (HGNNs) have emerged as a powerful machinery for addressing such tasks, yielding state-of-the-art performances in a broad range of domains, such as detection of drug side-effects (Saifuddin et al., 2023), chemical reaction monitoring (Chen & Schwaller, 2024), and analysis of recommender systems (Xia et al., 2021), just to name a few. With growing adoption of HGNNs in practice, it is important to study their robustness under adversarial settings to ensure HGNN's reliability and trustworthiness. This is particularly critical in the emerging HGNN applications, such as recommender systems (Nguyen et al., 2024). For example, Amazon's marketplace, where the lack of adversarial robustness could lead to user dissatisfaction and revenue loss. Deficiency in adversarial robustness becomes more crucial for the new technology of biomedical digital twins (Mylrea et al., 2021; Brussee et al., 2025), where HGNNs, based on multimodal information, are used for pathology analysis and survival prediction. Here, malignant attacks can result in misdiagnosis and loss of human lives. Nevertheless, while recent findings indicate that HGNNs are vulnerable to adversarial attacks, i.e., strategic perturbations to the dataset during training or inference, adversarial learning for hypergraphs remains substantially under-investigated. Our aim is to understand the key latent mechanisms behind the *susceptibility of HGNNs to adversarial attacks and possible defenses for semi-supervised node classification tasks.*

**Why We Need New Adversaries for HGNNs and Why We Cannot Adapt Attacks on GNNs to HGNNs?** Although there is an extensive literature on attacking GNNs (Jin et al., 2021; Arafat et al., 2025), including meta-learning mechanism (Zügner & Günnemann, 2019), directly using them for HGNNs is fundamentally problematic. This is due to the non-uniqueness of graph-to-hypergraph projection. A single graph can correspond to multiple hypergraphs. For instance, the graph $([ab], [bc], [ca])$ can be mapped to two hypergraphs $([abc], [ac])$ and $([abc])$. Consequently, perturbations designed in the graph domain may not correspond

---

[1]Department of Electrical Engineering and Computer Science, Howard University, Washington, DC, USA [2]DoD Center of Excellence in AI/ML (CoE-AIML), Howard University, Washington, DC, USA [3]Univ. Lille, Inria, CNRS, Centrale Lille, UMR 9189 – CRIStAL, F-59000 Lille, France [4]Department of Statistics, Virginia Tech, USA. Correspondence to: Naheed Anjum Arafat <Naheedanjum.Arafat@howard.edu>, Danda B. Rawat <danda.rawat@howard.edu>.

*Proceedings of the $43^{rd}$ International Conference on Machine Learning*, Seoul, South Korea. PMLR 306, 2026. Copyright 2026 by the author(s).

to valid or meaningful perturbations in the hypergraph domain. Furthermore, our analysis (Section 4.1) shows that while graph Laplacian (used in graph message passing) is sensitive to only node degrees, hypergraph Laplacian (used in hypergraph message passing) is sensitive to a coupling of node and hyperedge degrees. Hence, graph-based attacks cannot harness the key attack amplification mechanisms arising from hyperedge degrees, reinforcing the need for hypergraph-specific attacks.

**Related Works: Attacks on HGNNs.** Recently, several adversarial attacks on HGNNs have been designed. Hu et al. (2023) proposed *hyperattack*– a targeted, white-box structural attack for HGNN (Feng et al., 2019). Given a budget of $\delta$ perturbations for a chosen target node, it first selects top-$M$ hyperedges with the highest *link-gradients*, and then, selects $\delta < M$ with the highest integrated gradient to perturb. Chen et al. (2023) proposed *MGHGA*– a *momentum-gradient* based global, white-box, feature attack for HGNN. MGHGA is designed to attack both the node and the visual object classification tasks. In turn, Chen et al. (2024) proposed a white-box, untargeted attack based on the derivative graph. He et al. (2025) devised a targeted, node injection attack also in the white-box setting.

However, these existing attacks only work in white-box setting, and there is currently *no* attack on HGNNs in black-box and gray-box settings. However, in applications, such as HGNN-based recommender system, it is unrealistic to assume that the attacker has access to proprietary model parameters. Furthermore, the existing attacks are limited to either node features or hypergraph structure, but not both. This potentially limits their effectiveness and applicability. To bridge these gaps, we propose `MeLA`-PGD and `MeLA`-FGSM– *two gray-box hypergraph attacks* that simultaneously construct feature and structural perturbations using a hypergraph Laplacian-aware loss function. *We further propose `MeLA`-D, a robust training for defending HGNNs.*

In this context, **our contributions** are two-fold.

**i) Attack:** We propose Meta-Laplacian objective–based Attack framework (`MeLA`) – a novel gray-box, untargeted attack that derives adversarial perturbations by optimizing an objective defined on the induced Laplacian operator rather than directly maximizing the task loss. `MeLA`'s attack objective contains three components: *Laplacian smoothing loss*, *classification loss*, and *stealth violation loss*. The proposed Laplacian smoothing loss measures perturbation-induced changes in Laplacian-smoothed embeddings relative to the original hypergraph, thus capturing how structural changes alter the hypergraph's smoothing behavior. The stealth violation loss penalizes large shifts in hypergraph structural statistics, such as the node degree vector. The Laplacian smoothing loss is motivated by our theoretical analysis that suggests incorporating operator-level sensitivity makes the

attacks and defenses more potent (Theorem 2). The degree penalty complements the Laplacian term by discouraging concentrated node-degree changes. Our empirical results suggest that the proposed multi-step, iterative Meta Laplacian attack `MeLA`-PGD incurs up to 60% and 76% drops in accuracy in the poisoning and evasion settings, respectively, while maintaining controlled operator-level and feature-level perturbations.

**ii) Defense:** In contrast to the attack literature, *there is no adversarial training mechanism for HGNNs to defend against adversarial attacks*. To bridge this gap, we propose `MeLA`-D: an adversarial training mechanism for HGNNs that trains with the perturbations generated by `MeLA` to obtain a robust classifier. We prove convergence of `MeLA`-D to a first-order stationary point of the defense objective. `MeLA`-D performs surrogate retraining in each attack iteration, allowing the defender to simulate the effect of perturbations on model learning. Our experiments indicate that `MeLA`-D enhances the robustness of HGNNs, offering positive robustness gains across the reported settings and reaching up to 56% gain against a variety of attacks.

## 2. Background: Hypergraphs and HGNNs

We begin by introducing the basics of hypergraphs and HGNNs. An undirected **hypergraph** $\mathcal{H} \triangleq (V, E)$ consists of a set of nodes $V$ and a collection $E$ of subsets of $V$ as hyperedges, i.e., $E \triangleq \{e_i : e_i \subseteq V\}$. The degree of a node is the number of hyperedges containing that node. A hypergraph is $k$-uniform if every hyperedge has $k$ nodes. For simple graphs, the normalized graph Laplacian is $\boldsymbol{L} \triangleq \boldsymbol{I} - \boldsymbol{D}_v^{-1/2} \boldsymbol{A} \boldsymbol{D}_v^{-1/2}$. Here, $\boldsymbol{A}$ is the adjacency matrix and $\boldsymbol{D}_v$ is the node degree matrix. The symmetric normalized hypergraph Laplacian (Zhou et al., 2006) is defined as $\boldsymbol{L}_{\mathcal{H}} \triangleq \boldsymbol{I} - \boldsymbol{D}_v^{-1/2} \boldsymbol{H} \boldsymbol{D}_e^{-1} \boldsymbol{H}^T \boldsymbol{D}_v^{-1/2}$. Here $\boldsymbol{H}$ is the incidence matrix such that $\boldsymbol{H}_{v,e} = 1$ if node $v \in e$, and 0 otherwise. $\boldsymbol{D}_v \in \mathbb{R}^{n \times n}$ is the diagonal matrix of node degrees. $\boldsymbol{D}_e \in \mathbb{R}^{m \times m}$ is diagonal matrix of hyperedge cardinalities. There are also other types of hypergraph Laplacians, e.g., Hodge Laplacian and inner product Laplacian (Schaub et al., 2020; Young et al., 2024).

**Hypergraph Neural Networks (HGNNs).** There are broadly two families of HGNNs: *spectral* and *spatial* (Antelmi et al., 2023). The *spectral HGNNs* construct the hypergraph Laplacian matrix encoding the higher-order relationships, perform filtering operations in the spectral domain, and finally, transform node features into node embeddings using these spectral filters. The *spatial HGNNs* define two-stage message-passing and aggregation: node $\rightarrow$ hyperedge and hyperedge $\rightarrow$ node. Despite the apparent difference, the operations of certain spectral HGNNs can be interpreted as forms of message passing (Hayhoe et al., 2024). Feng et al. (2019) proposed the first spectral HGNN by general-

izing spectral graph convolutions to hypergraphs through the hypergraph Laplacian. Subsequently, several variants have been proposed– HyperGCN (Yadati et al., 2019) and HNHN (Dong et al., 2020). HyperGCN reformulates hypergraph convolution using clique expansion, and HNHN introduces attention mechanisms for hyperedge importance. Notable spatial HGNNs include HyperSAGE (Arya et al., 2020), HGNN+ (Gao et al., 2022), and UniGCN (Huang & Yang, 2021). Interested readers can find further details in (Antelmi et al., 2023; Kim et al., 2024).

## 3. Formalizing Attacks & Defenses for HGNNs

We consider the semi-supervised node classification task (Chien et al., 2021) on a hypergraph $\mathcal{H} \triangleq (V, E, \boldsymbol{X}) \equiv (\boldsymbol{H}, \boldsymbol{X})$. $V$ is the set of nodes. $E$ is the set of hyperedges. $\boldsymbol{H}$ is the incidence matrix of $\mathcal{H}$. $\boldsymbol{X} \in \mathbb{R}^{n \times F}$ is the node feature matrix. Let $V_L, V_U$ be the set of labeled and unlabeled nodes, and $\boldsymbol{y}_L \in \{1, \ldots, C\}^{|V_L|}$ be the label vector for the labeled nodes. The objective is to learn a parametric function (aka *model*) $h_{\phi^*} : V_U \rightarrow \{1, \ldots, C\}$ that predicts labels for nodes in $V_U$ (e.g. HGNNs). During training, all node attributes are known and used, and the parameters $\phi^*$ are learned by minimizing a loss $\mathcal{L}_{\text{train}}$ (e.g. cross-entropy): $\phi^* = \arg\min_{\phi} \mathcal{L}_{\text{train}}(h_{\phi}(\boldsymbol{H}, \boldsymbol{X}), \boldsymbol{y}_L)$.

**I. Designing Global, Stealth-Aware, and Gray-box Attacks on HGNNs.** Adversarial attacks are typically small deliberate perturbations of input data, aiming to strategically manipulate the outcome of a model trained on the perturbed data. The attacker defines its objective by considering (i) its knowledge of the data and the target model, and (ii) constraints on the permissible adversarial perturbations.

*a. Objective and Perturbation Sets.* We study *global attacks*, i.e., the attacker's objective is not to misclassify any pre-defined target node(s) but to decrease the overall accuracy of a target model for node classification. Formally, the attacker's objective is maximizing a loss function on the labeled nodes while using the target model's weights

$$\max_{\substack{\Delta \boldsymbol{X} \in B_{\boldsymbol{X}}(\delta_X) \\ \Delta \boldsymbol{H} \in B_{\boldsymbol{H}}(\delta_H)}} \mathcal{L}_{\text{atk}}(h_{\hat{\phi}^*}(\boldsymbol{H} + \Delta \boldsymbol{H}, \boldsymbol{X} + \Delta \boldsymbol{X}), \boldsymbol{y}_L), \quad (1)$$

such that $\hat{\phi}^* = \arg\min_{\phi} \mathcal{L}_{\text{train}}(h_{\phi}(\boldsymbol{H} + \Delta \boldsymbol{H}, \boldsymbol{X} + \Delta \boldsymbol{X}), \boldsymbol{y}_L)$. Here, we allow perturbing the incidence matrix ($\boldsymbol{H}$), i.e., structural, and the features ($\boldsymbol{X}$). $B_{\boldsymbol{H}}(\delta_H) \triangleq \{\|\Delta \boldsymbol{H}\|_1 \leq \delta_H\}$ and $B_{\boldsymbol{X}}(\delta_X) \triangleq \{\|\Delta \boldsymbol{X}\|_{\infty} \leq \delta_X\}$ are the feasible perturbations given structural and feature perturbation budgets $\delta_H$ and $\delta_X$, respectively. Since the attack aims to worsen the accuracy of predictions of $h_{\hat{\phi}^*}$ on the labeled nodes, the attacker can use the cross-entropy loss for both training and attack. Formally, $\mathcal{L}_{\text{atk}} = \mathcal{L}_{\text{train}} = \mathcal{L}_{CE}(h_{\hat{\phi}^*}(V_L), \boldsymbol{y}_L)$, where $h_{\hat{\phi}^*}(V_L)$ is the target model's predictions on labeled nodes $V_L$. Eq. (1) allows the flexibility to choose $\mathcal{L}_{\text{atk}}$ and $\mathcal{L}_{\text{train}}$ differently depending on

additional constraints on the attack, e.g., unnoticeability constraints (Section 4.1).

We observe that performing an attack is a bi-level optimization problem. The inner optimization is training a target HGNN on the perturbed data, and the outer optimization maximizes the attacker's loss on unlabeled nodes.

*b. Threat Model.* We consider the *gray-box* setting, where unlike the white-box setting (Hu et al., 2023; Chen et al., 2023), the attacker has no access to the target model parameters $\phi^*$. However, the attacker has access to the hypergraph structure, i.e., the incidence matrix $\boldsymbol{H}$ and features $\boldsymbol{X}$, and training node labels $\boldsymbol{y}_L$. Without the knowledge of the target model parameters, the attacker is allowed to learn a surrogate model $f_{\theta}$ to yield adversarial perturbations. The perturbations may include addition or deletion of hyperedges. The attacker's goal is to craft perturbations using trained surrogate $f_{\theta^*}$ that transfer to and are effective against the target model $h_{\phi^*}$. If the attacker also knows the target model architecture (white-box setting), the attacker can use that same model as the surrogate. Lacking knowledge of target model parameters, the gray-box setting is more challenging than the white-box one. In applications, e.g., on an e-commerce hypergraph, a malicious seller can easily observe which products are co-purchased together (the hyperedges) and product attributes (the node features), while model weights remain proprietary. Furthermore, knowledge of training label underlines a *worst-case baseline* to test how robust the model really is (Zügner & Günnemann, 2019). If a defense works when the attacker knows the labels, it will be safer when it does not. In poisoning setting, the attacker perturbs the hypergraph before training. In evasion, the attacker perturbs the hypergraph during inference (Jin et al., 2021).

**II. Designing Defenses for HGNNs.** Adversarial training ensures the robustness of a trained model against a given set of adversarial attacks. Adversarial training optimizes the parameters of HGNN $f_{\theta}$ on an adversarially perturbed input hypergraph to increase robustness against test-time attacks, i.e., attacks against the unlabeled hypergraph after training. If the attacker's and the defender's loss functions are the same, the objective of adversarial training is

$$\min_{\theta} \max_{\substack{\Delta \boldsymbol{X} \in B_{\boldsymbol{X}}(\delta_X) \\ \Delta \boldsymbol{H} \in B_{\boldsymbol{H}}(\delta_H)}} \mathcal{L}_{\text{train}}(f_{\theta}(\boldsymbol{H} + \Delta \boldsymbol{H}, \boldsymbol{X} + \Delta \boldsymbol{X}), \boldsymbol{y}_L).$$

Though this is common in defenses for GNNs (Gosch et al., 2023), CNNs (Madry et al., 2018a), and other architectures (Zhu et al., 2020) under white-box settings, we aim to allow the attacker and defender to choose different objectives and also to construct surrogate models. For example, unnoticeability constraints matter to the attacker, but they do not directly define the defender's prediction objective. Thus, we generalize the adversarial training by turning it

into a two-player game with different losses:

$$\hat{\phi}^* = \arg\min_{\phi} \mathcal{L}_{\text{train}}(h_\phi(\boldsymbol{H} + \Delta\boldsymbol{H}^*, \boldsymbol{X} + \Delta\boldsymbol{X}^*), \boldsymbol{y}_L)$$
(2)

where the perturbations $(\Delta\boldsymbol{H}^*, \Delta\boldsymbol{X}^*)$ are obtained by $\arg\max_{\substack{\Delta\boldsymbol{X} \in B_{\boldsymbol{X}}(\delta_X) \\ \Delta\boldsymbol{H} \in B_{\boldsymbol{H}}(\delta_H)}} \mathcal{L}_{\text{atk}}(f_{\theta^*}(\boldsymbol{H} + \Delta\boldsymbol{H}, \boldsymbol{X} + \Delta\boldsymbol{X}), \boldsymbol{y}_L)$.
The above formulation encompasses both the white-box and gray-box attack settings. If the attacker has access to the target model, $f_\theta = h_\phi$. Irrespective of the setting, adversarial training may or may not use the attacker's surrogate $f_\theta$ as its $h_\phi$. To address these problems, we propose MeLA and MeLA-D as gray-box attack and defense for HGNNs.

## 4. Meta Laplacian Attack and Defense

In this section, we propose a two-stage gray-box attack, MeLA, which first trains a surrogate HGNN on the clean data and subsequently optimizes adversarial perturbations via gradient-based updates on a Laplacian-aware meta-objective computed via the trained surrogate. Furthermore, we propose MeLA-D, an adversarial defense, that trains a model to be robust against the worst-case ($\ell_\infty$) perturbations by iteratively generating adversarial perturbations and retraining the model on the resulting perturbed hypergraph.

### 4.1. Designing the Meta-Laplacian Attack (MeLA)

The main idea is to design a meta-objective that allows the attacker to maximize the classification loss while inducing controlled drift in Laplacian-smoothed embeddings, and thus, identifying the worst-case perturbations.

**Designing Meta-Objective of Attacker.** There are three components of the meta-objective.

*(a)* $\mathcal{L}_{\text{train}}$. The attacker's primary goal is to reduce the accuracy of the target node classifier. Thus, it uses the training loss of the target, i.e., cross-entropy loss, in the objective and searches for $\Delta\boldsymbol{H}, \Delta\boldsymbol{X}$ maximizing this objective. Specifically, $\mathcal{L}_{\text{train}}(\theta^*, \Delta\boldsymbol{H}, \Delta\boldsymbol{X}) \triangleq \mathcal{L}_{CE}(\boldsymbol{y}_L, f_{\theta^*}(V_L))$.

*(b)* $\mathcal{L}_{\text{stealth}}$. To keep perturbations realistic and less trivially detectable, the attacker imposes unnoticeability constraints. Following the GNN attack literature (Zügner et al., 2018; Zügner & Günnemann, 2019), perturbations are typically constrained by (i) limiting the number of structural and feature changes, and (ii) avoiding large shifts in structural statistics such as node degrees. MeLA controls structural distortion by penalizing changes in the node-degree vector: $\mathcal{L}_{\text{stealth}}(\theta^*, \Delta\boldsymbol{H}, \Delta\boldsymbol{X}) \triangleq \frac{1}{n}\|\boldsymbol{d}_{pert} - \boldsymbol{d}_{orig}\|_2^2$, where $\boldsymbol{d}_{orig}$ and $\boldsymbol{d}_{pert}$ denote the node-degree vectors of the clean and perturbed hypergraphs, respectively. Other structural regularizers, such as hyperedge cardinality preservation, can also be incorporated.

---

**Algorithm 1** MeLA: Meta Laplacian Attack Framework

**Require:** Incidence matrix $\boldsymbol{H}$, features $\boldsymbol{X}$, budgets $\delta_H, \delta_X$, learning rates $\eta_H, \eta_X$, #steps $T$, surrogate epochs $\tau_s$. {Setting $T = 1$ and $\eta_X = \delta_X$ yields MeLA-FGSM}
1: Compute Laplacian $\boldsymbol{L}_{\text{orig}} \leftarrow \text{LAP}(\boldsymbol{H})$, degrees $\boldsymbol{d}_{\text{orig}}$
2: Train HGNN surrogate $f_{\theta^*}$ for $\tau_s$ epochs on $(\boldsymbol{H}, \boldsymbol{X})$.
3: Compute reference embeddings $\boldsymbol{Z}_{\text{orig}} \leftarrow f_{\theta^*}(\boldsymbol{H}, \boldsymbol{X})$.
4: Initialize $\Delta\boldsymbol{H}^{(0)} \leftarrow \boldsymbol{0}, \Delta\boldsymbol{X}^{(0)} \leftarrow \boldsymbol{0}$.
5: **for** $t = 1, \ldots, T$ **do**
6: $\quad \boldsymbol{H}^{(t)} \leftarrow \text{CLIP}(\boldsymbol{H} + \Delta\boldsymbol{H}^{(t-1)}, 0, 1)$
7: $\quad \boldsymbol{X}^{(t)} \leftarrow \boldsymbol{X} + \Delta\boldsymbol{X}^{(t-1)}$.
8: $\quad$ Compute $\boldsymbol{L}^{(t)} \leftarrow \text{LAP}(\boldsymbol{H}^{(t)})$
9: $\quad$ Compute $\boldsymbol{Z}^{(t)} \leftarrow f_{\theta^*}(\boldsymbol{H}^{(t)}, \boldsymbol{X}^{(t)})$.
10: $\quad$ Compute meta-objective $\mathcal{L}_{\text{meta}}^{(t)}$ (Equation (3)).
11: $\quad$ Compute $(g_{\boldsymbol{H}}^{(t)}, g_{\boldsymbol{X}}^{(t)}) \leftarrow (\nabla_{\Delta\boldsymbol{H}}\mathcal{L}_{\text{meta}}^{(t)}, \nabla_{\Delta\boldsymbol{X}}\mathcal{L}_{\text{meta}}^{(t)})$.
12: $\quad$ Compute flip gain $\boldsymbol{S} \leftarrow (\boldsymbol{1} - 2\boldsymbol{H}) \odot g_{\boldsymbol{H}}^{(t)}$
13: $\quad$ Select flip set $\Omega^{(t)} \leftarrow \text{Top-}\delta_H\{S_{ij} > 0\}$.
14: $\quad$ Compute flip mask $\boldsymbol{M}_{ij}^{(t)} \leftarrow \mathbb{I}[(i, j) \in \Omega^{(t)}]$.
15: $\quad \Delta\boldsymbol{X}^{(t)} \leftarrow \pi_{\|\cdot\|_\infty \leq \delta_X}(\Delta\boldsymbol{X}^{(t-1)} + \eta_X \cdot \text{Sign}(g_{\boldsymbol{X}}^{(t)}))$.
16: $\quad$ //*Only for MeLA-PGD*//
$\quad\quad \Delta\boldsymbol{H}^{(t)} \leftarrow \eta_H \cdot \text{Sign}(g_{\boldsymbol{H}}^{(t)}) \odot \boldsymbol{M}^{(t)}$
17: **end for**
18: //*Discretization for MeLA-PGD*//
$\quad \boldsymbol{M} \leftarrow \mathbb{I}[\Delta\boldsymbol{H}^{(T)} \neq 0]$.
19: //*Discretization for MeLA-FGSM*//
$\quad \boldsymbol{M} \leftarrow \boldsymbol{M}^{(1)}$
20: $\boldsymbol{H}_{\text{pert}} \leftarrow \boldsymbol{H} + (\boldsymbol{1} - 2\boldsymbol{H}) \odot \boldsymbol{M}, \boldsymbol{X}_{\text{pert}} \leftarrow \boldsymbol{X} + \Delta\boldsymbol{X}^{(T)}$.
21: **Return** poisoned data $(\boldsymbol{H}_{\text{pert}}, \boldsymbol{X}_{\text{pert}})$.

---

*(c)* $\mathcal{L}_{\text{smooth}}$. Hypergraph Laplacian smoothing encourages nodes connected through shared hyperedges to have similar embeddings (Agarwal et al., 2006; Feng et al., 2019). In MeLA, the attacker tries to maximize the discrepancy in smoothed embeddings before and after the attack. By doing so, the attack disrupts the propagation behavior induced by the original hypergraph structure. We define Laplacian smoothing loss as $\mathcal{L}_{\text{smooth}}(\theta^*, \Delta\boldsymbol{H}, \Delta\boldsymbol{X}) \triangleq \|\boldsymbol{L}_{pert}\boldsymbol{Z}_{pert} - \boldsymbol{L}_{orig}\boldsymbol{Z}_{orig}\|_2$. Here, $\boldsymbol{Z} = f_{\theta^*}(\boldsymbol{H}, \boldsymbol{X})$ and $\boldsymbol{Z}_{pert} = f_{\theta^*}(\boldsymbol{H} + \Delta\boldsymbol{H}, \boldsymbol{X} + \Delta\boldsymbol{X})$ are node embeddings obtained from the clean graph and the poisoned graph, respectively. We adopt Zhou's Laplacian (Section 2), though MeLA is compatible with other Laplacians. Putting these components together, the attacker optimizes the following meta-objective:

$$\mathcal{L}_{\text{meta}}(\theta^*, \Delta\boldsymbol{H}, \Delta\boldsymbol{X}) = \alpha\mathcal{L}_{\text{smooth}}(\theta^*, \Delta\boldsymbol{H}, \Delta\boldsymbol{X}) -$$
$$\beta\mathcal{L}_{\text{stealth}}(\theta^*, \Delta\boldsymbol{H}, \Delta\boldsymbol{X}) + \gamma\mathcal{L}_{\text{train}}(\theta^*, \Delta\boldsymbol{H}, \Delta\boldsymbol{X}) \quad (3)$$

such that $\theta^* = \arg\min_\theta \mathcal{L}_{\text{train}}(f_\theta(\boldsymbol{H} + \Delta\boldsymbol{H}, \boldsymbol{X} + \Delta\boldsymbol{X}), y_L)$, and $\alpha, \beta, \gamma > 0$ are regularizing coefficients. The surrogate parameters $\theta^*$ are obtained by training on the clean data and are held fixed during the attack.

**Deriving Perturbation Strategies.** Given perturbation bud-

gets $\delta_H$ and $\delta_X$, MeLA employs gradient-based updates to approximately maximize the meta-objective. In particular, under $\ell_\infty$-bounded feature perturbations, a first-order approximation yields

$$\Delta \boldsymbol{X} = \delta_X \, \texttt{Sign}(\nabla_{\Delta X} \mathcal{L}_{\text{meta}}),$$

which is an analogue of the Fast Gradient Sign Method (FGSM) (Goodfellow et al., 2015). FGSM enjoys properties such as satisfying First-Order Stationary Condition (FOSC) for constrained optimization that ensures convergence of the attack (Wang et al., 2019), and being the best response of an attacker against any robust training algorithm (Pal & Vidal, 2020). For structural perturbations on binary incidence matrices, MeLA further expresses the attack as flipping a limited number of entries. The following result (proof deferred to Appendix) characterizes the optimal flip selection under strict cardinality ($\ell_0$) budget.

**Proposition 1.** *Let* $\boldsymbol{H} \in \{0,1\}^{n \times m}$ *be the binary incidence matrix. Assume the attacker can flip at most* $\delta_H$ *entries of* $\boldsymbol{H}$. *For any index set* $\Omega \subseteq [n] \times [m]$ *with* $|\Omega| \leq \delta_H$, *define the* ***binary flip operation*** *as* $H'_{ij} = 1 - H_{ij}$, *if* $(i,j) \in \Omega$, *and* $H_{ij}$, *otherwise. Let* $\boldsymbol{M} \in \{0,1\}^{n \times m}$ *be* $M_{ij} \triangleq \mathbb{I}[(i,j) \in \Omega]$, *with* $\boldsymbol{H}' \triangleq \boldsymbol{H} + (1 - 2\boldsymbol{H}) \odot \boldsymbol{M}$ *and* $g_H \triangleq \nabla_H \mathcal{L}(\boldsymbol{H})$ *be the gradient of the loss w.r.t.* $\boldsymbol{H}$. *Then the optimal flip set* $\Omega^*$ *is top-$\delta_H$ entries of* $\{(i,j) : S \triangleq g_H(1 - 2\boldsymbol{H}) > 0\}$.

**Structural Perturbations: MeLA-FGSM vs. MeLA-PGD.** Following Proposition 1, MeLA selects, at each iteration, the Top-$\delta_H$ incidence entries whose flips yield the largest first-order increase in the meta-objective (Lines 12–14 of Algorithm 1). When $T = 1$, this procedure reduces to MeLA-FGSM, which applies a single gradient step followed by a discrete Top-$\delta_H$ incidence flip using the single-step mask $\boldsymbol{M}^{(1)}$ (Line 19). For $T > 1$, it becomes MeLA-PGD that iteratively refines the structural perturbation via masked gradient updates (Line 16) and applies a discretization step based on the final perturbation (Line 18). Throughout, the surrogate model remains fixed after initial training (Lines 2-3) and is only used to compute gradients (line 11), yielding a unified Laplacian-aware gray-box attack framework.

### 4.2. Revealing Gray-box: Learning a Surrogate

In the gray-box setting, the attacker does not have access to the parameters of the target HGNN. As a workaround, the attacker employs a lightweight, trainable surrogate model $f_\theta$ to approximate the behavior of the target model and provide gradient information for optimizing the meta-objective. Here, MeLA employs a linearized 2-layer hypergraph convolutional network (Feng et al., 2019) (referred to as LinHGNN) with parameters $\theta \triangleq (\theta^{(0)}, \theta^{(1)})$.

Layer 1: $\boldsymbol{X}^{(1)} = (\boldsymbol{I} - \mathrm{L}_{\mathcal{H}})\boldsymbol{X}^{(0)}\theta^{(0)}$
Layer 2: $f_\theta(\boldsymbol{X} = \boldsymbol{X}^{(0)}, \boldsymbol{H}) = \mathrm{softmax}((\mathrm{I} - \mathrm{L}_{\mathcal{H}})\mathrm{X}^{(1)}\theta^{(1)})$

---

**Algorithm 2** MeLA-D: Meta Laplacian Defense

**Require:** Hypergraph ($\boldsymbol{H}, \boldsymbol{X}$), Budgets $\delta_X, \delta_H$, learning rates $\eta_H, \eta_X$, iterations $T$, training epochs $\tau_d$
1: Compute $\boldsymbol{L}_{orig} \leftarrow \texttt{LAP}(\boldsymbol{H})$.
2: Initialize perturbations
$\quad \Delta \boldsymbol{H}^{(0)} \sim \mathcal{N}(0, \boldsymbol{I}), \Delta \boldsymbol{X}^{(0)} \sim \mathcal{N}(0, \boldsymbol{I})$
3: **for** $t = 1, \ldots, T$ **do**
4: $\quad \boldsymbol{H}_{pert} \leftarrow \texttt{CLIP}(\boldsymbol{H} + \Delta \boldsymbol{H}^{(t-1)}, 0, 1), \quad \boldsymbol{X}_{pert} \leftarrow$
$\quad \boldsymbol{X} + \Delta \boldsymbol{X}^{(t-1)}, \boldsymbol{L}^{(t)}_{pert} \leftarrow \texttt{LAP}(\boldsymbol{H}_{pert})$
5: $\quad h^t_{\phi*} \leftarrow \text{Train } h^t_\phi \text{ for } \tau_d \text{ epochs on } (\boldsymbol{H}_{pert}, \boldsymbol{X}_{pert}).$
6: $\quad$ Compute clean logits $\boldsymbol{Z}^{(t)}_{orig} \leftarrow h^t_{\phi*}(\boldsymbol{H}, \boldsymbol{X})$,
$\quad$ perturbed logits $\boldsymbol{Z}^{(t)}_{pert} \leftarrow h^t_{\phi*}(\boldsymbol{H}^{(t)}_{pert}, \boldsymbol{X}^{(t)}_{pert})$.
7: $\quad$ Compute meta-loss: $\mathcal{L}^{(t)}_{\text{meta}}$ (Equation (3))
8: $\quad$ Compute meta-gradients:
$\quad g^t_{\boldsymbol{H}} \leftarrow \nabla_{\Delta H} \mathcal{L}^{(t)}_{\text{meta}}, g^t_{\boldsymbol{X}} \leftarrow \nabla_{\Delta X} \mathcal{L}^{(t)}_{\text{meta}}$
9: $\quad$ Update perturbations:
$\quad \Delta \boldsymbol{H}^{(t)} \leftarrow \Delta \boldsymbol{H}^{(t-1)} + \eta_H \, \text{sign}(g^t_{\boldsymbol{H}})$
$\quad \Delta \boldsymbol{X}^{(t)} \leftarrow \pi_{\|.\|_\infty \leq \delta_X} \left( \Delta \boldsymbol{X}^{(t-1)} + \eta_X \, \text{sign}(g^t_{\boldsymbol{X}}) \right)$
10: $\quad$ Top-$\delta_H$ projection:
$\quad M^{(t)}_{ij} \leftarrow \mathbb{I}\big[(i,j) \in \arg \text{TopK}_{\delta_H}\big(|\Delta \boldsymbol{H}^{(t)}|\big)\big]$
11: $\quad \Delta \boldsymbol{H}^{(t)} \leftarrow \Delta \boldsymbol{H}^{(t)} \odot \boldsymbol{M}^{(t)}$
12: **end for**
13: **Return** robust model $h^T_{\hat\phi*}$

---

Note that, we use LinHGNN as surrogate due to its computational efficiency, and superior attack effectiveness for HGNNs than alternatives, e.g., MLP (see Appendix Table 10). Two surrogate training paradigms emerge under MeLA depending on attack and defense settings.

*(a) Non-adaptive training.* In the gray-box attack setting, we first train a surrogate model $f_{\theta*}$ on the clean hypergraph ($\boldsymbol{H}, \boldsymbol{X}$). After training, the surrogate parameters $\theta^*$ are kept fixed, and the attacker optimizes the meta-objective $\mathcal{L}_{\text{meta}}(\theta^*, \Delta \boldsymbol{H}, \Delta \boldsymbol{X})$ using gradients computed through the surrogate. This non-adaptive training scheme is used in both MeLA-FGSM and MeLA-PGD (Algorithm 1).

*(b) Adaptive surrogate training.* Beyond attack construction, the surrogate can also be trained adaptively in the presence of adversarial perturbations. In this setting, the learner repeatedly updates its parameters to minimize the task loss under perturbations generated by a Laplacian-aware attacker, effectively performing adversarial training. This adaptive training paradigm underlies the proposed defense mechanism MeLA-D, which we discuss below.

### 4.3. Designing the Meta-Laplacian Defense (MeLA-D)

We propose MeLA-D (Algorithm 2), an adversarial training framework for HGNNs that iteratively solves Equation (2) using the meta gray-box attack to yield an adversarially robust model. MeLA-D consists of two interacting compo-

nents: an *attacker* (Lines 4-11) and a *defender* (Line 5). At each iteration $t$, the attacker first constructs a relaxed perturbed hypergraph (Line 4) and trains a surrogate model on the perturbed hypergraph (Line 5). Using this updated model, the attacker evaluates the clean and perturbed logits (Line 6) and computes the Laplacian-aware meta-loss and then computes the meta-objective $\mathcal{L}_{\text{meta}}$ using it (Line 7). Then, it performs projected gradient updates to generate new perturbations $\Delta \boldsymbol{H}^{(t)}, \Delta \boldsymbol{X}^{(t)}$ under the prescribed budget constraints (Line 9-11). This alternating process is repeated for $T$ iterations to yield a robust model $h_{\hat{\phi}^*}^T$.

Note that we choose to use PGD-style rather than FGSM-style perturbations, as multi-step updates provide more flexibility and often yield stronger adversarial examples in high-dimensional and non-linear settings (Madry et al., 2018b). From an optimization perspective, MeLA-D simulates a non-zero-sum min-max game between the attacker and the defender. When the attacker is assumed to have access to the target model (i.e., $f_\theta = h_\phi$), the framework reduces to a white-box adversarial training setting: $\hat{\phi}_{\text{white}}^* \triangleq \arg\min_\phi \max_{\Delta \boldsymbol{X} \in B_{\boldsymbol{X}}(\delta_X), \Delta \boldsymbol{H} \in B_{\boldsymbol{H}}(\delta_H)} \mathcal{L}_{\text{train}}(h_\phi(\boldsymbol{H} + \Delta \boldsymbol{H}, \boldsymbol{X} + \Delta \boldsymbol{X}), \boldsymbol{y}_L)$. Then, MeLA-D solves the bi-level optimization without training a surrogate model. We evaluate MeLA-D under this setting in our experiments.

### 4.4. Theoretical Analysis

**I. Perturbations on Graphs vs. Hypergraphs.** Suppose two adversaries are given the same row-perturbation budget $\epsilon_{\text{row}}$ and column-perturbation budget $\epsilon_{\text{col}}$ to perturb an undirected graph $\mathcal{G}$ (Adversary-G) and a undirected hypergraph $\mathcal{H}$ (Adversary-H). Due to symmetry of graph adjacency matrix $\boldsymbol{A}$, $\|\Delta \boldsymbol{A}\|_1 = \|\Delta \boldsymbol{A}\|_\infty$, thus only row-perturbation budget suffices. Now, we analyze the adversaries in terms of how they impact the graph/hypergraph Laplacian under same budget. Throughout the theoretical analysis, we will use induced $\ell_\infty$- and $\ell_1$-norm to decouple the impact of row and column perturbations on the Laplacian.

**Theorem 1** (Impact of Adversary–G). *Let the Adversary-G introduce a symmetric graph perturbation $\Delta \boldsymbol{A}$ satisfying the row-sum perturbation budget $\|\Delta \boldsymbol{A}\|_\infty \triangleq \max_i \sum_j |\Delta A_{ij}| \leq \epsilon_{\text{row}}$, such that the adjacency matrix, the degree matrix, and the graph Laplacian of the perturbed graph $\mathcal{G}'$ are $\boldsymbol{A}' = \boldsymbol{A} + \Delta \boldsymbol{A}$, $\boldsymbol{D}' = \boldsymbol{D} + \Delta \boldsymbol{D}$, $\boldsymbol{S}' = (\boldsymbol{D}')^{-1/2}$ and $\boldsymbol{L}_{\mathcal{G}'} = \boldsymbol{I} - \boldsymbol{S}'\boldsymbol{A}'\boldsymbol{S}'$, respectively. Then*

$$\frac{\epsilon_{\text{row}}}{d_{\min} + \epsilon_{\text{row}}}(1 + \kappa(\boldsymbol{D})) \leq$$
$$\|\boldsymbol{L}_{\mathcal{G}} - \boldsymbol{L}_{\mathcal{G}'}\|_2 \leq \left(\frac{\epsilon_{\text{row}}}{d_{\min}} + \left(\frac{\epsilon_{\text{row}}}{d_{\min}}\right)^2\right)(1 + \kappa(\boldsymbol{D})) \quad (4)$$

*where $\kappa(\boldsymbol{D}) \triangleq \frac{d_{max}}{d_{\min}} \triangleq \frac{\max_i d_i}{\min_i d_i}$ is the condition number of $\boldsymbol{D}$, $\epsilon_{\text{row}} \leq d_{\max}$, and $d_{\min} > 0$.*

*Implications:* (1) The lower bound implies that any nonzero degree-bounded structural perturbation induces Laplacian drift. Thus, operator-level changes are unavoidable once the structure is perturbed, even under strict degree constraints. (2) The impact of a perturbation on the Laplacian scales inversely with the minimum degree $d_{\min}$. Thus, attacks that target low-degree nodes exploit this sensitivity and can induce disproportionately large operator-level changes. (3) Degree-statistics or local connectivity guided defenses are insufficient to detect attacks that exploit Laplacian sensitivity by targeting low-degree nodes, since small degree-bounded perturbations could still yield large Laplacian drift, e.g., due to ill-conditioned graph or low $d_{\min}$.

**Theorem 2** (Impact of Adversary-H on Hypergraph Laplacian). *Given an adversary-H with $\boldsymbol{H}' = \boldsymbol{H} + \Delta \boldsymbol{H}$ with perturbation satisfying row-sum and column-sum budgets $\|\Delta \boldsymbol{H}\|_\infty \triangleq \max_v \sum_e |\Delta H_{ve}| \leq \epsilon_{\text{row}}, \|\Delta \boldsymbol{H}\|_1 \triangleq \max_e \sum_v |\Delta H_{ve}| \leq \epsilon_{\text{col}}$, for some $\epsilon_{\text{row}} \in (0, d_{\max}], \epsilon_{\text{col}} \in (0, \delta_{\max}]$. Let $\boldsymbol{D}_v' = \boldsymbol{D}_v + \Delta \boldsymbol{D}_v$ and $\boldsymbol{D}_e' = \boldsymbol{D}_e + \Delta \boldsymbol{D}_e$ be the perturbed degree matrices induced by $\boldsymbol{H}'$, let $\boldsymbol{S}' = (\boldsymbol{D}_v')^{-1/2}$, and define $\boldsymbol{A} \triangleq \boldsymbol{H}\boldsymbol{D}_e^{-1}\boldsymbol{H}^\top, \boldsymbol{A}' \triangleq \boldsymbol{H}'(\boldsymbol{D}_e')^{-1}(\boldsymbol{H}')^\top, \Delta \boldsymbol{A} \triangleq \boldsymbol{A}' - \boldsymbol{A}$. Then, for the condition numbers $\kappa_d \triangleq \frac{d_{\max}}{d_{\min}}$ and $\kappa_\delta \triangleq \frac{\delta_{\max}}{\delta_{\min}}$,*

$$2\frac{\sqrt{\epsilon_{\text{col}}\epsilon_{\text{row}}}}{\sqrt{d_{\min}\delta_{\min}}} + \frac{\epsilon_{\text{col}}\epsilon_{\text{row}}}{d_{\min}\delta_{\min}} \leq \|\boldsymbol{L}_{\boldsymbol{H}} - \boldsymbol{L}_{\boldsymbol{H}'}\|_2 \leq \quad (5)$$
$$\left(2\frac{\sqrt{\epsilon_{\text{col}}\epsilon_{\text{row}}}}{\sqrt{d_{\min}\delta_{\min}}} + \frac{\epsilon_{\text{col}}\epsilon_{\text{row}}}{d_{\min}\delta_{\min}}\right)\left((1 + \frac{\epsilon_{\text{row}}}{d_{\min}})\sqrt{\kappa_d\kappa_\delta} + \kappa_d\kappa_\delta\right).$$

*Implications:* (1) The lower bound implies that under nonzero node- and hyperedge-degree budgets, operator-level changes are unavoidable even under strict degree constraints. (2) Attacks focusing on low-degree nodes or small hyperedges can induce disproportionately large Laplacian drift under the same budget. (3) The bound depends on the product of $\epsilon_{\text{row}}$ and $\epsilon_{\text{col}}$, and not on either budget alone. This suggests that balanced attacks jointly targeting node and hyperedge degrees can be more potent than attacks that change only one. (4) The bound contains a first-order term scaling as $\sqrt{\epsilon_{\text{row}}\epsilon_{\text{col}}}$ and a higher-order term scaling as $\epsilon_{\text{row}}\epsilon_{\text{col}}$. When budgets are small, Laplacian drift increases approximately linearly. But as budgets grow the higher-order term becomes non-negligible, reflecting nonlinear amplification via degree matrix. Multi-step PGD is more likely to exploit nonlinear degree-matrix effects than single-step FGSM, since it iteratively updates perturbations while recomputing gradients under the perturbed Laplacian. (5) Even if an attack is node- and hyperedge-degree bounded, targeting low degree nodes may cause large drift in Laplacian. Hence, degree-aware defenses are insufficient and operator-level sensitivity aware defenses (like MeLA-D) are desirable.

**II. Convergence of MeLA-D.** We now prove the convergence of the robust training to stationary point(s).

**Lemma 1.** *Let us assume that the gradients of the loss are Lipschitz w.r.t. the parameters and the input, the loss function is locally concave on inputs, and the stochastic gradient has bounded variance. If the learning rate is set to $\Theta(T^{-1/2})$ and $T = \Omega(\mathcal{L}_{\text{train}}(\phi_0) - \mathcal{L}_{\text{train}}(\phi^*))$, MeLA-D satisfies $\frac{1}{T} \sum_{t=1}^{T} \mathbb{E}\left[\|\nabla \mathcal{L}_{\text{train}}(\phi_t)\|_2^2\right] = \mathcal{O}(T^{-1/2})$.*

Thus, under the stated smoothness and local concavity assumptions, MeLA-D drives the expected squared gradient norm of the defense objective to zero at rate $\mathcal{O}(T^{-1/2})$. This guarantees convergence to a first-order stationary point of the robust training objective, not global optimality.

**III. Time Complexity.** Let $d$ the feature dimension, $s_H = \|H\|_0$ the number of nonzero entries in the incidence matrix, $C_{\text{fwd}}$ and $C_{\text{bwd}}$ be the cost of one forward and backward pass of the surrogate respectively, and $C_{\text{LAP}} = \mathcal{O}(s_H)$ be the cost of computing the hypergraph Laplacian in the sparse setting. The overall time complexity of MeLA-PGD consists of one-time preprocessing cost $O\big(\tau_s(C_{\text{fwd}} + C_{\text{bwd}})\big)$ for training the surrogate model, followed by $T$ attack iterations. Each iteration requires one forward and backward pass through the HGNN, Laplacian construction, a top-$\delta_H$ selection over $s_H$ entries, and projection of perturbed node features, resulting in a per-step cost of $\mathcal{O}\big(C_{\text{fwd}} + C_{\text{bwd}} + s_H \log \delta_H + nd\big)$.

Hence, the total time complexity is $\mathcal{O}(\tau_s(C_{\text{fwd}} + C_{\text{bwd}}) + T(C_{\text{fwd}} + C_{\text{bwd}} + s_H \log \delta_H + nd))$. The $nd$ term comes from per-step updates and projection of node features $X$. On large-scale dataset, the forward-backward passes term $\mathcal{O}((\tau_s + T)(C_{\text{fwd}} + C_{\text{bwd}}))$ dominates the cost.

## 5. Experimental Analysis

We evaluate the proposed attacks and defenses on node classification tasks. Unless otherwise stated, we repeated each experiment over five random seeds and reported mean and standard deviations of test accuracies. All the algorithms are run on a system with 64 x 2 CPU cores, 500 GB RAM and a NVIDIA Tesla V100 GPU with 32 GB memory. We set $\delta_X = 0.05$, and $\delta_H = 20\%$ in all experiments. We defer the sensitivity analysis for $\delta_X, \delta_H$ to Appendix. In addition, Appendix contains (a) ablation studies of our attacks and defense (e.g., choice of Laplacians, choice of surrogates, impact of the loss components, significance of structural and feature perturbations), (b) node embedding and decision boundary visualization due to attack, (c) Node embedding drifts due to attack, (d) statistical significance tests. and (e) our hyperparameter choices. Our source codes are available at https://github.com/toggled/mela.

**Datasets and Target Neural Networks.** Following existing works (Chien et al., 2021; Wang et al., 2023; Duta et al., 2023), our datasets include **Cora** and **Citeseer** cocitation networks from (Yadati et al., 2019), and the Cora coauthorship network **Cora-CA** (Chien et al., 2021) with

*Table 1.* Statistics of datasets.

| Datasets | $|V|$ | $|E|$ | avg. edge size | # features | # classes |
|---|---|---|---|---|---|
| **Cora-CA** | 2708 | 1072 | $4.2 \pm 4.1$ | 1433 | 7 |
| **Cora** | 2708 | 1579 | $3.0 \pm 1.1$ | 1433 | 7 |
| **Citeseer** | 3312 | 1079 | $3.2 \pm 2.0$ | 3703 | 6 |

50%/25%/25% train/validation/test split.

As for target HGNNs, we have chosen HypergraphMLP (Tang et al., 2024), HGNN (Feng et al., 2019), and AllsetTransformer (Chien et al., 2021) as representative hypergraph neural networks. For defense, we train a HypergraphMLP. Experimental results with other SOTA HGNNs and datasets are deferred to the Appendix.

**Baseline Attacks.** We consider five baselines for comparison. Following (He et al., 2026), our choice represents two mainstream structural attack types: Random perturbation and Gradient-based. We also consider gray-box adaptation of existing white-box attacks: DICE (Waniek et al., 2018) and HyperAttack (Hu et al., 2023). **(i) RandFlip** (Dai et al., 2018) randomly flips $\delta_H$ entries in the incidence matrix $H$. **(ii) RandFeat** first generates a random matrix $R$ where each entry $R_{ij} \in \{-1, +1\}$ with equal probability. In other words, $R_{ij} = 2B_{ij} - 1$ where $B_{ij} \sim \text{Bernoulli}(\frac{1}{2})$. Finally, it constructs perturbed features $X_{\text{pert}} = X_{\text{orig}} + R_{ij}\delta_X$ so that $\|\Delta X\|_\infty \leq \delta_X$. **(iii) GradArgMax** (Dai et al., 2018) selects top-$\delta_H$ pairs of indices from $H$ as flipping them causes the largest change in the gradient of the loss function $\mathcal{L}_{\text{train}}$. For a fair comparison, we use LinHGNN as a surrogate for computing $\left|\frac{\partial \mathcal{L}}{\partial H}\right|$ across all the attacks. **(iv) DICE** assumes access to all node labels. We adapt it to hypergraphs by assigning each hyperedge a majority-vote proxy label, then randomly deleting same-label incidences or adding different-label incidences, subject to the budget constraint. **(v) HyperAttack**, originally a white-box attack is adapted to gray-box setting using LinHGNN surrogate.

**I. Attack Effectiveness.** We craft adversarial perturbations using various attacks and target models. Tables 2–3 show the comparison among the attacks in both poisoning and evasion settings (results for Cora are in Appendix). We measure attack effectiveness by median relative drop (%) in clean accuracy.

*Poisoning Setting.* Across all datasets, MeLA-PGD yields significantly higher degradation in test accuracy (up to 60%) compared to baseline methods such as RandFlip, RandFeat, and GradArgMax. In contrast, MeLA-FGSM, which relies on a single-step first-order approximation of the meta-gradient, is usually weaker than MeLA-PGD but often stronger than non-adaptive baselines. These results highlight the necessity of multi-step optimization, as single-step approximations are insufficient to exploit the nonlinear dependence of the hypergraph Laplacian on structural perturbations (Theorem 2).

*Table 2.* Comparison of different adversarial attacks (poisoning and evasion) for Citeseer. Drop = median relative % accuracy drop.

| Attack | HyperMLP (72.20 ± 0.40) | | AllsetTrans. (72.90 ± 0.25) | | HGNN (74.13 ± 0.10) | |
| --- | --- | --- | --- | --- | --- | --- |
| | Poison / Drop (P) | Evasion / Drop (E) | Poison / Drop (P) | Evasion / Drop (E) | Poison / Drop (P) | Evasion / Drop (E) |
| RandFeat | 69.42 ± 0.59 / 4.0 | 70.48 ± 0.92 / 2.3 | 71.84 ± 1.17 / 2.3 | 72.56 ± 0.77 / 0.7 | 71.64 ± 0.52 / 3.3 | 72.78 ± 0.49 / 2.0 |
| RandFlip | 72.05 ± 0.25 / 0.2 | 72.20 ± 0.40 / 0.0 | 67.44 ± 1.28 / 6.8 | 66.74 ± 1.16 / 8.6 | 64.49 ± 1.04 / 13.2 | 65.77 ± 0.98 / 11.6 |
| GradArgMax | 72.22 ± 0.40 / 0.0 | 72.20 ± 0.40 / 0.0 | 71.38 ± 0.76 / 1.8 | 71.06 ± 1.27 / 2.8 | 71.26 ± 0.51 / 3.7 | 71.91 ± 0.25 / 2.9 |
| DICE | 72.17 ± 0.45 / 0.0 | 72.20 ± 0.40 / 0.0 | 69.49 ± 1.17 / 5.5 | 70.05 ± 0.90 / 4.3 | 68.62 ± 1.08 / 7.8 | 69.28 ± 1.17 / 5.9 |
| HyperAttack | 72.22 ± 0.34 / 0.0 | 72.20 ± 0.40 / 0.0 | 71.47 ± 0.74 / 2.0 | 70.82 ± 0.72 / 3.0 | 63.91 ± 1.94 / 12.9 | 71.69 ± 0.33 / 3.1 |
| MeLA-FGSM | 66.55 ± 1.55 / 7.9 | 47.61 ± 0.87 / 34.3 | 61.59 ± 3.78 / 14.6 | 46.30 ± 0.50 / 36.3 | 68.41 ± 0.82 / 7.2 | 46.45 ± 0.82 / 37.9 |
| MeLA-PGD | 60.72 ± 3.69 / **15.4** | 42.54 ± 1.83 / **41.1** | 28.82 ± 0.75 / **60.1** | 41.30 ± 1.43 / **43.7** | 54.37 ± 1.17 / **26.5** | 39.81 ± 3.45 / **44.6** |

*Table 3.* Comparison of different adversarial attacks (poisoning and evasion) for Cora-CA. Drop = median relative % accuracy drop.

| Attack | HyperMLP (75.83 ± 1.10) | | AllsetTrans. (83.66 ± 0.56) | | HGNN (82.90 ± 0.24) | |
| --- | --- | --- | --- | --- | --- | --- |
| | Poison / Drop (P) | Evasion / Drop (E) | Poison / Drop (P) | Evasion / Drop (E) | Poison / Drop (P) | Evasion / Drop (E) |
| RandFeat | 72.67 ± 1.38 / 3.9 | 72.91 ± 0.95 / 3.5 | 82.33 ± 0.87 / 1.9 | 82.57 ± 0.54 / 1.4 | 82.04 ± 0.67 / 0.9 | 82.10 ± 0.91 / 1.1 |
| RandFlip | 73.21 ± 0.27 / 3.9 | 75.83 ± 1.10 / 0.0 | 79.97 ± 0.58 / 4.3 | 80.24 ± 0.83 / 4.2 | 76.22 ± 0.82 / 7.7 | 76.99 ± 0.74 / 7.3 |
| GradArgMax | 75.30 ± 1.00 / 0.8 | 75.83 ± 1.10 / 0.0 | 81.51 ± 0.66 / 2.8 | 81.60 ± 1.12 / 2.3 | 80.15 ± 0.81 / 3.2 | 80.32 ± 0.88 / 3.4 |
| DICE | 73.44 ± 0.74 / 2.9 | 75.83 ± 1.10 / 0.0 | 79.23 ± 0.74 / 6.1 | 80.47 ± 0.83 / 3.7 | 76.69 ± 0.19 / 7.3 | 77.90 ± 0.65 / 5.9 |
| HyperAttack | 73.29 ± 0.88 / 3.5 | 75.83 ± 1.10 / 0.0 | 77.78 ± 1.16 / 6.1 | 80.35 ± 0.62 / 3.7 | 75.81 ± 2.06 / 8.7 | 79.50 ± 1.28 / 4.1 |
| MeLA-FGSM | 70.37 ± 1.06 / 8.3 | 29.31 ± 1.68 / 60.3 | 74.59 ± 1.12 / 10.8 | 23.90 ± 3.06 / 72.2 | 73.44 ± 0.62 / 11.4 | 27.27 ± 1.91 / 67.2 |
| MeLA-PGD | 68.45 ± 1.01 / **9.3** | 29.28 ± 0.35 / **61.5** | 71.91 ± 0.98 / **14.5** | 22.19 ± 0.42 / **73.5** | 72.64 ± 0.38 / **12.4** | 19.97 ± 0.34 / **75.9** |

*Table 4.* Adversarial-training results for MeLA-D+HyperMLP under poisoning attacks. Gain = avg. accuracy gain over runs.

| Attack | CiteSeer | | Cora-CA | | Cora | |
| --- | --- | --- | --- | --- | --- | --- |
| | HyperMLP | MeLA-D+HyperMLP / Gain | HyperMLP | MeLA-D+HyperMLP / Gain | HyperMLP | MeLA-D+HyperMLP / Gain |
| Rand-Feat | 53.94 ± 1.54 | 69.49 ± 1.19 / 15.56 | 69.69 ± 2.03 | 73.86 ± 1.00 / 4.17 | 62.69 ± 1.12 | 73.29 ± 0.83 / 10.61 |
| Rand-Flip | 70.48 ± 0.31 | 72.80 ± 1.39 / 2.32 | 50.90 ± 2.54 | 73.68 ± 1.16 / 22.78 | 51.73 ± 1.67 | 73.74 ± 0.92 / 22.01 |
| GradArgMax | 70.48 ± 0.31 | 73.12 ± 1.09 / 2.63 | 50.90 ± 2.54 | 73.47 ± 0.73 / 22.57 | 51.73 ± 1.67 | 73.21 ± 0.53 / 21.48 |
| DICE | 70.48 ± 0.31 | 72.80 ± 1.39 / 2.32 | 50.90 ± 2.54 | 73.68 ± 1.16 / 22.78 | 51.73 ± 1.67 | 73.74 ± 0.92 / 22.01 |
| HyperAttack | 70.48 ± 0.31 | 72.80 ± 1.39 / 2.32 | 50.90 ± 2.54 | 73.68 ± 1.16 / 22.78 | 51.73 ± 1.67 | 73.74 ± 0.92 / 22.01 |
| MeLA-FGSM | 47.46 ± 0.65 | 69.61 ± 1.19 / 22.15 | 28.01 ± 0.36 | 63.99 ± 1.43 / 35.98 | 39.05 ± 0.71 | 68.33 ± 0.59 / 29.28 |
| MeLA-PGD | 41.69 ± 0.76 | 69.81 ± 0.75 / **28.12** | 25.05 ± 1.08 | 66.65 ± 1.00 / **41.60** | 38.38 ± 0.38 | 69.51 ± 0.47 / **31.14** |

*Evasion Setting.* (i) MeLA-PGD is the most effective, achieving up to 76% accuracy drop (Cora-CA). (ii) MeLA-FGSM also outperforms the non-adaptive baselines by a large margin. Both MeLA-FGSM and MeLA-PGD optimize a meta-loss evaluated on a fixed surrogate model, which closely matches the evasion setting where the classifier parameters are frozen at test time. This objective alignment allows both attacks to outperform the baselines.

**II. Effectiveness of Defense.** We evaluate the robustness of adversarially trained variant MeLA-D+HyperMLP against all attacks and report the average accuracy gain in Table 4. We observe that MeLA-D consistently improves accuracy under adversarial perturbations. Reinforcing its defensive ability, MeLA-D leads to roughly 22–42% gain against stronger gradient-based attacks such as MeLA-FGSM and MeLA-PGD. Figure 1 further illustrates this robustness gain on MeLA-FGSM perturbed Cora during adversarial training: MeLA-D+HyperMLP consistently yields higher perturbed accuracy than the baseline HyperMLP.

**III. Characterizing Attack Perturbations.** We characterize attack perturbations from two complementary views: operator-level drift, measured by raw Laplacian change,

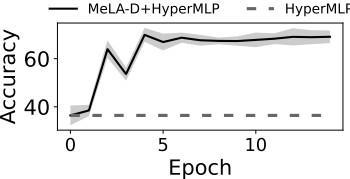

*Figure 1.* Evolution of test accuracy of the robustified HyperMLP and the base HyperMLP during adversarial training.

and feature-level distortion, measured by node-feature cosine similarity. Appendix F.1 provides an additional local-structure analysis through node-degree drift.

**(a) Operator-level:** As the hypergraph Laplacian directly defines the linear operator used in hypergraph message passing, raw Laplacian drift measures how much the attack changes this operator. Figure 2 (left) shows that among structure-perturbing attacks, MeLA-PGD achieves the smallest Laplacian drift, thus it is more operator-preserving. This is not contradictory to the attack objective: MeLA optimizes the drift in Laplacian-smoothed embeddings, whereas here we measure the raw operator perturbation $\|L_{orig} - L_{pert}\|_F$. As the hypergraph Laplacian depends nonlinearly on in-

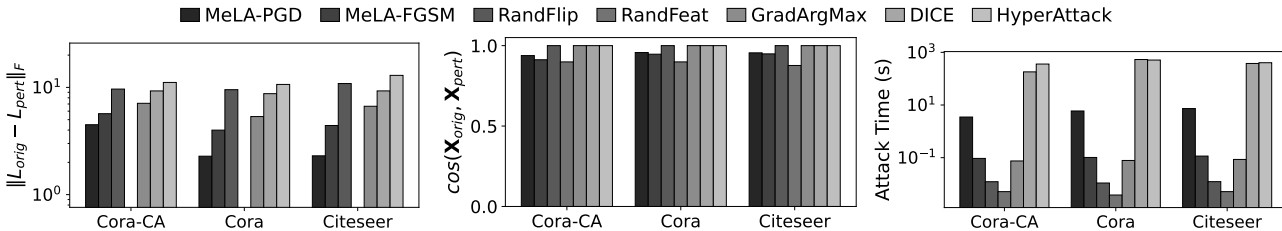

*Figure 2.* Attack perturbation and runtime comparison: Laplacian operator drift (left; lower is more operator-preserving), feature cosine similarity (middle; higher is more feature-preserving), and execution time (right; lower is faster).

verse node and hyperedge degree normalizations, degree drift alone does not determine operator drift. Theorem 2 further supports this distinction by showing that low-degree nodes and small hyperedges can amplify Laplacian sensitivity under the same perturbation budget.

**(b) Semantic Distortion:** We measure the change in semantic/feature distortion using mean cosine similarity between the original and perturbed node features. A high cosine similarity indicates limited distortion of the feature space; therefore, stronger feature preservation. *Figure 2 (middle) shows that MeLA-PGD maintains high similarity due to its use of gradient-aligned feature updates with constrained step sizes.* RandFlip, GradArgMax, DICE, and HyperAttack are structure-only attacks, hence cosine similarity $\sim$ 1. In contrast, RandFeat perturbs feature coordinates independently at random, so the injected noise is not aligned with the original feature direction; this rotates node-feature vectors more strongly, yielding lower cosine similarity than the gradient-based MeLA variants.

**IV. Computational Efficiency.** Figure 2 (right) shows that RandFeat is the most efficient (4–5 ms), while MeLA-FGSM provides a fast gradient-based alternative (90–120 ms). MeLA-PGD is slower than these baselines (3–7 sec), but substantially faster than DICE and HyperAttack.

**V. Adapting MeLA-PGD for Large-scale hypergraphs.** To scale MeLA-PGD, we use a mini-batch variant that optimizes perturbations on local incidence blocks rather than the full $|V| \times |E|$ incidence matrix. For each sampled node batch, we collect the incident hyperedges, update only the corresponding local structure and features, and then write these updates back to global perturbation. This keeps the clean incidence representation sparse while allowing the PGD relaxation to operate on a small dense block.

*Table 5.* Effectiveness of mini-batch MeLA-PGD on large-scale hypergraphs. We report clean accuracy, attacked accuracy, relative % accuracy drop, and execution time.

| Dataset | $|V|$ | $|E|$ | Clean | Evasion / Drop | Exec. time (sec) |
|---|---|---|---|---|---|
| OGBN-MAG | 736,389 | 7,145,660 | $24.18 \pm 0.23$ | $22.11 \pm 0.23$ / 8.6% | $4614.94 \pm 66.63$ |
| Yelp | 50,758 | 4,523,594 | $31.98 \pm 0.40$ | $15.65 \pm 1.11$ / 52.1% | $2036.91 \pm 20.17$ |
| Trivago | 172,738 | 726,861 | $37.27 \pm 1.96$ | $24.01 \pm 1.82$ / 36.3% | $718.61 \pm 3.34$ |
| Walmart | 88,860 | 460,630 | $97.70 \pm 0.06$ | $95.96 \pm 0.31$ / 1.8% | $366.05 \pm 15.27$ |
| Flickr | 7,575 | 479,476 | $89.53 \pm 0.78$ | $35.75 \pm 2.68$ / 58.8% | $113.13 \pm 19.51$ |

We further reduce the structural search space by scoring only a compact candidate set of incidence flips instead of all entries in the local block. The surrogate model is refreshed once per outer PGD iteration on sampled mini-batches and then reused, avoiding full surrogate retraining for every batch. Together, local sparse blocks, candidate-only structural updates, and periodic surrogate refresh make MeLA-PGD practical on hypergraphs with hundreds of thousands of vertices and millions of incidences. Experiments on five large-scale datasets, OGBN-MAG, Yelp, Trivago, Walmart, and Flickr (Chien et al., 2021), are reported in Table 5. The results show that the mini-batch MeLA-PGD remains effective at scale, reducing HGNN accuracy by 1.8–58.8% across datasets.

## 6. Conclusion and Future Directions

We proposed a novel generic meta-objective based framework MeLA for gray-box attacks on HGNNs designed for node classification tasks. We have introduced an adversarial training algorithm MeLA-D that constructs a robust model by harnessing adversarial perturbations generated from the Laplacian-based meta-objective. We proved the convergence of MeLA-D and showed that the new attacks MeLA-FGSM and MeLA-PGD are more effective than the baselines, while the defense yields significant robustness gains. Although MeLA-D is an efficient defense, it does not yield a robustness certification. We plan to investigate ways to close this gap in future work.

*Limitations.* A limitation of our adversarial training strategy is that structural perturbations of the hypergraph may not always preserve label-invariant semantics; consequently, training on perturbed structures with the original labels can potentially induce incorrect structure–label associations (Li et al., 2024). Furthermore, MeLA-PGD still incurs substantial training cost on large-scale hypergraphs.

## Acknowledgement

This work at Howard University was supported by the DoD Center of Excellence in AI/ML (CoE-AIML) at Howard University under Contract W911NF-20-2-0277 with the

U.S. Army Research Laboratory. The work of Yulia R. Gel was supported in part by the U.S. Department of Energy (DOE), Office of Science, Advanced Scientific Computing Research (ASCR) program under the Scientific Discovery through Advanced Computing (SciDAC) Institute "LEADS: LEarning-Accelerated Domain Science". The work of Debabrota Basu was supported in part by the French National Research Agency ANR JCJC project REPUBLIC (ANR-22-CE23-0003-01) and PEPR project FOUNDRY (ANR23-PEIA-0003). However, any opinions, findings, conclusions, or recommendations expressed in this document are those of the authors and should not be interpreted as representing the official policies, either expressed or implied, of the funding agencies.

## Impact Statement

This work advances the study of adversarial robustness for hypergraph neural networks, which are increasingly used to model higher-order relationships in domains such as recommendation, biological networks, and scientific data analysis. By exposing vulnerabilities of HGNNs under realistic gray-box perturbations, the proposed attacks can help researchers and practitioners evaluate failure modes before deploying such models in sensitive applications. At the same time, adversarial attack methods are inherently dual-use: they may be misused to degrade deployed systems if applied without authorization. We therefore frame the attacks as tools for controlled robustness evaluation and pair them with an adversarial training defense. We encourage users of this work to apply the methods only in authorized settings, report robustness results transparently, and consider domain-specific safety, privacy, and fairness implications when HGNNs are used in consequential decision-making workflow.

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

# A. Details of Theoretical Analysis

## A.1. Proof for `MeLA`'s optimal incidence matrix flip selection under $\ell_0$ budget

**Proposition 1.** *Let $\boldsymbol{H} \in \{0,1\}^{n \times m}$ be the binary incidence matrix. Assume the attacker can flip at most $\delta_H$ entries of $\boldsymbol{H}$. For any index set $\Omega \subseteq [n] \times [m]$ with $|\Omega| \leq \delta_H$, define the binary flip operation as $H'_{ij} \triangleq 1 - H_{ij}$, if $(i,j) \in \Omega$, and $H_{ij}$, otherwise. Let $\boldsymbol{M} \in \{0,1\}^{n \times m}$ be $M_{ij} \triangleq \mathbb{I}[(i,j) \in \Omega]$, with $\boldsymbol{H}' \triangleq \boldsymbol{H} + (1 - 2\boldsymbol{H}) \odot \boldsymbol{M}$ and $g_H \triangleq \nabla_H \mathcal{L}(\boldsymbol{H})$ be the gradient of the loss w.r.t. $\boldsymbol{H}$. Then the optimal flip set $\Omega^*$ is top-$\delta_H$ entries of $\{(i,j) : S \triangleq g_H(1 - 2\boldsymbol{H}) > 0\}$.*

*Proof.* By construction,

$$\Delta \boldsymbol{H} = \boldsymbol{H}' - \boldsymbol{H} = (1 - 2\boldsymbol{H}) \odot \boldsymbol{M}.$$

Since $H_{ij} \in \{0,1\}$, we have $(1 - 2H_{ij}) \in \{+1, -1\}$, and $\Delta H_{ij}$ equals $+1$ if we flip a 0 to 1, equals $-1$ if we flip a 1 to 0, and equals 0 if we do not flip.

Next, we linearize the loss using a first-order approximation.

For small perturbations, the first-order Taylor expansion gives

$$\mathcal{L}(\boldsymbol{H}') = \mathcal{L}(\boldsymbol{H}) + \langle \nabla_H \mathcal{L}(\boldsymbol{H}), \Delta \boldsymbol{H} \rangle + o(\|\Delta \boldsymbol{H}\|).$$

Thus, maximizing $\mathcal{L}(\boldsymbol{H}')$ is equivalent to maximizing the linear term $\langle \nabla_H \mathcal{L}(\boldsymbol{H}), \Delta \boldsymbol{H} \rangle = \langle g_H, \Delta \boldsymbol{H} \rangle$.

Substituting $\Delta \boldsymbol{H} = (1 - 2\boldsymbol{H}) \odot \boldsymbol{M}$ yields

$$\langle g_H, \Delta \boldsymbol{H} \rangle = \sum_{i=1}^{n} \sum_{j=1}^{m} g_{ij}(1 - 2H_{ij}) M_{ij} = \sum_{i,j} S_{ij} M_{ij},$$

where $S_{ij} \triangleq g_{ij}(1 - 2H_{ij})$.

The constraint $|\Omega| \leq \delta_H$ is exactly

$$\sum_{i,j} M_{ij} \leq \delta_H, \quad M_{ij} \in \{0,1\}.$$

So the optimization becomes

$$\max_{M \in \{0,1\}^{n \times m}} \sum_{i,j} S_{ij} M_{ij} \quad \text{s.t.} \quad \sum_{i,j} M_{ij} \leq \delta_H.$$

Because each flip-decision variable $M_{ij}$ contributes independently and incurs a cost of 1, the optimal strategy is to (1) never include an index with $S_{ij} \leq 0$, since setting $M_{ij} = 1$ would not increase the objective, and (2) Among the indices with $S_{ij} > 0$, include those with the largest $S_{ij}$ until the budget $\delta_H$ is exhausted.

Thus, the optimal flip set $\Omega^*$ is the set of the top-$\delta_H$ indices among $\{(i,j) : S_{ij} > 0\}$, proving the claim. $\square$

## A.2. Proofs for Impacts of Graph Adversaries

**Definition 1** (Adversary-G: Graph perturbations). *Let $\boldsymbol{A} \in \mathbb{R}^{n \times n}$ be symmetric with $A_{ij} \geq 0$ and $A_{ii} = 0$. Then, the degree matrix be $\boldsymbol{D} = \mathrm{diag}(d_1, \ldots, d_n)$ with $d_i = \sum_j A_{ij}$, the normalized graph Laplacian be*

$$\boldsymbol{L}_{\mathcal{G}} = \boldsymbol{I} - \boldsymbol{S} \boldsymbol{A} \boldsymbol{S},$$

*where $\boldsymbol{S} = \boldsymbol{D}^{-1/2}$. Then, the Adversary-G introduces a symmetric graph perturbation $\Delta \boldsymbol{A}$ satisfying the row-sum perturbation budget*

$$\|\Delta \boldsymbol{A}\|_{\infty} \triangleq \max_i \sum_j |\Delta A_{ij}| \leq \epsilon_{\mathrm{row}},$$

*such that the adjacency matrix, the degree matrix, and the graph Laplacian of the perturbed graph $\mathcal{G}'$ are $\boldsymbol{A}' = \boldsymbol{A} + \Delta \boldsymbol{A}$, $\boldsymbol{D}' = \boldsymbol{D} + \Delta \boldsymbol{D}$, $\boldsymbol{S}' = (\boldsymbol{D}')^{-1/2}$, and $\boldsymbol{L}_{\mathcal{G}'} = \boldsymbol{I} - \boldsymbol{S}' \boldsymbol{A}' \boldsymbol{S}'$, respectively.*

**Theorem 1** (Impact of Adversary–G). *Given an Adversary-G with perturbation budget $\epsilon_{\mathrm{row}} \in (0, d_{\mathrm{max}}]$, we get*

$$\frac{\epsilon_{\mathrm{row}}}{d_{\mathrm{min}} + \epsilon_{\mathrm{row}}}(1 + \kappa(\boldsymbol{D})) \leq \|\boldsymbol{L}_{\mathcal{G}} - \boldsymbol{L}_{\mathcal{G}'}\|_2$$

$$\leq \left( \frac{\epsilon_{\mathrm{row}}}{d_{\mathrm{min}}} + \left( \frac{\epsilon_{\mathrm{row}}}{d_{\mathrm{min}}} \right)^2 \right)(1 + \kappa(\boldsymbol{D})), \tag{6}$$

*where $\kappa(\boldsymbol{D}) \triangleq \frac{d_{max}}{d_{\mathrm{min}}} \triangleq \frac{\max_i d_i}{\min_i d_i}$ is the condition number of $\boldsymbol{D}$ and $d_{\mathrm{min}} > 0$.*

*Proof of Theorem 1.* We develop the proof in two separate parts– Part a corresponds to the upper bound, while Part b establishes the lower bound.

**Part a: Proof of the upper bound.**

**Step 1: Quantifying the perturbation of Laplacian.** Given Adversary-G (Definition 1), we can define the graph Laplacian of the perturbed graph $\mathcal{G}'$ as

$$\begin{aligned}
\boldsymbol{L}_{\mathcal{G}'} &\triangleq \boldsymbol{I} - \boldsymbol{S}' \boldsymbol{A}' \boldsymbol{S}' \\
&= \boldsymbol{I} - (\boldsymbol{S} + \Delta \boldsymbol{S})(\boldsymbol{A} + \Delta \boldsymbol{A})(\boldsymbol{S} + \Delta \boldsymbol{S}) \\
&= \boldsymbol{I} - (\boldsymbol{S} \boldsymbol{A} \boldsymbol{S} + \boldsymbol{S} \boldsymbol{A} \Delta \boldsymbol{S} + \boldsymbol{S} \Delta \boldsymbol{A} \boldsymbol{S} + \Delta \boldsymbol{S} \boldsymbol{A} \boldsymbol{S} \\
&\quad + \boldsymbol{S} \Delta \boldsymbol{A} \Delta \boldsymbol{S} + \Delta \boldsymbol{S} \Delta \boldsymbol{A} \boldsymbol{S} + \Delta \boldsymbol{S} \boldsymbol{A} \Delta \boldsymbol{S} + \Delta \boldsymbol{S} \Delta \boldsymbol{A} \Delta \boldsymbol{S}).
\end{aligned}$$

Here, $\Delta \boldsymbol{S} = \boldsymbol{S}' - \boldsymbol{S} = (\boldsymbol{D} + \Delta \boldsymbol{D})^{-1/2} - \boldsymbol{D}^{-1/2}$.

Therefore, the change in the graph Laplacian is

$$\begin{aligned}
&\boldsymbol{L}_{\mathcal{G}} - \boldsymbol{L}_{\mathcal{G}'} \\
&= \left( \boldsymbol{I} - \boldsymbol{S} \boldsymbol{A} \boldsymbol{S} \right) - \left( \boldsymbol{I} - (\boldsymbol{S} + \Delta \boldsymbol{S})(\boldsymbol{A} + \Delta \boldsymbol{A})(\boldsymbol{S} + \Delta \boldsymbol{S}) \right) \\
&= (\boldsymbol{S} + \Delta \boldsymbol{S})(\boldsymbol{A} + \Delta \boldsymbol{A})(\boldsymbol{S} + \Delta \boldsymbol{S}) - \boldsymbol{S} \boldsymbol{A} \boldsymbol{S} \\
&= \boldsymbol{S} \Delta \boldsymbol{A} \boldsymbol{S} + \Delta \boldsymbol{S} \boldsymbol{A} \boldsymbol{S} + \boldsymbol{S} \boldsymbol{A} \Delta \boldsymbol{S} + \boldsymbol{R}. \tag{7}
\end{aligned}$$

The residual $\boldsymbol{R}$ collects the terms containing at least two perturbation factors:

$$\boldsymbol{R} \triangleq \boldsymbol{S}\Delta\boldsymbol{A}\Delta\boldsymbol{S} + \Delta\boldsymbol{S}\Delta\boldsymbol{A}\boldsymbol{S} + \Delta\boldsymbol{S}\boldsymbol{A}\Delta\boldsymbol{S} + \Delta\boldsymbol{S}\Delta\boldsymbol{A}\Delta\boldsymbol{S}. \tag{8}$$

**Step 2: Upper bounding the norms of the perturbations.**
First, using the triangle inequality and sub-multiplicative property of spectral norms, we get

$$\|\boldsymbol{L}_{\mathcal{G}} - \boldsymbol{L}_{\mathcal{G}'}\|_2$$
$$\leq \|\boldsymbol{S}\Delta\boldsymbol{A}\boldsymbol{S}\|_2 + \|\Delta\boldsymbol{S}\boldsymbol{A}\boldsymbol{S}\|_2 + \|\boldsymbol{S}\boldsymbol{A}\Delta\boldsymbol{S}\|_2 + \|\boldsymbol{R}\|_2$$
$$\leq \|\boldsymbol{S}\|_2^2\|\Delta\boldsymbol{A}\|_2 + 2\|\Delta\boldsymbol{S}\|_2\|\boldsymbol{A}\|_2\|\boldsymbol{S}\|_2 + \|\boldsymbol{R}\|_2. \tag{9}$$

Now, we derive the norm bounds on adjacency matrices and $\boldsymbol{S}$'s for the original graph $\mathcal{G}$ and the Adversary-G, i.e.,

$$\|\boldsymbol{S}\|_2 = d_{\min}^{-1/2}, \quad \|\boldsymbol{A}\|_2 \leq d_{\max}, \qquad \text{(Lemma 2)}$$
$$\|\Delta\boldsymbol{A}\|_2 \leq \epsilon_{\text{row}}, \qquad\qquad\qquad \text{(Lemma 3)}$$
$$\|\Delta\boldsymbol{S}\|_2 \leq \frac{1}{2}d_{\min}^{-3/2}\epsilon_{\text{row}}. \qquad\qquad \text{(Lemma 4)}$$

Thus, from Equation (9), we get

$$\|\boldsymbol{L}_{\mathcal{G}} - \boldsymbol{L}_{\mathcal{G}'}\|_2 \leq \frac{\epsilon_{\text{row}}}{d_{\min}} + \frac{d_{\max}\epsilon_{\text{row}}}{d_{\min}^2} + \|\boldsymbol{R}\|_2$$
$$\triangleq \frac{\epsilon_{\text{row}}}{d_{\min}}(1 + \kappa(\boldsymbol{D})) + \|\boldsymbol{R}\|_2. \tag{10}$$

Now, we bound $\|\boldsymbol{R}\|_2$ (Equation (8)) term-by-term using again the triangle inequality and sub-multiplicative property of spectral norms.

$$\|\boldsymbol{R}\|_2$$
$$\leq \|\boldsymbol{S}\Delta\boldsymbol{A}\Delta\boldsymbol{S}\|_2 + \|\Delta\boldsymbol{S}\Delta\boldsymbol{A}\boldsymbol{S}\|_2 + \|\Delta\boldsymbol{S}\boldsymbol{A}\Delta\boldsymbol{S}\|_2$$
$$+ \|\Delta\boldsymbol{S}\Delta\boldsymbol{A}\Delta\boldsymbol{S}\|_2$$
$$\leq \|\boldsymbol{S}\|_2\|\Delta\boldsymbol{A}\|_2\|\Delta\boldsymbol{S}\|_2 + \|\Delta\boldsymbol{S}\|_2\|\Delta\boldsymbol{A}\|_2\|\boldsymbol{S}\|_2$$
$$+ \|\Delta\boldsymbol{S}\|_2^2\|\boldsymbol{A}\|_2 + \|\Delta\boldsymbol{S}\|_2^2\|\Delta\boldsymbol{A}\|_2$$
$$= 2\|\boldsymbol{S}\|_2\|\Delta\boldsymbol{A}\|_2\|\Delta\boldsymbol{S}\|_2 + \|\boldsymbol{A}\|_2\|\Delta\boldsymbol{S}\|_2^2 + \|\Delta\boldsymbol{A}\|_2\|\Delta\boldsymbol{S}\|_2^2$$
$$\leq 2(d_{\min}^{-1/2})(\epsilon_{\text{row}})\left(\frac{1}{2}d_{\min}^{-3/2}\epsilon_{\text{row}}\right) + d_{\max}\left(\frac{1}{2}d_{\min}^{-3/2}\epsilon_{\text{row}}\right)^2$$
$$+ (\epsilon_{\text{row}})\left(\frac{1}{2}d_{\min}^{-3/2}\epsilon_{\text{row}}\right)^2$$
$$= \left(\frac{\epsilon_{\text{row}}}{d_{\min}}\right)^2\left(1 + \frac{1}{4}\kappa(\boldsymbol{D}) + \frac{1}{4}\frac{\epsilon_{\text{row}}}{d_{\min}}\right)$$
$$\leq \left(\frac{\epsilon_{\text{row}}}{d_{\min}}\right)^2\left(1 + \frac{1}{2}\kappa(\boldsymbol{D})\right). \tag{11}$$

The last inequality holds for any $\epsilon_{\text{row}} \leq d_{\max}$.

Now, combining Equation (10) and (11), we obtain

$$\|\boldsymbol{L}_{\mathcal{G}} - \boldsymbol{L}_{\mathcal{G}'}\|_2$$

$$\leq \frac{\epsilon_{\text{row}}}{d_{\min}}(1 + \kappa(\boldsymbol{D})) + \left(\frac{\epsilon_{\text{row}}}{d_{\min}}\right)^2\left(1 + \frac{1}{2}\kappa(\boldsymbol{D})\right)$$
$$\leq \left(\frac{\epsilon_{\text{row}}}{d_{\min}} + \left(\frac{\epsilon_{\text{row}}}{d_{\min}}\right)^2\right)(1 + \kappa(\boldsymbol{D})).$$

This concludes the proof of the upper bound.

**Part b: Proof of the lower bound.**

We derive an existential lower bound through an explicit construction of the graph and the perturbations.

**Step 1: Base graph construction.** Let $n = 4$ and label the vertices $\{1, 2, 3, 4\}$. Fix any $d > 0$ and define the base adjacency matrix $\boldsymbol{A}$ by setting

$$A_{ij} = \begin{cases} d/3, & i \neq j, \\ 0, & i = j. \end{cases}$$

Then $\boldsymbol{A}$ is symmetric, entrywise nonnegative, and has zero diagonal. $\boldsymbol{A}$ can be written as

$$\boldsymbol{A} = \frac{d}{3}(\mathbf{1}\mathbf{1}^\top - \boldsymbol{I})$$

For each node $i$ the degree is

$$d_i = \sum_{j=1}^{4} A_{ij} = \sum_{\substack{j=1 \\ j \neq i}}^{4} \frac{d}{3} = 3 \cdot \frac{d}{3} = d.$$

Hence

$$d_{\min} = \min_i d_i = d,$$

$$\boldsymbol{D} = \text{diag}(d_1, d_2, d_3, d_4),$$

with condition number $\kappa(\boldsymbol{D}) = \frac{d}{d} = 1$, and

$$\boldsymbol{S} = \boldsymbol{D}^{-1/2} = d^{-1/2}\boldsymbol{I}.$$

**Step 2: Defining perturbation.** Let us fix

$$t \triangleq \frac{\epsilon_{\text{row}}}{2}.$$

Let us define $\Delta\boldsymbol{A}$ to be supported on the 4-cycle $(1 - 2, 2 - 3, 3 - 4, 4 - 1)$ by $\Delta A_{12} = \Delta A_{21} = t$, $\Delta A_{23} = \Delta A_{32} = t$, $\Delta A_{34} = \Delta A_{43} = t$, $\Delta A_{41} = \Delta A_{14} = t$, and $\Delta A_{ij} = 0$ for all other pairs $(i, j)$, including $\Delta A_{ii} = 0$.

**(i) $\Delta\boldsymbol{A}$ is $\ell_\infty$ budget satisfying.** For node 1, the only nonzero entries in row 1 are $\Delta A_{12} = t$ and $\Delta A_{14} = t$, so

$$\sum_{j=1}^{4} |\Delta A_{1j}| = |t| + |t| = 2|t| = \epsilon_{\text{row}}.$$

Similarly, for node 2, the only nonzero entries are $\Delta A_{21} = t$ and $\Delta A_{23} = t$, so

$$\sum_{j=1}^{4} |\Delta A_{2j}| = |t| + |t| = 2|t| = \epsilon_{\text{row}}.$$

The same computation holds for vertices 3 and 4. Therefore,

$$\|\Delta \boldsymbol{A}\|_{\infty} = \max_{i} \sum_{j} |\Delta A_{ij}| = \epsilon_{\text{row}}.$$

**(ii) $\Delta \boldsymbol{D} = \epsilon_{\text{row}} \boldsymbol{I}$.** For each $i$, $\Delta D_{ii}$ equals the row sum of $\Delta \boldsymbol{A}$. Hence,

$$D_{ii} = \sum_{j=1}^{4} \Delta A_{ij} = 2t = \epsilon_{\text{row}}.$$

implying, $\Delta \boldsymbol{D} = \epsilon_{\text{row}} \boldsymbol{I}$

**Step 3: Expressing $\boldsymbol{L}_{\mathcal{G}} - \boldsymbol{L}_{\mathcal{G}'}$.** Let us define the perturbed adjacency $\boldsymbol{A}' = \boldsymbol{A} + \Delta \boldsymbol{A}$. Then the perturbed $\boldsymbol{D}'$ is

$$\boldsymbol{D}' = \boldsymbol{D} + \Delta \boldsymbol{D} = (d_{\min} + \epsilon_{\text{row}}) \boldsymbol{I}.$$

Therefore

$$\boldsymbol{S}' = (\boldsymbol{D}')^{-1/2} = \boldsymbol{D}^{-1/2} = (d_{\min} + \epsilon_{\text{row}})^{-1/2} \boldsymbol{I}.$$

Hence the Laplacian difference

$$\begin{aligned}
\boldsymbol{L}_{\mathcal{G}} - \boldsymbol{L}_{\mathcal{G}'} &= (\boldsymbol{I} - \boldsymbol{S}\boldsymbol{A}\boldsymbol{S}) - (\boldsymbol{I} - \boldsymbol{S}'\boldsymbol{A}'\boldsymbol{S}') \\
&= \boldsymbol{S}'\boldsymbol{A}'\boldsymbol{S}' - \boldsymbol{S}\boldsymbol{A}\boldsymbol{S} \\
&= \frac{1}{(d_{\min} + \epsilon_{\text{row}})} \boldsymbol{A}' - \frac{1}{d_{\min}} \boldsymbol{A} \\
&= \left( \frac{1}{(d_{\min} + \epsilon_{\text{row}})} - \frac{1}{d_{\min}} \right) \boldsymbol{A} + \frac{1}{(d_{\min} + \epsilon_{\text{row}})} \Delta \boldsymbol{A} \\
&= \frac{\epsilon_{\text{row}}}{d_{\min}(d_{\min} + \epsilon_{\text{row}})} \boldsymbol{A} + \frac{1}{(d_{\min} + \epsilon_{\text{row}})} \Delta \boldsymbol{A}
\end{aligned}$$

**Step 4: Lower-bound via quadratic form.** Since $\boldsymbol{L}_G - \boldsymbol{L}'_G = \boldsymbol{S}\Delta \boldsymbol{A}\boldsymbol{S}$ is symmetric, the variational characterization gives

$$\begin{aligned}
&\|\boldsymbol{L}_{\mathcal{G}} - \boldsymbol{L}_{\mathcal{G}'}\|_2 \\
&= \max_{\|\boldsymbol{x}\|_2=1} \left| \boldsymbol{x}^{\top} \left( \frac{\epsilon_{\text{row}}}{d_{\min}(d_{\min} + \epsilon_{\text{row}})} \boldsymbol{A} + \frac{1}{(d_{\min} + \epsilon_{\text{row}})} \Delta \boldsymbol{A} \right) \boldsymbol{x} \right| \\
&\geq \left| \frac{\epsilon_{\text{row}}}{d_{\min}(d_{\min} + \epsilon_{\text{row}})} \boldsymbol{x}^{\top} \boldsymbol{A} \boldsymbol{x} + \frac{1}{(d_{\min} + \epsilon_{\text{row}})} \boldsymbol{x}^{\top} \Delta \boldsymbol{A} \boldsymbol{x} \right| \\
&= \left| \alpha \, \boldsymbol{x}^{\top} \boldsymbol{A} \boldsymbol{x} + \beta \, \boldsymbol{x}^{\top} \Delta \boldsymbol{A} \boldsymbol{x} \right|,
\end{aligned}$$

where

$$\alpha \triangleq \frac{\epsilon_{\text{row}}}{d_{\min}(d_{\min} + \epsilon_{\text{row}})},$$

$$\beta \triangleq \frac{1}{d_{\min} + \epsilon_{\text{row}}}.$$

We now choose a witness vector

$$\boldsymbol{x} \triangleq \frac{1}{2} \begin{bmatrix} 1 \\ 1 \\ 1 \\ 1 \end{bmatrix},$$

which is a unit vector, and evaluate the quadratic forms $\boldsymbol{x}^{\top} \boldsymbol{A} \boldsymbol{x}$ and $\boldsymbol{x}^{\top} \Delta \boldsymbol{A} \boldsymbol{x}$.

*Step 4a: Evaluating $\boldsymbol{x}^{\top} \boldsymbol{A} \boldsymbol{x}$.*

$$\begin{aligned}
\boldsymbol{x}^{\top} \boldsymbol{A} \boldsymbol{x} &= \frac{d}{3} (\boldsymbol{x}^{\top} \boldsymbol{1} \boldsymbol{1}^{\top} \boldsymbol{x} - \boldsymbol{x}^{\top} \boldsymbol{I} \boldsymbol{x}) \\
&= \frac{d}{3} ((\boldsymbol{x}^{\top} \boldsymbol{1})(\boldsymbol{1}^{\top} \boldsymbol{x}) - \boldsymbol{x}^{\top} \boldsymbol{x}) \\
&= \frac{d}{3} ((\boldsymbol{1}^{\top} \boldsymbol{x})^2 - (\|\boldsymbol{x}\|)^2) \\
&= \frac{d}{3} ((\sum_{i=1}^{4} x_i)^2 - 1) \\
&= \frac{d}{3} (4 - 1) = d = d_{\min}
\end{aligned}$$

*Step 4b: Evaluating $\boldsymbol{x}^{\top} \Delta \boldsymbol{A} \boldsymbol{x}$.* Because $\Delta \boldsymbol{A}$ has non-zero value $(t)$ only on $(1, 2), (2, 3), (3, 4), (4, 1)$ along with their symmetric counterparts, and $x_i = 1/2$ for all $i$, we have

$$\begin{aligned}
\boldsymbol{x}^{\top} \Delta \boldsymbol{A} \boldsymbol{x} &= \sum_{i=1}^{4} \sum_{j=1}^{4} x_i \, \Delta A_{ij} \, x_j \\
&= \tfrac{1}{4} \sum_{i=1}^{4} \sum_{j=1}^{4} \Delta A_{ij} \\
&= \tfrac{1}{4} \cdot 2 \Big( \Delta A_{12} + \Delta A_{23} + \Delta A_{34} + \Delta A_{41} \Big) \\
&= \tfrac{1}{2} \cdot 4t = 2t = \epsilon_{\text{row}}.
\end{aligned}$$

Hence

$$\begin{aligned}
&\|\boldsymbol{L}_{\mathcal{G}} - \boldsymbol{L}_{\mathcal{G}'}\|_2 \\
&\geq |\alpha d_{\min} + \beta \epsilon_{\text{row}}| \\
&= \left| \frac{\epsilon_{\text{row}}}{d_{\min}(d_{\min} + \epsilon_{\text{row}})} d_{\min} + \frac{1}{(d_{\min} + \epsilon_{\text{row}})} \epsilon_{\text{row}} \right| \\
&= \frac{2\epsilon_{\text{row}}}{d_{\min} + \epsilon_{\text{row}}} \\
&= \frac{\epsilon_{\text{row}}}{d_{\min} + \epsilon_{\text{row}}} (1 + \kappa(\boldsymbol{D}))
\end{aligned}$$

$\square$

### A.2.1. TECHNICAL RESULTS FOR GRAPHS

**Lemma 2** (Spectral norm of adjacency)**.** *Let $\boldsymbol{A} \in \mathbb{R}^{n \times n}$ be symmetric with $A_{ij} \geq 0$ and $A_{ii} = 0$. Let $d_{\max} =$*

$\max_i \sum_j A_{ij}$. *Then*

$$\|\boldsymbol{A}\|_2 \leq d_{\max}.$$

*Proof.* Since $\boldsymbol{A}$ is symmetric, its spectral norm equals its largest eigenvalue; therefore, by the variational characterization of the Rayleigh quotient:

$$\|\boldsymbol{A}\|_2 = \lambda_{\max}(\boldsymbol{A}) = \max_{\|\boldsymbol{x}\|_2 = 1} \boldsymbol{x}^\top \boldsymbol{A} \boldsymbol{x}.$$

For any unit vector $\boldsymbol{x} \in \mathbb{R}^n$,

$$
\begin{aligned}
\boldsymbol{x}^\top \boldsymbol{A} \boldsymbol{x} &= \sum_{i,j} A_{ij} x_i x_j \\
&\leq \sum_{i,j} A_{ij} \tfrac{1}{2}(x_i^2 + x_j^2) \quad \text{(by } |ab| \leq \tfrac{a^2+b^2}{2}) \\
&= \sum_i x_i^2 \sum_j A_{ij} \\
&= \sum_i d_i x_i^2 \\
&\leq \sum_i d_{\max} x_i^2 \\
&\leq d_{\max} \sum_i x_i^2 \\
&= d_{\max},
\end{aligned}
$$

where the last equality uses $\|\boldsymbol{x}\|_2 = 1$. Taking the maximum over all such $\boldsymbol{x}$ yields $\|\boldsymbol{A}\|_2 = \lambda_{\max}(\boldsymbol{A}) \leq d_{\max}$. $\quad\square$

**Lemma 3** (Bounding $\|\Delta \boldsymbol{A}\|_2$ under row-sum constraint). *Let $\Delta \boldsymbol{A} \in \mathbb{R}^{n \times n}$ be symmetric. If*

$$\|\Delta \boldsymbol{A}\|_\infty \leq \epsilon_{\text{row}},$$

*then*

$$\|\Delta \boldsymbol{A}\|_2 \leq \epsilon_{\text{row}}.$$

*Proof.* For any matrix $M$,

$$\|M\|_2 \leq \sqrt{\|M\|_1 \|M\|_\infty}.$$

Since $\Delta \boldsymbol{A}$ is symmetric, $\|\Delta \boldsymbol{A}\|_1 = \|\Delta \boldsymbol{A}\|_\infty$. Hence,

$$\|\Delta \boldsymbol{A}\|_2 \leq \sqrt{\|\Delta \boldsymbol{A}\|_\infty^2} = \|\Delta \boldsymbol{A}\|_\infty \leq \epsilon_{\text{row}}.$$

$\quad\square$

**Lemma 4** (Bounding perturbation to $\boldsymbol{S} = \boldsymbol{D}^{-1/2}$ under row-sum constraint). *Let $\boldsymbol{S} = \boldsymbol{D}^{-1/2}$ and $\boldsymbol{S}' = (\boldsymbol{D} + \Delta \boldsymbol{D})^{-1/2}$, with $d_{\min} = \min_i d_i > 0$. If*

$$\|\Delta \boldsymbol{A}\|_\infty \leq \epsilon_{\text{row}},$$

*then*

$$\|\Delta \boldsymbol{S}\|_2 \leq \frac{1}{2} d_{\min}^{-3/2} \epsilon_{\text{row}}.$$

*Proof.* Since $\boldsymbol{S}$ and $\boldsymbol{S}'$ are diagonal,

$$(\Delta S)_{ii} = (d_i + \Delta d_i)^{-1/2} - d_i^{-1/2}.$$

Let $f(x) = x^{-1/2}$ on $[d_{\min}, \infty)$. By the mean value theorem,

$$|(\Delta S)_{ii}| \leq \sup_{x \geq d_{\min}} |f'(x)| \, |\Delta d_i| = \frac{1}{2} d_{\min}^{-3/2} \max_i |\Delta d_i|.$$

Since

$$|\Delta d_i| = \left| \sum_j \Delta A_{ij} \right| \leq \sum_j |\Delta A_{ij}| \leq \|\Delta \boldsymbol{A}\|_\infty \leq \epsilon_{\text{row}},$$

taking the maximum over $i$ yields the claim. $\quad\square$

### A.3. Proofs for Impacts of Hypergraph Adversaries

**Definition 2** (Adversary-H: Hypergraph perturbations). *Let $\boldsymbol{H} \in \mathbb{R}_{\geq 0}^{n \times m}$ be a hypergraph incidence matrix. The hyperedge degrees and node degrees are defined by*

$$\delta_e \triangleq \sum_{v=1}^n H_{ve}, \qquad d_v \triangleq \sum_{e=1}^m H_{ve}.$$

*Let*

$$
\begin{aligned}
\boldsymbol{D}_e &\triangleq \text{diag}(\delta_1, \ldots, \delta_m), \\
\boldsymbol{D}_v &\triangleq \text{diag}(d_1, \ldots, d_n), \\
\boldsymbol{S} &\triangleq \boldsymbol{D}_v^{-1/2},
\end{aligned}
$$

*and define the normalized hypergraph Laplacian*

$$\boldsymbol{L_H} \triangleq \boldsymbol{I} - \boldsymbol{S} \boldsymbol{H} \boldsymbol{D}_e^{-1} \boldsymbol{H}^\top \boldsymbol{S}.$$

*Let us assume*

$$d_{\min} \triangleq \min_v d_v > 0, \qquad \delta_{\min} \triangleq \min_e \delta_e > 0.$$

*Given an adversary-H with $\boldsymbol{H}' = \boldsymbol{H} + \Delta \boldsymbol{H}$ with perturbation satisfying the row-sum and column-sum budgets*

$$
\begin{aligned}
\|\Delta \boldsymbol{H}\|_\infty &\triangleq \max_v \sum_e |\Delta H_{ve}| \leq \epsilon_{\text{row}}, \\
\|\Delta \boldsymbol{H}\|_1 &\triangleq \max_e \sum_v |\Delta H_{ve}| \leq \epsilon_{\text{col}}, \quad (12)
\end{aligned}
$$

*for some $\epsilon_{\text{row}} \in (0, d_{\max}], \epsilon_{\text{col}} \in (0, \delta_{\max}]$.*

*Let $\boldsymbol{D}_v' = \boldsymbol{D}_v + \Delta \boldsymbol{D}_v$ and $\boldsymbol{D}_e' = \boldsymbol{D}_e + \Delta \boldsymbol{D}_e$ be the perturbed degree matrices induced by $\boldsymbol{H}'$, let $\boldsymbol{S}' = (\boldsymbol{D}_v')^{-1/2}$, and define*

$$
\begin{aligned}
\boldsymbol{A} &\triangleq \boldsymbol{H} \boldsymbol{D}_e^{-1} \boldsymbol{H}^\top, \\
\boldsymbol{A}' &\triangleq \boldsymbol{H}' (\boldsymbol{D}_e')^{-1} (\boldsymbol{H}')^\top, \\
\Delta \boldsymbol{A} &\triangleq \boldsymbol{A}' - \boldsymbol{A}.
\end{aligned}
$$

**Theorem 2** (Impact of Adversary-H on Hypergraph Laplacian). *Let us assume the perturbation remain unnoticeable, in particular*

$$d'_{\min} \triangleq \min_v d'_v \geq d_{\min}, \qquad \delta'_{\min} \triangleq \min_e \delta'_e \geq \delta_{\min}.$$

*Then*

$$2 \frac{\sqrt{\epsilon_{\text{col}} \epsilon_{\text{row}}}}{\sqrt{d_{\min} \delta_{\min}}} + \frac{\epsilon_{\text{col}} \epsilon_{\text{row}}}{d_{\min} \delta_{\min}} \leq \|\boldsymbol{L_H} - \boldsymbol{L_{H'}}\|_2$$

$$\leq \left( 2 \frac{\sqrt{\epsilon_{\text{col}} \epsilon_{\text{row}}}}{\sqrt{d_{\min} \delta_{\min}}} + \frac{\epsilon_{\text{col}} \epsilon_{\text{row}}}{d_{\min} \delta_{\min}} \right) \left( (1 + \frac{\epsilon_{\text{row}}}{d_{\min}}) \sqrt{\kappa_d \kappa_\delta} + \kappa_d \kappa_\delta \right)$$

$$\tag{13}$$

*where $\kappa_d \triangleq \frac{d_{\max}}{d_{\min}}$ and $\kappa_\delta \triangleq \frac{\delta_{\max}}{\delta_{\min}}$.*

*Proof of the main theorem.* We develop the proof in two separate parts– Part a corresponds to the upper bound, while Part b establishes the lower bound.

**Part a: Proof of the upper bound.**

**Step 1: Quantifying the perturbation of Laplacian.** The hypergraph laplacian matrices of $\boldsymbol{H}$ and $\boldsymbol{H'}$ are

$$\boldsymbol{L_H} = \boldsymbol{I} - \boldsymbol{SAS}, \qquad \boldsymbol{L_{H'}} = \boldsymbol{I} - \boldsymbol{S'A'S'}.$$

Then

$$\boldsymbol{L_H} - \boldsymbol{L_{H'}} = \boldsymbol{S'A'S'} - \boldsymbol{SAS}.$$

Add and subtract $\boldsymbol{SA'S}$ and $\boldsymbol{S'A'S}$:

$$\boldsymbol{S'A'S'} - \boldsymbol{SAS} = \boldsymbol{S'A'S'} - \boldsymbol{SA'S} + \boldsymbol{SA'S}$$
$$- \boldsymbol{S'A'S} + \boldsymbol{S'A'S} - \boldsymbol{SAS}$$
$$= \boldsymbol{S}(\boldsymbol{A'} - \boldsymbol{A})\boldsymbol{S} + (\boldsymbol{S'} - \boldsymbol{S})\boldsymbol{A'S} +$$
$$\boldsymbol{S'A'}(\boldsymbol{S'} - \boldsymbol{S})$$
$$= \boldsymbol{S}(\Delta\boldsymbol{A})\boldsymbol{S} + (\Delta\boldsymbol{S})\boldsymbol{A'S} +$$
$$\boldsymbol{S'A'}(\Delta\boldsymbol{S}).$$

Taking $\|\cdot\|_2$ and using the triangle inequality yields

$$\|\boldsymbol{L_H} - \boldsymbol{L_{H'}}\|_2 = \|\boldsymbol{SAS} - \boldsymbol{S'A'S'}\|_2$$
$$\leq \|\boldsymbol{S}(\Delta\boldsymbol{A})\boldsymbol{S}\|_2 + \|(\Delta\boldsymbol{S})\boldsymbol{A'S}\|_2 +$$
$$\|\boldsymbol{S'A'}(\Delta\boldsymbol{S})\|_2.$$

Using submultiplicativity on each term gives

$$\|\boldsymbol{S}(\Delta\boldsymbol{A})\boldsymbol{S}\|_2 \leq \|\boldsymbol{S}\|_2^2 \|\Delta\boldsymbol{A}\|_2,$$
$$\|(\Delta\boldsymbol{S})\boldsymbol{A'S}\|_2 \leq \|\Delta\boldsymbol{S}\|_2 \|\boldsymbol{A'}\|_2 \|\boldsymbol{S}\|_2,$$
$$\|\boldsymbol{S'A'}(\Delta\boldsymbol{S})\|_2 \leq \|\boldsymbol{S'}\|_2 \|\boldsymbol{A'}\|_2 \|\Delta\boldsymbol{S}\|_2.$$

Summing these yields

$$\|\boldsymbol{L_H} - \boldsymbol{L_{H'}}\|_2 \leq$$
$$\|\boldsymbol{S}\|_2^2 \|\Delta\boldsymbol{A}\|_2 + (\|\boldsymbol{S}\|_2 + \|\boldsymbol{S'}\|_2) \|\boldsymbol{A'}\|_2 \|\Delta\boldsymbol{S}\|_2. \tag{14}$$

Since $\boldsymbol{S} = \boldsymbol{D_v}^{-1/2}$ diagonal with positive entries, its operator norm equals its largest diagonal entry:

$$\|\boldsymbol{S}\|_2 = \max_v d_v^{-1/2}.$$

The maximum of $d_v^{-1/2}$ occurs at the minimum degree $d_{\min}$, hence

$$\|\boldsymbol{S}\|_2 = d_{\min}^{-1/2}.$$

Similarly,

$$\|\boldsymbol{S'}\|_2 = (d'_{\min})^{-1/2}.$$

Under the assumption $d'_{\min} \geq d_{\min}$,

$$\|\boldsymbol{S'}\|_2 = (d'_{\min})^{-1/2} \leq d_{\min}^{-1/2}.$$

Therefore,

$$\|\boldsymbol{S}\|_2^2 = d_{\min}^{-1}, \qquad \|\boldsymbol{S}\|_2 + \|\boldsymbol{S'}\|_2 \leq 2\, d_{\min}^{-1/2}. \tag{15}$$

$$\|\boldsymbol{L_H} - \boldsymbol{L_{H'}}\|_2 \leq \|\boldsymbol{S}\|_2^2 \|\Delta\boldsymbol{A}\|_2 + (\|\boldsymbol{S}\|_2 + \|\boldsymbol{S'}\|_2) \|\boldsymbol{A'}\|_2 \|\Delta\boldsymbol{S}\|_2$$
$$\leq \|\boldsymbol{S}\|_2^2 \|\Delta\boldsymbol{A}\|_2 + (\|\boldsymbol{S}\|_2 + \|\boldsymbol{S'}\|_2) (\|\boldsymbol{A}\|_2 + \|\Delta\boldsymbol{A}\|_2) \|\Delta\boldsymbol{S}\|_2$$
$$\leq d_{\min}^{-1} \|\Delta\boldsymbol{A}\|_2 + (2\, d_{\min}^{-1/2} \|\boldsymbol{A}\|_2 + 2\, d_{\min}^{-1/2} \|\Delta\boldsymbol{A}\|_2) \|\Delta\boldsymbol{S}\|_2$$
$$= d_{\min}^{-1} \|\Delta\boldsymbol{A}\|_2 + 2\, d_{\min}^{-1/2} \|\boldsymbol{A}\|_2 \|\Delta\boldsymbol{S}\|_2 + 2\, d_{\min}^{-1/2} \|\Delta\boldsymbol{A}\|_2 \|\Delta\boldsymbol{S}\|_2$$

In the following, let us assume $\varepsilon \triangleq \frac{\epsilon_{\text{col}} \epsilon_{\text{row}}}{d_{\min} \delta_{\min}}$. Furthermore, we will use the fact that $\frac{\epsilon_{\text{row}}}{d_{\max}} \leq 1$ and $\frac{\epsilon_{\text{col}}}{\delta_{\max}} \leq 1$.

**(i) The first term:**

$$\frac{1}{d_{\min}} \|\Delta\boldsymbol{A}\|_2 \leq$$
$$\frac{1}{d_{\min}} \left( \frac{2\sqrt{\epsilon_{\text{col}} \epsilon_{\text{row}}}}{\delta_{\min}} \|\boldsymbol{H}\|_2 + \frac{\epsilon_{\text{col}}}{\delta_{\min}^2} \|\boldsymbol{H}\|_2^2 + \frac{\epsilon_{\text{col}} \epsilon_{\text{row}}}{\delta_{\min}} \right)$$
$$= \frac{2\sqrt{\epsilon_{\text{col}} \epsilon_{\text{row}}}}{d_{\min} \delta_{\min}} \|\boldsymbol{H}\|_2 + \frac{\epsilon_{\text{col}}}{d_{\min} \delta_{\min}^2} \|\boldsymbol{H}\|_2^2 + \frac{\epsilon_{\text{col}} \epsilon_{\text{row}}}{d_{\min} \delta_{\min}}$$
$$\leq \frac{2\sqrt{\epsilon_{\text{col}} \epsilon_{\text{row}}}}{d_{\min} \delta_{\min}} (\sqrt{\delta_{\max} d_{\max}}) + \frac{\delta_{\max} d_{\max}}{d_{\min} \delta_{\min}^2} \epsilon_{\text{col}} + \frac{\epsilon_{\text{col}} \epsilon_{\text{row}}}{d_{\min} \delta_{\min}}$$
$$= 2 \frac{\sqrt{\epsilon_{\text{col}} \epsilon_{\text{row}}}}{\sqrt{d_{\min} \delta_{\min}}} \sqrt{\kappa_d \kappa_\delta} + \frac{\kappa_d \kappa_\delta}{\delta_{\min}} \epsilon_{\text{col}} + \frac{\epsilon_{\text{col}} \epsilon_{\text{row}}}{d_{\min} \delta_{\min}}$$
$$= 2\sqrt{\kappa_d \kappa_\delta} \sqrt{\varepsilon} + \kappa_d \kappa_\delta \frac{\epsilon_{\text{col}}}{\delta_{\min}} + \varepsilon$$

**(ii) The second term:** From Lemma (6)

$$\|\Delta\boldsymbol{S}\|_2 \leq \frac{1}{2} d_{\min}^{-3/2} \epsilon_{\text{row}}.$$

Thus

$$2 \frac{\|\boldsymbol{A}\|_2}{\sqrt{d_{\min}}} \|\Delta\boldsymbol{S}\|_2 \leq \frac{\|\boldsymbol{A}\|_2}{d_{\min}^2} \epsilon_{\text{row}} \leq \frac{1}{d_{\min}^2} \frac{d_{\max} \delta_{\max}}{\delta_{\min}} \epsilon_{\text{row}}$$
$$= \kappa_d \kappa_\delta \frac{\epsilon_{\text{row}}}{d_{\min}}$$

**(iii) The third term:**

$2d_{\min}^{-1/2}\,\|\Delta\boldsymbol{A}\|_2\,\|\Delta\boldsymbol{S}\|_2$

$\leq 2d_{\min}^{-1/2}\left(\dfrac{2\sqrt{\epsilon_{\text{col}}\epsilon_{\text{row}}}}{\delta_{\min}}\,\|\boldsymbol{H}\|_2+\dfrac{\epsilon_{\text{col}}}{\delta_{\min}^2}\,\|\boldsymbol{H}\|_2^2+\dfrac{\epsilon_{\text{col}}\epsilon_{\text{row}}}{\delta_{\min}}\right)\tfrac{1}{2}d_{\min}^{-3/2}\epsilon_{\text{row}}$

$=\left(\dfrac{2\sqrt{\epsilon_{\text{col}}\epsilon_{\text{row}}}}{d_{\min}^2\delta_{\min}}\,\|\boldsymbol{H}\|_2+\dfrac{\epsilon_{\text{col}}}{d_{\min}^2\delta_{\min}^2}\,\|\boldsymbol{H}\|_2^2+\dfrac{\epsilon_{\text{col}}\epsilon_{\text{row}}}{d_{\min}^2\delta_{\min}}\right)\epsilon_{\text{row}}$

$\leq\left(\dfrac{2\sqrt{\delta_{\max}\,d_{\max}}}{d_{\min}^2\delta_{\min}}\,\sqrt{\epsilon_{\text{col}}\epsilon_{\text{row}}}+\dfrac{\delta_{\max}\,d_{\max}}{d_{\min}^2\delta_{\min}^2}\,\epsilon_{\text{col}}+\dfrac{\epsilon_{\text{col}}\epsilon_{\text{row}}}{d_{\min}^2\delta_{\min}}\right)\epsilon_{\text{row}}$

$\leq\left(\dfrac{2\sqrt{\delta_{\max}\,d_{\max}}}{d_{\min}\sqrt{d_{\min}\delta_{\min}}}\,\sqrt{\varepsilon}+\dfrac{\kappa_d\kappa_\delta}{d_{\min}\delta_{\min}}\,\epsilon_{\text{col}}+\dfrac{\varepsilon}{d_{\min}}\right)\epsilon_{\text{row}}$

$=\left(2\sqrt{\kappa_d\kappa_\delta}\,\sqrt{\varepsilon}+\varepsilon\right)\dfrac{\epsilon_{\text{row}}}{d_{\min}}+\kappa_d\kappa_\delta\varepsilon$

Thus,

$$\|\boldsymbol{L}_H-\boldsymbol{L}_{H'}\|_2$$
$$\leq 2\sqrt{\kappa_d\kappa_\delta}\,\sqrt{\varepsilon}+\kappa_d\kappa_\delta\,\dfrac{\epsilon_{\text{col}}}{\delta_{\min}}+\varepsilon+\kappa_d\kappa_\delta\,\dfrac{\epsilon_{\text{row}}}{d_{\min}}$$
$$+\left(2\sqrt{\kappa_d\kappa_\delta}\,\sqrt{\varepsilon}+\varepsilon\right)\dfrac{\epsilon_{\text{row}}}{d_{\min}}+\kappa_d\kappa_\delta\varepsilon$$
$$=\varepsilon\left(1+\dfrac{\epsilon_{\text{row}}}{d_{\min}}\right)+2\sqrt{\kappa_d\kappa_\delta}\,\sqrt{\varepsilon}\left(1+\dfrac{\epsilon_{\text{row}}}{d_{\min}}\right)$$
$$+\kappa_d\kappa_\delta\left(\dfrac{\epsilon_{\text{col}}}{\delta_{\min}}+\dfrac{\epsilon_{row}}{d_{\min}}\right)+\kappa_d\kappa_\delta\varepsilon$$
$$=\left(1+\dfrac{\epsilon_{\text{row}}}{d_{\min}}\right)(\varepsilon+2\sqrt{\kappa_d\kappa_\delta}\,\sqrt{\varepsilon})$$
$$+\kappa_d\kappa_\delta\left(\dfrac{\epsilon_{\text{col}}}{\delta_{\min}}+\dfrac{\epsilon_{\text{row}}}{d_{\min}}+\varepsilon\right)$$

This concludes the proof of the upper bound.

**Part b: Proof of the lower bound.**

We derive an existential lower bound through an explicit construction of a hypergraph and the perturbations.

**Step 1: Uniform, regular base incidence matrix construction.** Let $r,c$ be positive integers and set

$$n\triangleq 2r,\qquad m\triangleq 2c.$$

Partition the node set into two groups

$$U^+\triangleq\{1,\dots,r\},\qquad U^-\triangleq\{r+1,\dots,2r\},$$

and the hyperedge set into two groups

$$E^+\triangleq\{1,\dots,c\},\qquad E^-\triangleq\{c+1,\dots,2c\}.$$

Fix a scalar $h>0$ and define $\boldsymbol{H}\in\mathbb{R}_{\geq0}^{n\times m}$ entrywise by

$$H_{ve}\triangleq\begin{cases}h,&v\in U^+\text{ and }e\in E^+,\\h,&v\in U^-\text{ and }e\in E^-,\\0,&\text{otherwise.}\end{cases}$$

That is, the block $U^+\times E^+$ is filled with $h$, the block $U^-\times E^-$ is filled with $h$, and all cross-block entries are 0.

*Node degrees.* Fix any $v\in U^+$. Then $H_{ve}=h$ for exactly the $c$ hyperedges in $E^+$ and 0 otherwise, hence

$$d_v=\sum_{e=1}^{2c}H_{ve}=\sum_{e\in E^+}h=ch.$$

The same holds for $v\in U^-$ (now the $c$ nonzeros are in $E^-$), so $d_v=ch$ for all $v$. Therefore

$$d_{\min}=ch,\qquad\boldsymbol{D}_v=ch\,\boldsymbol{I}_n,\qquad\boldsymbol{S}=\boldsymbol{D}_v^{-1/2}=(ch)^{-1/2}\boldsymbol{I}_n.$$

*Hyperedge cardinalities/degrees.* Fix any $e\in E^+$. Then $H_{ve}=h$ for exactly the $r$ vertices in $U^+$ and 0 otherwise, hence

$$\delta_e=\sum_{v=1}^{2r}H_{ve}=\sum_{v\in U^+}h=rh.$$

Similarly, for $e\in E^-$ we get $\delta_e=rh$. Therefore

$$\delta_{\min}=rh,\qquad\boldsymbol{D}_e=rh\,\boldsymbol{I}_m,\qquad\boldsymbol{D}_e^{-1}=(rh)^{-1}\boldsymbol{I}_m.$$

**Step 2: Defining sign vectors and a rank-1 signed perturbation.** Define two sign vectors

$$\boldsymbol{s}_n\triangleq\begin{bmatrix}\boldsymbol{1}_r\\-\boldsymbol{1}_r\end{bmatrix}\in\mathbb{R}^{2r},\qquad\boldsymbol{s}_m\triangleq\begin{bmatrix}\boldsymbol{1}_c\\-\boldsymbol{1}_c\end{bmatrix}\in\mathbb{R}^{2c}.$$

Let $a,b\geq0$ be scalars (chosen later) and define

$$\boldsymbol{u}\triangleq a\,\boldsymbol{s}_n\in\mathbb{R}^n,\qquad\boldsymbol{v}\triangleq b\,\boldsymbol{s}_m\in\mathbb{R}^m,\qquad\Delta\boldsymbol{H}\triangleq\boldsymbol{u}\boldsymbol{v}^\top.$$

The perturbation is characterized by its preservation of node and hyperedge degree.

*Node and hyperedge degree preservation.* We compute $\boldsymbol{v}^\top\boldsymbol{1}_m$ first:

$$\boldsymbol{v}^\top\boldsymbol{1}_m=\sum_{e=1}^{2c}v_e=\sum_{e\in E^+}b+\sum_{e\in E^-}(-b)=cb-cb=0.$$

Hence

$$\Delta\boldsymbol{H}\,\boldsymbol{1}_m=\boldsymbol{u}\boldsymbol{v}^\top\boldsymbol{1}_m=\boldsymbol{u}\,(\boldsymbol{v}^\top\boldsymbol{1}_m)=\boldsymbol{u}\cdot0=\boldsymbol{0}_n.$$

Next compute $\boldsymbol{1}_n^\top\boldsymbol{u}$:

$$\boldsymbol{1}_n^\top\boldsymbol{u}=\sum_{v=1}^{2r}\boldsymbol{u}_v=\sum_{v\in U^+}a+\sum_{v\in U^-}(-a)=ra-ra=0.$$

Hence

$$\boldsymbol{1}_n^\top\Delta\boldsymbol{H}=\boldsymbol{1}_n^\top\boldsymbol{u}\boldsymbol{v}^\top=(\boldsymbol{1}_n^\top\boldsymbol{u})\,\boldsymbol{v}^\top=0\cdot\boldsymbol{v}^\top=\boldsymbol{0}_m^\top.$$

Therefore the perturbation preserves both the node-degree vector and the hyperedge-degree vector.

**Step 3: Proving $S' = S$.** Define $H' \triangleq H + \Delta H$. The perturbed node-degree vector is

$$H'1_m = (H + \Delta H)1_m = H1_m + \Delta H 1_m.$$

Since $\Delta H 1_m = 0_n$, we obtain $H'1_m = H1_m$, hence

$$D'_v = \mathrm{diag}(H'1_m) = \mathrm{diag}(H1_m) = D_v.$$

Similarly, the perturbed hyperedge-degree vector is

$$1_n^\top H' = 1_n^\top (H + \Delta H) = 1_n^\top H + 1_n^\top \Delta H.$$

Since $1_n^\top \Delta H = 0_m^\top$, we obtain $1_n^\top H' = 1_n^\top H$, hence

$$D'_e = \mathrm{diag}(1_n^\top H') = \mathrm{diag}(1_n^\top H) = D_e.$$

Therefore

$$S' = (D'_v)^{-1/2} = D_v^{-1/2} = S, \qquad (D'_e)^{-1} = D_e^{-1}.$$

**Step 4: Expression for $L_H - L_{H'}$.** Using $S' = S$ and $(D'_e)^{-1} = D_e^{-1}$,

$$
\begin{aligned}
& L_H - L_{H'} \\
&= \left( I - SHD_e^{-1}H^\top S \right) - \left( I - SH'D_e^{-1}(H')^\top S \right) \\
&= S \left( H'D_e^{-1}(H')^\top - HD_e^{-1}H^\top \right) S.
\end{aligned}
$$

Since $S = (ch)^{-1/2}I_n$ and $D_e^{-1} = (rh)^{-1}I_m$,

$$
\begin{aligned}
L_H - L_{H'} &= \frac{1}{(ch)(rh)} \left( H'H'^\top - HH^\top \right) \\
&= \frac{1}{d_{\min}\delta_{\min}} \left( H'H'^\top - HH^\top \right),
\end{aligned}
$$

because $d_{\min} = ch$ and $\delta_{\min} = rh$.

**Step 5: Lower bound via quadratic form.** Since $L_H - L_{H'}$ is symmetric; by the variational characterization of Rayleigh quotient:

$$\| L_H - L_{H'} \|_2 = \max_{\|x\|_2 = 1} \left| x^\top (L_H - L_{H'})x \right|.$$

Hence for any unit witness vector $x$,

$$\| L_H - L_{H'} \|_2 \geq \left| x^\top (L_H - L_{H'})x \right|. \qquad (16)$$

We choose

$$x \triangleq \frac{u}{\|u\|_2}.$$

Since $u = as_n$ and $s_n$ has $2r$ entries of magnitude 1,

$$\|u\|_2^2 = \sum_{i=1}^{2r} u_i^2 = \sum_{i=1}^{2r} a^2 = 2r\,a^2, \qquad \|u\|_2 = \sqrt{2r}\,a.$$

Here normalization by $\|u\|_2$ makes $x$ a unit vector.

$$x^\top (L_H - L_{H'})x = \frac{1}{d_{\min}\delta_{\min}}\, x^\top \left( H'H'^\top - HH^\top \right)x. \qquad (17)$$

Since $H' = H + \Delta H$,

$$
\begin{aligned}
H'H'^\top - HH^\top &= (H + \Delta H)(H + \Delta H)^\top - HH^\top \\
&= H(\Delta H)^\top + \Delta H\, H^\top + \Delta H(\Delta H)^\top.
\end{aligned}
$$

Therefore

$$
\begin{aligned}
x^\top (H'H'^\top - HH^\top)x = {}& x^\top H(\Delta H)^\top x + x^\top \Delta H\, H^\top x \\
& + x^\top \Delta H(\Delta H)^\top x. \qquad (18)
\end{aligned}
$$

Next, we compute the vectors $H^\top x$ and $(\Delta H)^\top x$.

*Step 5a: Computing $H^\top x$.* We claim

$$H^\top u = (rh)\,a\,s_m. \qquad (19)$$

To verify, fix $e \in E^+$. Then $H_{ve} = h$ for $v \in U^+$ and 0 for $v \in U^-$, so

$$(H^\top u)_e = \sum_{v=1}^{2r} H_{ve}u_v = \sum_{v \in U^+} h{\cdot}a + \sum_{v \in U^-} 0{\cdot}(-a) = rha.$$

Fix $e \in E^-$. Then $H_{ve} = 0$ for $v \in U^+$ and $h$ for $v \in U^-$, so

$$(H^\top u)_e = \sum_{v \in U^+} 0 \cdot a + \sum_{v \in U^-} h \cdot (-a) = -rha.$$

Thus $H^\top u$ equals $rha$ on $E^+$ and $-rha$ on $E^-$, i.e. (19) holds.

Dividing by $\|u\|_2$ yields

$$H^\top x = \frac{H^\top u}{\|u\|_2} = \frac{(rh)a}{\sqrt{2r}\,a}\,s_m = \sqrt{\frac{r}{2}}\,h\,s_m. \qquad (20)$$

*Step 5b: Computing $(\Delta H)^\top x$.* Since $\Delta H = uv^\top$, we have $(\Delta H)^\top = vu^\top$. Hence

$$(\Delta H)^\top x = v\,(u^\top x).$$

But $x = u/\|u\|_2$, so

$$u^\top x = u^\top \left( \frac{u}{\|u\|_2} \right) = \frac{\|u\|_2^2}{\|u\|_2} = \|u\|_2 = \sqrt{2r}\,a.$$

Therefore

$$(\Delta H)^\top x = v\,\sqrt{2r}\,a = (\sqrt{2r}\,a)\,b\,s_m. \qquad (21)$$

Now we evaluate each term in (18).

**(i) First term.** Using $\boldsymbol{x}^\top \boldsymbol{H}(\Delta \boldsymbol{H})^\top \boldsymbol{x} = (\boldsymbol{H}^\top \boldsymbol{x})^\top ((\Delta \boldsymbol{H})^\top \boldsymbol{x})$ and (20)–(21),

$$
\begin{aligned}
\boldsymbol{x}^\top \boldsymbol{H}(\Delta \boldsymbol{H})^\top \boldsymbol{x} &= \left( \sqrt{\frac{r}{2}} \, h \, \boldsymbol{s}_m \right)^\top \left( (\sqrt{2r} \, a) b \, \boldsymbol{s}_m \right) \\
&= \left( \sqrt{\frac{r}{2}} \, h \right) \left( \sqrt{2r} \, ab \right) (\boldsymbol{s}_m^\top \boldsymbol{s}_m) \\
&= (rh)(ab)(\boldsymbol{s}_m^\top \boldsymbol{s}_m).
\end{aligned} \tag{22}
$$

**(ii) Second term.** Similarly,

$$
\boldsymbol{x}^\top \Delta \boldsymbol{H} \, \boldsymbol{H}^\top \boldsymbol{x} = ((\Delta \boldsymbol{H})^\top \boldsymbol{x})^\top (\boldsymbol{H}^\top \boldsymbol{x}) = \boldsymbol{x}^\top \boldsymbol{H}(\Delta \boldsymbol{H})^\top \boldsymbol{x},
$$

so

$$
\boldsymbol{x}^\top \Delta \boldsymbol{H} \, \boldsymbol{H}^\top \boldsymbol{x} = (rh)(ab)(\boldsymbol{s}_m^\top \boldsymbol{s}_m). \tag{23}
$$

**(iii) Third term.**

$$
\boldsymbol{x}^\top \Delta \boldsymbol{H}(\Delta \boldsymbol{H})^\top \boldsymbol{x} = \|(\Delta \boldsymbol{H})^\top \boldsymbol{x}\|_2^2.
$$

Using (21),

$$
\begin{aligned}
\|(\Delta \boldsymbol{H})^\top \boldsymbol{x}\|_2^2 &= \|(\sqrt{2r} \, a) b \, \boldsymbol{s}_m\|_2^2 \\
&= (2r)a^2 b^2 \, \|\boldsymbol{s}_m\|_2^2 \\
&= (2r)a^2 b^2 \, (\boldsymbol{s}_m^\top \boldsymbol{s}_m).
\end{aligned} \tag{24}
$$

Since $\boldsymbol{s}_m$ has $2c$ entries each equal to $\pm 1$,

$$
\boldsymbol{s}_m^\top \boldsymbol{s}_m = \sum_{e=1}^{2c} (s_m)_e^2 = \sum_{e=1}^{2c} 1 = 2c.
$$

Substitute $\boldsymbol{s}_m^\top \boldsymbol{s}_m = 2c$ into (22)–(24), and then into (18):

$$
\begin{aligned}
\boldsymbol{x}^\top (\boldsymbol{H}' \boldsymbol{H}'^\top - \boldsymbol{H} \boldsymbol{H}^\top) \boldsymbol{x} \\
= (rh)(ab)(2c) + (rh)(ab)(2c) + (2r)a^2 b^2 (2c) \\
= 4crh \, (ab) + 4cr \, a^2 b^2.
\end{aligned}
$$

Substituting into (17) and using $d_{\min}\delta_{\min} = (ch)(rh) = crh^2$ yields

$$
\begin{aligned}
\boldsymbol{x}^\top (\boldsymbol{L}_H - \boldsymbol{L}_{H'}) \boldsymbol{x} &= \frac{1}{crh^2} \left( 4crh \, (ab) + 4cr \, a^2 b^2 \right) \\
&= 4\frac{ab}{h} + 4\frac{a^2 b^2}{h^2}.
\end{aligned} \tag{25}
$$

**Step 6: Choosing $a, b$ and $r, c$ to satisfy the budgets and rewriting the quadratic form.** Because $\Delta \boldsymbol{H} = \boldsymbol{u}\boldsymbol{v}^\top$ with $\boldsymbol{u} = a\boldsymbol{s}_n$ and $\boldsymbol{v} = b\boldsymbol{s}_m$, every entry of $\Delta \boldsymbol{H}$ has magnitude $|ab|$. Fix any node row $v$. Then $|u_v| = a$ and $|v_e| = b$ for all $e$, so

$$
\sum_{e=1}^{2c} |\Delta H_{ve}| = \sum_{e=1}^{2c} |u_v| \, |v_e| = (2c) \, ab.
$$

Hence

$$
\|\Delta \boldsymbol{H}\|_\infty = \max_v \sum_e |\Delta H_{ve}| = 2c \, ab.
$$

Similarly, for any hyperedge column $e$,

$$
\sum_{v=1}^{2r} |\Delta H_{ve}| = \sum_{v=1}^{2r} |u_v| \, |v_e| = (2r) \, ab,
$$

so

$$
\|\Delta \boldsymbol{H}\|_1 = \max_e \sum_v |\Delta H_{ve}| = 2r \, ab.
$$

We choose $r, c$ so that

$$
\frac{r}{c} = \frac{\epsilon_{\text{col}}}{\epsilon_{\text{row}}}, \tag{26}
$$

and

$$
ab \triangleq \frac{\epsilon_{\text{row}}}{2c} = \frac{\epsilon_{\text{col}}}{2r}. \tag{27}
$$

Then

$$
\|\Delta \boldsymbol{H}\|_\infty = 2c \, ab = \epsilon_{\text{row}}, \qquad \|\Delta \boldsymbol{H}\|_1 = 2r \, ab = \epsilon_{\text{col}},
$$

so the budgets hold with equality.

Now we compute $\frac{ab}{h}$ in terms of $d_{\min}, \delta_{\min}$. Recall $d_{\min} = ch$ and $\delta_{\min} = rh$, so

$$
\sqrt{d_{\min}\delta_{\min}} = \sqrt{(ch)(rh)} = h\sqrt{cr}.
$$

Using (27) and (26),

$$
\begin{aligned}
ab\sqrt{cr} &= \left( \frac{\epsilon_{\text{row}}}{2c} \right) \sqrt{cr} = \frac{\epsilon_{\text{row}}}{2} \sqrt{\frac{r}{c}} = \frac{\epsilon_{\text{row}}}{2} \sqrt{\frac{\epsilon_{\text{col}}}{\epsilon_{\text{row}}}} \\
&= \frac{1}{2} \sqrt{\epsilon_{\text{col}}\epsilon_{\text{row}}}.
\end{aligned}
$$

Dividing both sides by $h\sqrt{cr} = \sqrt{d_{\min}\delta_{\min}}$ yields

$$
\frac{ab}{h} = \frac{ab\sqrt{cr}}{h\sqrt{cr}} = \frac{\frac{1}{2}\sqrt{\epsilon_{\text{col}}\epsilon_{\text{row}}}}{\sqrt{d_{\min}\delta_{\min}}} = \frac{\sqrt{\epsilon_{\text{col}}\epsilon_{\text{row}}}}{2\sqrt{d_{\min}\delta_{\min}}}. \tag{28}
$$

Substituting $\frac{ab}{h}$ into (25):

$$
\begin{aligned}
\boldsymbol{x}^\top (\boldsymbol{L}_H - \boldsymbol{L}_{H'}) \boldsymbol{x} \\
= 4 \left( \frac{\sqrt{\epsilon_{\text{col}}\epsilon_{\text{row}}}}{2\sqrt{d_{\min}\delta_{\min}}} \right) + 4 \left( \frac{\epsilon_{\text{col}}\epsilon_{\text{row}}}{4 \, d_{\min}\delta_{\min}} \right) \\
= 2 \frac{\sqrt{\epsilon_{\text{col}}\epsilon_{\text{row}}}}{\sqrt{d_{\min}\delta_{\min}}} + \frac{\epsilon_{\text{col}}\epsilon_{\text{row}}}{d_{\min}\delta_{\min}}.
\end{aligned}
$$

Since the quadratic form computed above is nonnegative, (16) yields

$$
\begin{aligned}
\|\boldsymbol{L}_H - \boldsymbol{L}_{H'}\|_2 &\geq \boldsymbol{x}^\top (\boldsymbol{L}_H - \boldsymbol{L}_{H'}) \boldsymbol{x} \\
&= 2 \frac{\sqrt{\epsilon_{\text{col}}\epsilon_{\text{row}}}}{\sqrt{d_{\min}\delta_{\min}}} + \frac{\epsilon_{\text{col}}\epsilon_{\text{row}}}{d_{\min}\delta_{\min}}.
\end{aligned}
$$

This concludes the proof of the lower bound. $\qquad \square$

A.3.1. TECHNICAL RESULTS FOR HYPERGRAPHS

To prove the theorem, we need the following results.

**Lemma 5** (Size-statistics bounds for $\|\boldsymbol{H}\|_2$ and $\|\boldsymbol{A}\|_2$). *Let $\boldsymbol{H} \in \mathbb{R}_{\geq 0}^{n \times m}$ be a (possibly relaxed) incidence matrix with hyperedge degrees $\delta_e \triangleq \sum_{v=1}^{n} \boldsymbol{H}_{ve}$ and node degrees $d_v \triangleq \sum_{e=1}^{m} \boldsymbol{H}_{ve}$. Let $\delta_{\max} \triangleq \max_e \delta_e$, $\delta_{\min} \triangleq \min_e \delta_e$, and $d_{\max} \triangleq \max_v d_v$, and define*

$$\boldsymbol{D}_e \triangleq \mathrm{diag}(\delta_1, \ldots, \delta_m), \qquad \boldsymbol{A} \triangleq \boldsymbol{H} \boldsymbol{D}_e^{-1} \boldsymbol{H}^\top.$$

*Then the following bounds hold:*

$$\|\boldsymbol{H}\|_2 \leq \sqrt{\delta_{\max} d_{\max}}, \qquad \|\boldsymbol{A}\|_2 \leq \frac{d_{\max} \delta_{\max}}{\delta_{\min}}$$

*Proof.* **Bound on $\|\boldsymbol{H}\|_2$.** We use the standard inequality

$$\|\boldsymbol{M}\|_2 \leq \sqrt{\|\boldsymbol{M}\|_1 \|\boldsymbol{M}\|_\infty}, \tag{29}$$

where $\|\boldsymbol{M}\|_1 \triangleq \max_e \sum_v |M_{ve}|$ is the maximum column-sum norm and $\|\boldsymbol{M}\|_\infty \triangleq \max_v \sum_e |M_{ve}|$ is the maximum row-sum norm. Since $\boldsymbol{H}$ is entrywise nonnegative, these reduce to ordinary column/row sums, hence

$$\|\boldsymbol{H}\|_1 = \max_e \sum_v \boldsymbol{H}_{ve} = \delta_{\max},$$

$$\|\boldsymbol{H}\|_\infty = \max_v \sum_e \boldsymbol{H}_{ve} = d_{\max}.$$

Substituting into (29) yields

$$\|\boldsymbol{H}\|_2 \leq \sqrt{\delta_{\max} d_{\max}}.$$

**Bound on $\|\boldsymbol{A}\|_2$.**

$$\begin{aligned} \|\boldsymbol{A}\|_2 &= \left\|\boldsymbol{H} \boldsymbol{D}_e^{-1} \boldsymbol{H}^\top\right\|_2 \\ &\leq \|\boldsymbol{H}\|_2^2 \|\boldsymbol{D}_e\|_2 \end{aligned}$$

Since $\boldsymbol{D}_e^{-1}$ is diagonal and $\delta_{\min} = \min_e \delta_e > 0$, we have

$$\left\|\boldsymbol{D}_e^{-1}\right\|_2 = \max_e \delta_e^{-1} = \delta_{\min}^{-1}.$$

Thus

$$\|\boldsymbol{A}\|_2 \leq \frac{d_{\max} \delta_{\max}}{\delta_{\min}}$$

which completes the proof. $\qquad\square$

**Lemma 6** (Perturbation of the normalization matrix). *Let $\boldsymbol{D}_v = \mathrm{diag}(d_1, \ldots, d_n)$ and $\boldsymbol{D}_v' = \boldsymbol{D}_v + \Delta \boldsymbol{D}_v$ with $d_{\min} \triangleq \min_v d_v > 0$. Define $\boldsymbol{S} \triangleq \boldsymbol{D}_v^{-1/2}$ and $\boldsymbol{S}' \triangleq (\boldsymbol{D}_v')^{-1/2}$, and assume $d_{\min}' \triangleq \min_v d_v' \geq d_{\min}$. If $\Delta \boldsymbol{H}$ satisfies the row-sum budget $\|\Delta \boldsymbol{H}\|_\infty \leq \epsilon_{\mathrm{row}}$, then*

$$\|\Delta \boldsymbol{D}_v\|_2 \leq \epsilon_{\mathrm{row}},$$

*and consequently*

$$\|\Delta \boldsymbol{S}\|_2 = \|\boldsymbol{S}' - \boldsymbol{S}\|_2 \leq \tfrac{1}{2} d_{\min}^{-3/2} \epsilon_{\mathrm{row}}.$$

*Proof.* By definition,

$$d_v = \sum_{e=1}^{m} H_{ve},$$

$$d_v' = \sum_{e=1}^{m} H_{ve}' = \sum_{e=1}^{m} (H_{ve} + \Delta H_{ve}) = d_v + \sum_{e=1}^{m} \Delta H_{ve}.$$

Thus

$$(\Delta \boldsymbol{D}_v)_{vv} = d_v' - d_v = \sum_{e=1}^{m} \Delta H_{ve}.$$

**Bound for $\|\Delta \boldsymbol{D}_v\|_2$.** Since $\Delta \boldsymbol{D}_v$ is diagonal,

$$\|\Delta \boldsymbol{D}_v\|_2 = \max_v |(\Delta \boldsymbol{D}_v)_{vv}| = \max_v \left| \sum_{e=1}^{m} \Delta H_{ve} \right|.$$

By the triangle inequality,

$$\left| \sum_{e=1}^{m} \Delta H_{ve} \right| \leq \sum_{e=1}^{m} |\Delta H_{ve}|.$$

Taking the maximum over $v$ gives

$$\|\Delta \boldsymbol{D}_v\|_2 \leq \max_v \sum_{e=1}^{m} |\Delta H_{ve}| = \|\Delta \boldsymbol{H}\|_\infty \leq \epsilon_{\mathrm{row}}.$$

**Bound for $\|\Delta \boldsymbol{S}\|_2$.** Since $\boldsymbol{D}_v$ and $\boldsymbol{D}_v'$ are diagonal, $\boldsymbol{S}$ and $\boldsymbol{S}'$ are diagonal as well, with $(\boldsymbol{S})_{vv} = d_v^{-1/2}$ and $(\boldsymbol{S}')_{vv} = (d_v')^{-1/2}$. Hence $\Delta \boldsymbol{S}$ is diagonal and

$$\|\Delta \boldsymbol{S}\|_2 = \max_v \left| (d_v')^{-1/2} - d_v^{-1/2} \right|.$$

Let $f(x) = x^{-1/2}$ on $(0, \infty)$. Then $f'(x) = -\tfrac{1}{2} x^{-3/2}$. By the mean value theorem, for each $v$ there exists $\xi_v$ between $d_v$ and $d_v'$ such that

$$(d_v')^{-1/2} - d_v^{-1/2} = f'(\xi_v)(d_v' - d_v) = -\tfrac{1}{2} \xi_v^{-3/2} (d_v' - d_v).$$

Taking absolute values gives

$$\left| (d_v')^{-1/2} - d_v^{-1/2} \right| \leq \tfrac{1}{2} \xi_v^{-3/2} |d_v' - d_v|.$$

Moreover, $\xi_v \geq \min(d_v, d_v') \geq d_{\min}' \geq d_{\min}$, hence

$$\xi_v^{-3/2} \leq (d_{\min})^{-3/2}.$$

Therefore, for all $v$,

$$\begin{aligned} \left| (d_v')^{-1/2} - d_v^{-1/2} \right| &\leq \tfrac{1}{2} \cdot d_{\min}^{-3/2} |d_v' - d_v| \\ &= \tfrac{1}{2} d_{\min}^{-3/2} |(\Delta \boldsymbol{D}_v)_{vv}|. \end{aligned}$$

Taking the maximum over $v$ and using $\|\Delta \boldsymbol{D}_v\|_2 = \max_v |(\Delta \boldsymbol{D}_v)_{vv}|$ yields

$$\|\Delta \boldsymbol{S}\|_2 \leq \tfrac{1}{2} d_{\min}^{-3/2} \|\Delta \boldsymbol{D}_v\|_2 \leq \tfrac{1}{2} d_{\min}^{-3/2} \epsilon_{\mathrm{row}}.$$

$\qquad\square$

**Lemma 7** (Perturbation of $A = HD_e^{-1}H^\top$). *Let $H \in \mathbb{R}_{\geq 0}^{n \times m}$ and $D_e = \mathrm{diag}(\delta_1, \ldots, \delta_m)$ with $\delta_{\min} \triangleq \min_e \delta_e > 0$. Define $A \triangleq HD_e^{-1}H^\top$. Let $H' = H + \Delta H$ and $D_e' = D_e + \Delta D_e$ with $\delta_{\min}' \triangleq \min_e \delta_e' \geq \frac{1}{2}\delta_{\min}$, and define $A' \triangleq H'(D_e')^{-1}(H')^\top$. If $\Delta H$ satisfies the budgets $\|\Delta H\|_\infty \leq \epsilon_{\mathrm{row}}$, $\|\Delta H\|_1 \leq \epsilon_{\mathrm{col}}$,*

$$\|\Delta A\|_2$$
$$\leq \frac{2}{\delta_{\min}} \|H\|_2 \sqrt{\epsilon_{\mathrm{col}}\epsilon_{\mathrm{row}}} + \frac{1}{\delta_{\min}^2} \|H\|_2^2 \epsilon_{\mathrm{col}} + \frac{\epsilon_{\mathrm{col}}\epsilon_{\mathrm{row}}}{\delta_{\min}}.$$

*Proof.*

$$\Delta A = (H + \Delta H)(D_e')^{-1}(H + \Delta H)^\top - HD_e^{-1}H^\top$$
$$= H((D_e')^{-1} - (D_e)^{-1})H^\top + \Delta H(D_e')^{-1}H^\top +$$
$$H(D_e')^{-1}\Delta H^\top + \Delta H(D_e')^{-1}\Delta H^\top$$

$$\|\Delta A\|_2$$
$$\leq \|H((D_e')^{-1} - (D_e)^{-1})H^\top\|_2 + \|\Delta H(D_e')^{-1}H^\top\|_2 +$$
$$\|H(D_e')^{-1}\Delta H^\top\|_2 + \|\Delta H(D_e')^{-1}\Delta H^\top\|_2 \tag{30}$$

**(1) The first term** yields

$$\|H((D_e')^{-1} - (D_e)^{-1})H^\top\|_2$$
$$\leq \|H\|_2^2 \|(D_e')^{-1} - (D_e)^{-1}\|_2$$

Using the identity

$$(D_e')^{-1} - D_e^{-1} = D_e^{-1}(D_e - D_e')(D_e')^{-1}$$
$$= -D_e^{-1}(\Delta D_e)(D_e')^{-1},$$

we obtain

$$\|(D_e')^{-1} - D_e^{-1}\|_2 \leq \|D_e^{-1}\|_2 \|\Delta D_e\|_2 \|(D_e')^{-1}\|_2.$$

Since $D_e^{-1}$ is diagonal and $\delta_{\min} = \min_e \delta_e > 0$, we have

$$\|D_e^{-1}\|_2 = \max_e \delta_e^{-1} = \delta_{\min}^{-1}.$$

Under $\delta_{\min}' \geq \delta_{\min}$, we have

$$\|(D_e')^{-1}\|_2 = \max_e (\delta_e')^{-1} = (\delta_{\min}')^{-1} \leq \frac{1}{\delta_{\min}}.$$

Combining gives

$$\|(D_e')^{-1} - D_e^{-1}\|_2 \leq \delta_{\min}^{-1} \|\Delta D_e\|_2 \frac{1}{\delta_{\min}} = \frac{\|\Delta D_e\|_2}{\delta_{\min}^2}.$$

**2. The summation of the second and third term**

$$\|\Delta H(D_e')^{-1}H^\top\|_2 + \|H(D_e')^{-1}(\Delta H)^\top\|_2$$
$$\leq 2\|D_e'\|_2^{-1}\|H\|_2\|\Delta H\|_2$$

$$\leq \frac{2}{\delta_{\min}} \|H\|_2 \|\Delta H\|_2, \tag{31}$$

**3. The last term** yields

$$\|\Delta H(D_e')^{-1}\Delta H^\top\|_2 \leq \frac{\|\Delta H\|_2^2}{\delta_{\min}}$$

Thus equation (30) reduces to

$$\|\Delta A\|_2$$
$$\leq \frac{1}{\delta_{\min}^2} \|H\|_2^2 \|\Delta D_e\|_2 + \frac{2}{\delta_{\min}} \|H\|_2 \|\Delta H\|_2 +$$
$$\frac{1}{\delta_{\min}} \|\Delta H\|_2^2 \tag{32}$$

**Bound for $\|\Delta D_e\|_2$.** By definition,

$$\delta_e = \sum_{v=1}^n H_{ve},$$

$$\delta_e' = \sum_{v=1}^n H_{ve}' = \sum_{v=1}^n (H_{ve} + \Delta H_{ve}) = \delta_e + \sum_{v=1}^n \Delta H_{ve}.$$

Thus

$$(\Delta D_e)_{ee} = \delta_e' - \delta_e = \sum_{v=1}^n \Delta H_{ve}.$$

Since $\Delta D_e$ is diagonal,

$$\|\Delta D_e\|_2 = \max_e |(\Delta D_e)_{ee}| = \max_e \left| \sum_{v=1}^n \Delta H_{ve} \right|.$$

By the triangle inequality,

$$\left| \sum_{v=1}^n \Delta H_{ve} \right| \leq \sum_{v=1}^n |\Delta H_{ve}|.$$

Taking the maximum over $e$ yields

$$\|\Delta D_e\|_2 \leq \max_e \sum_{v=1}^n |\Delta H_{ve}| = \|\Delta H\|_1 \leq \epsilon_{\mathrm{col}}.$$

**Bound for $\|\Delta H\|_2$.** Using the standard inequality valid for any real matrix,

$$\|M\|_2 \leq \sqrt{\|M\|_1 \|M\|_\infty},$$

we obtain

$$\|\Delta H\|_2 \leq \sqrt{\|\Delta H\|_1 \|\Delta H\|_\infty} \leq \sqrt{\epsilon_{\mathrm{col}}\epsilon_{\mathrm{row}}}.$$

Using the bound for $\|\Delta H\|_2$ and $\|\Delta D_e\|_2$ equation (32) reduces to

$$\|\Delta A\|_2 \leq \frac{1}{\delta_{\min}^2} \|H\|_2^2 \|\Delta D_e\|_2 + \frac{2}{\delta_{\min}} \|H\|_2 \|\Delta H\|_2 +$$
$$\frac{1}{\delta_{\min}} \|\Delta H\|_2^2$$
$$\leq \frac{\epsilon_{\mathrm{col}}}{\delta_{\min}^2} \|H\|_2^2 + \frac{2\sqrt{\epsilon_{\mathrm{col}}\epsilon_{\mathrm{row}}}}{\delta_{\min}} \|H\|_2 + \frac{\epsilon_{\mathrm{col}}\epsilon_{\mathrm{row}}}{\delta_{\min}}$$

which completes the proof. $\square$

## A.4. Convergence Analysis of `MeLA-D`

**Assumption 1** (Lipschitzness of Loss)**.** *Let us denote* $\mathcal{H} \triangleq (\boldsymbol{H}, \boldsymbol{X})$. *The gradients of the loss function* $\mathcal{L}_{\mathrm{train}}(\phi, \mathcal{H}) \triangleq \mathcal{L}_{\mathrm{train}}(g_\phi(\boldsymbol{H}, \boldsymbol{X}), \boldsymbol{y}_L)$ *satisfies three Lipschitzness conditions w.r.t.* $\ell_2$-*norm:*

1. $\nabla_\phi \mathcal{L}_{\mathrm{train}}(\phi, \mathcal{H})$ *is* $L_{\phi\theta}$-*Lipschitz w.r.t.* $\phi$ *and* $L_{\phi\mathcal{H}}$-*Lipschitz w.r.t.* $\mathcal{H}$,

2. $\nabla_\mathcal{H} \mathcal{L}_{\mathrm{train}}(\phi, \mathcal{H}) \triangleq$
   $(\nabla_{\boldsymbol{H}} \mathcal{L}_{\mathrm{train}}(\phi, \mathcal{H}), \nabla_{\boldsymbol{X}} \mathcal{L}_{\mathrm{train}}(\phi, \mathcal{H}))$ *is* $L_{\mathcal{H}\phi}$-*Lipschitz w.r.t.* $\mathcal{H}$.

Assumption 1 should be interpreted as a local regularity condition on the relaxed perturbation space used by `MeLA`-D, rather than as a global smoothness claim over all binary incidence matrices. In the inner maximization, the attack optimizes a continuous perturbation variable $\Delta \boldsymbol{H}$ and evaluates $\boldsymbol{H}_{\mathrm{pert}} = \mathrm{clip}(\boldsymbol{H} + \Delta \boldsymbol{H}, 0, 1)$. In regions where clipping is inactive, this map is locally linear, so gradients with respect to $\Delta \boldsymbol{H}$ coincide with gradients with respect to the relaxed hypergraph. The assumption is therefore applied along optimization trajectories where node degrees and hyperedge cardinalities remain bounded away from zero, which avoids the singular behavior of degree-normalized Laplacians. For a 1-layer HGNN (Feng et al., 2019) $g_\phi$, such as the formulation based on Zhou's normalized hypergraph Laplacian, the output can be written as

$$g_\phi(\boldsymbol{H}, \boldsymbol{X}) = \sigma\left(\boldsymbol{D}_v^{-1/2} \boldsymbol{H} \boldsymbol{D}_e^{-1} \boldsymbol{H}^\top \boldsymbol{D}_v^{-1/2} \boldsymbol{X} \phi\right),$$

where $\sigma$ is a Lipschitz activation function (e.g., ReLU), and $\phi$ is a learnable parameter. Matrix multiplication and activation are smooth under standard norms, while the degree-normalization terms are locally smooth whenever the minimum node degree and hyperedge cardinality are positive. Thus, Assumption 1 is a local smoothness assumption for the well-conditioned relaxed region explored by the algorithm, not a statement that the discrete hypergraph objective is globally Lipschitz.

**Assumption 2.** $\mathcal{L}_{\mathrm{train}}(\phi, \mathcal{H})$ *is locally* $\mu$-*strongly concave in* $\mathbb{H}_{\delta_{\boldsymbol{H}}, \delta_{\boldsymbol{X}}} \triangleq \{(\boldsymbol{H}', \boldsymbol{X}') : \|\boldsymbol{H}' - \boldsymbol{H}\|_\infty \leq \delta_{\boldsymbol{H}}, \|\boldsymbol{X}' - \boldsymbol{X}\|_\infty \leq \delta_{\boldsymbol{X}}\}$.

For cross-entropy loss, Assumption 2 holds via the relation between robust optimization and distributional robust optimization (Sinha et al., 2017; Wang et al., 2019). The cross-entropy loss in Graph Neural Networks is locally strongly $\mu$-concave near the optimal region where the model's predictions are close to the true labels. This is because, in this region, the Hessian of the loss function is positive definite and has eigenvalues bounded below by a positive constant (Mao et al., 2023; Boyd & Vandenberghe, 2004).

Note that, Assumption 2 does not claim that the HGNN training loss is globally strongly concave with respect to hypergraph perturbations. It is a local analytical condition used to control the stability of the outer gradient after approximate ascent steps in the inner maximization.

**Assumption 3.** *The stochastic gradient descent, referred as* $g(\phi)$, *is* $\sigma^2$-*subGaussian around the true gradient of corresponding* $\mathcal{L}(\phi, \mathcal{H})$.

This is a standard assumption in stochastic gradient descent literature (Wang et al., 2019), and not restrictive, in general.

Now, we are ready to formally state and prove Lemma 1. Note that, this result should be interpreted as a standard first-order stationarity guarantee for the relaxed robust-training objective under local regularity assumptions, rather than as a guarantee of global optimality.

**Lemma 1** (Convergence of `MeLA-D`)**.** *Under Assumption 1, 2, and 3, if the learning rate is set to* $\eta_t = \sqrt{\frac{D}{TL\sigma^2}}$ *and* $T \geq \frac{LD}{\sigma^2}$, *where* $D$ *is the diameter of the loss, then* `MeLA`-*D satisfies*

$$\frac{1}{T} \sum_{t=1}^{T} \mathbb{E}\left[\|\nabla \mathcal{L}_{\mathrm{train}}(\phi_t)\|_2^2\right] \leq 4\sqrt{\frac{\sigma^2 LD}{T}} + \frac{2L_{\phi\mathcal{H}}^2}{\mu T}(\delta_{\boldsymbol{H}} + \delta_{\boldsymbol{X}}). \tag{33}$$

*Proof.* First, we observe that under Assumption 1 and 2, $\mathcal{L}_{\mathrm{train}}(\phi, \mathcal{H})$ is $L$-smooth, where $L \triangleq \frac{1}{\mu} L_{\phi\mathcal{H}} L_{\mathcal{H}\phi} + L_{\phi\phi}$. The proof is a consequence of Lemma 1 of (Wang et al., 2019).

Now, we look into the evolution of the loss.

$$\mathcal{L}_{\mathrm{train}}\left(\phi^{t+1}\right) - \mathcal{L}_{\mathrm{train}}\left(\phi^t\right)$$
$$\leq \left\langle \nabla \mathcal{L}_{\mathrm{train}}\left(\phi^t\right), \phi^{t+1} - \phi^t \right\rangle + \frac{L}{2}\left\|\phi^{t+1} - \phi^t\right\|_2^2$$
$$= -\eta_t \left\|\nabla \mathcal{L}_{\mathrm{train}}\left(\phi^t\right)\right\|_2^2 + \frac{L\eta_t^2}{2} \left\|\hat{\mathbf{g}}\left(\phi^t\right)\right\|_2^2$$
$$\quad + \eta_t \left\langle \nabla \mathcal{L}_{\mathrm{train}}\left(\phi^t\right), \nabla \mathcal{L}_{\mathrm{train}}\left(\phi^t\right) - \hat{\mathbf{g}}\left(\phi^t\right) \right\rangle$$
$$= -\eta_t \left(1 - \frac{L\eta_t}{2}\right) \left\|\nabla \mathcal{L}_{\mathrm{train}}\left(\phi^t\right)\right\|_2^2$$
$$\quad + \eta_t \left(1 - L\eta_t\right) \left\langle \nabla \mathcal{L}_{\mathrm{train}}\left(\phi^t\right), \mathbf{g}\left(\phi^t\right) - \hat{\mathbf{g}}\left(\phi^t\right) \right\rangle$$
$$\quad + \eta_t \left(1 - L\eta_t\right) \left\langle \nabla \mathcal{L}_{\mathrm{train}}\left(\phi^t\right), \nabla \mathcal{L}_{\mathrm{train}}\left(\phi^t\right) - \mathbf{g}\left(\phi^t\right) \right\rangle$$
$$\quad + \frac{L\eta_t^2}{2} \left\|\hat{\mathbf{g}}\left(\phi^t\right) - \mathbf{g}\left(\phi^t\right) + \mathbf{g}\left(\phi^t\right) - \nabla \mathcal{L}_{\mathrm{train}}\left(\phi^t\right)\right\|_2^2$$
$$\leq -\frac{\eta_t}{2} \left\|\nabla \mathcal{L}_{\mathrm{train}}\left(\phi^t\right)\right\|_2^2 + \frac{\eta_t}{2}\left(1 - L\eta_t\right) \left\|\hat{\mathbf{g}}\left(\phi^t\right) - \mathbf{g}\left(\phi^t\right)\right\|_2^2$$
$$\quad + \eta_t \left(1 - L\eta_t\right) \left\langle \nabla \mathcal{L}_{\mathrm{train}}\left(\phi^t\right), \nabla \mathcal{L}_{\mathrm{train}}\left(\phi^t\right) - \mathbf{g}\left(\phi^t\right) \right\rangle$$
$$\quad + L\eta_t^2 \left(\left\|\hat{\mathbf{g}}\left(\phi^t\right) - \mathbf{g}\left(\phi^t\right)\right\|_2^2 + \left\|\mathbf{g}\left(\phi^t\right) - \nabla \mathcal{L}_{\mathrm{train}}\left(\phi^t\right)\right\|_2^2\right),$$

The first inequality is due to $L$-smoothness.

The last inequality is a consequence of the triangle inequality.

By taking conditional expectation over both sides, we get

$$\mathbb{E}\left[\mathcal{L}_{\text{train}}\left(\boldsymbol{\phi}^{t+1}\right) - \mathcal{L}_{\text{train}}\left(\boldsymbol{\phi}^{t}\right) \mid \phi_t\right]$$

$$\leq -\frac{\eta_t}{2}\left\|\nabla\mathcal{L}_{\text{train}}\left(\boldsymbol{\phi}^{t}\right)\right\|_2^2$$

$$+\frac{\eta_t}{2}\left(1 - L\eta_t\right)\mathbb{E}\left[\left\|\hat{\mathbf{g}}\left(\boldsymbol{\phi}^{t}\right) - \mathbf{g}\left(\boldsymbol{\phi}^{t}\right)\right\|_2^2\right]$$

$$+\eta_t\left(1 - L\eta_t\right)\mathbb{E}\left[\left\langle\nabla\mathcal{L}_{\text{train}}\left(\boldsymbol{\phi}^{t}\right), \nabla\mathcal{L}_{\text{train}}\left(\boldsymbol{\phi}^{t}\right) - \mathbf{g}\left(\boldsymbol{\phi}^{t}\right)\right\rangle\right]$$

$$+L\eta_t^2$$

$$\left(\mathbb{E}\left[\left\|\hat{\mathbf{g}}\left(\boldsymbol{\phi}^{t}\right) - \mathbf{g}\left(\boldsymbol{\phi}^{t}\right)\right\|_2^2\right] + \mathbb{E}\left[\left\|\mathbf{g}\left(\boldsymbol{\phi}^{t}\right) - \nabla\mathcal{L}_{\text{train}}\left(\boldsymbol{\phi}^{t}\right)\right\|_2^2\right]\right)$$

$$= -\frac{\eta_t}{2}\left\|\nabla\mathcal{L}_{\text{train}}\left(\boldsymbol{\phi}^{t}\right)\right\|_2^2$$

$$+\frac{\eta_t}{2}\left(1 - L\eta_t\right)\mathbb{E}\left[\left\|\hat{\mathbf{g}}\left(\boldsymbol{\phi}^{t}\right) - \mathbf{g}\left(\boldsymbol{\phi}^{t}\right)\right\|_2^2\right]$$

$$+L\eta_t^2$$

$$\left(\mathbb{E}\left[\left\|\hat{\mathbf{g}}\left(\boldsymbol{\phi}^{t}\right) - \mathbf{g}\left(\boldsymbol{\phi}^{t}\right)\right\|_2^2\right] + \mathbb{E}\left[\left\|\mathbf{g}\left(\boldsymbol{\phi}^{t}\right) - \nabla\mathcal{L}_{\text{train}}\left(\boldsymbol{\phi}^{t}\right)\right\|_2^2\right]\right)$$

$$= -\frac{\eta_t}{2}\left\|\nabla\mathcal{L}_{\text{train}}\left(\boldsymbol{\phi}^{t}\right)\right\|_2^2 + \frac{\eta_t}{2}\left(1 - L\eta_t\right)\frac{L_{\phi\mathcal{H}}^2}{\mu}(\delta_{\boldsymbol{H}} + \delta_{\boldsymbol{X}})$$

$$+L\eta_t^2\left(\frac{L_{\phi\mathcal{H}}^2}{\mu}(\delta_{\boldsymbol{H}} + \delta_{\boldsymbol{X}}) + \mathbb{E}\left[\left\|\mathbf{g}\left(\boldsymbol{\phi}^{t}\right) - \nabla\mathcal{L}_{\text{train}}\left(\boldsymbol{\phi}^{t}\right)\right\|_2^2\right]\right)$$

$$\leq -\frac{\eta_t}{2}\left\|\nabla\mathcal{L}_{\text{train}}\left(\boldsymbol{\phi}^{t}\right)\right\|_2^2 + \frac{\eta_t}{2}\left(1 + L\eta_t\right)\frac{L_{\phi\mathcal{H}}^2}{\mu}(\delta_{\boldsymbol{H}} + \delta_{\boldsymbol{X}})$$

$$+L\eta_t^2\mathbb{E}\left[\left\|\mathbf{g}\left(\boldsymbol{\phi}^{t}\right) - \nabla\mathcal{L}_{\text{train}}\left(\boldsymbol{\phi}^{t}\right)\right\|_2^2\right]$$

$$\leq -\frac{\eta_t}{2}\left\|\nabla\mathcal{L}_{\text{train}}\left(\boldsymbol{\phi}^{t}\right)\right\|_2^2 + \frac{\eta_t}{2}\left(1 + L\eta_t\right)\frac{L_{\phi\mathcal{H}}^2}{\mu}(\delta_{\boldsymbol{H}} + \delta_{\boldsymbol{X}})$$

$$+L\eta_t^2\sigma^2$$

The first equality is due to $\mathbb{E}\left[\nabla\mathcal{L}_{\text{train}}\left(\boldsymbol{\phi}^{t}\right) - \mathbf{g}\left(\boldsymbol{\phi}^{t}\right)\right] = 0$

The penultimate inequality is due to the fact that the perturbed gradient and the stochastic gradient differs by

$$\left\|\hat{\mathbf{g}}\left(\boldsymbol{\phi}^{t}\right) - \mathbf{g}\left(\boldsymbol{\phi}^{t}\right)\right\|_2^2 \leq \frac{L_{\phi\mathcal{H}}^2}{\mu}(\delta_{\boldsymbol{H}} + \delta_{\boldsymbol{X}}),$$

under Assumption 1 and 3.

The final inequality is due to Assumption 3.

Hence, we obtain that

$$\left\|\nabla\mathcal{L}_{\text{train}}\left(\boldsymbol{\phi}^{t}\right)\right\|_2^2 \leq \frac{2}{\eta_t}\mathbb{E}\left[\mathcal{L}_{\text{train}}\left(\boldsymbol{\phi}^{t}\right) - \mathcal{L}_{\text{train}}\left(\boldsymbol{\phi}^{t+1}\right) \mid \phi_t\right] +$$

$$\left(1 + L\eta_t\right)\frac{L_{\phi\mathcal{H}}^2}{\mu}(\delta_{\boldsymbol{H}} + \delta_{\boldsymbol{X}}) + 2L\eta_t\sigma^2$$

Finally, taking a telescopic sum over $t = \{1, \ldots, T\}$ yields

$$\sum_{t=0}^{T-1}\left\|\nabla\mathcal{L}_{\text{train}}\left(\boldsymbol{\phi}^{t}\right)\right\|_2^2$$

$$\leq \sum_{t=0}^{T-1}\frac{2}{\eta_t}\mathbb{E}\left[\mathcal{L}_{\text{train}}\left(\boldsymbol{\phi}^{t}\right) - \mathcal{L}_{\text{train}}\left(\boldsymbol{\phi}^{t+1}\right) \mid \phi_t\right]$$

$$+\sum_{t=0}^{T-1}\left(1 + L\eta_t\right)\frac{L_{\phi\mathcal{H}}^2}{\mu}(\delta_{\boldsymbol{H}} + \delta_{\boldsymbol{X}}) + \sum_{t=0}^{T-1}2L\eta_t\sigma^2$$

$$\leq 4\sqrt{\sigma^2 LDT} + \frac{2L_{\phi\mathcal{H}}^2}{\mu}(\delta_{\boldsymbol{H}} + \delta_{\boldsymbol{X}})$$

The last line is due to the choice that $\eta_t = \sqrt{\frac{D}{TL\sigma^2}}$ and $T \geq \frac{LD}{\sigma^2}$, where $D$ is the diameter of the loss.

Dividing both sides by $T$ concludes the proof. $\qquad\square$

# B. Details of the Threat Model and Attacker's Goals

### B.1. Threat Model

Adversarial attacks are small deliberate perturbations of data samples in order to achieve the outcome desired by the attacker when applied to the machine learning model. The attacker thus defines its own goal by considering (i) its knowledge about the data and the target model, and (ii) constraints it has about the adversarial perturbations it is allowed to perform.

**Attack setting.** We consider *gray-box* setting where the attacker does not have access to the learned model parameters $\theta^*$. However, it has access to the entire hypergraph structure as represented by the incidence matrix $\boldsymbol{H}$, training node labels $\boldsymbol{y}_L$, and model architecture. Hence, the attacker's goal is to devise a gray-box, untargeted/global attack. Without the knowledge of the model architecture, the attacker is allowed to use a surrogate model to construct perturbations.

**Attacker's constraints.** Following the GNN attack literature (Zügner et al., 2018; Zügner & Günnemann, 2019), we model realistic attacks through explicit unnoticeability constraints. In our setting, these constraints limit the number of structural edits, bound feature perturbations, and regularize changes in local hypergraph statistics such as node degrees.

**I. Structural perturbation budget:** The number of modifications to the hypergraph structure is limited by

$$\|\Delta\boldsymbol{H}\|_0 \leq \delta_H,$$

where $\delta_H$ is a predefined budget. Note that, as the entries in $\Delta\boldsymbol{H}$ are in $\{0, \text{-}1, 1\}$, the elementwise matrix norms $\|\Delta\boldsymbol{H}\|_1 = \|\Delta\boldsymbol{H}\|_0$. Hence this is consistent with the budget formulation in Equation 1.

**II. Feature perturbation bound:** The perturbation to node features is bounded in $l_\infty$-norm:

$$\|\Delta\boldsymbol{X}\|_\infty \leq \delta_X,$$

where $\delta_X$ controls the maximum allowable change per feature.

**III. Local-structure regularization:** To discourage overly concentrated structural edits, the attack objective penalizes large node-degree deviations.

## B.2. Attacker's Goal

The attacker's goal is not to misclassify any pre-defined target node(s), but rather to cause an overall drop in a target model's performance on the node classification task. To be precise, it wants to decrease the accuracy of a node classification algorithm after training on the perturbed hypergraph. In other words, it wants to have the test nodes classified as a class different from the true class. Specifically, the attacker seeks to find perturbations $\Delta H$ and $\Delta X$ such that the trained hypergraph neural network $f_{\theta^*}$ misclassifies nodes in $V_U$. Formally, the attack objective is to maximise the classification loss on unlabeled nodes:

$$\max_{\Delta X, \Delta H} \mathcal{L}_{atk}(f_{\theta^*}(H + \Delta H, X + \Delta X), y_U) \quad \textbf{s.t.}$$

$$\theta^* = \arg\min_{\theta} \mathcal{L}_{tr}(f_{\theta}(H + \Delta H, X + \Delta X), y_L)$$

and the unnoticeability constraints I-III. Here, $\mathcal{L}_{atk}$ is the loss function that the attacker wants to maximise. This formulation represents a bilevel optimisation problem, where the inner optimisation corresponds to training the target Hypergraph Neural Network on the perturbed data, and the outer optimisation aims to maximise the attacker's loss on unlabeled nodes.

In our case, the attacker wants to worsen the generalisation ability of $f_{\theta^*}$ on the unlabeled nodes. One way to achieve this is to use $\mathcal{L}_{atk} = \mathcal{L}_{tr} = \mathcal{L}_{CE}(y_L, f_{\theta^*}(V_L))$ where CE indicates cross-entropy loss, and by $f_{\theta^*}(V_L)$ we mean the evaluation of trained model on labeled nodes $V_L$. The reason for using labeled node is not because it is realistic, but rather it serves as a worst-case baseline, since a model with a bad training accuracy would most likely have a worse test accuracy.

Let us suppose the attacker obtained its optimal perturbations $(\Delta H^*, \Delta X^*)$ and constructs the perturbed hypergraph structure $H + \Delta H^*$ and feature matrix $X + \Delta X^*$. Finally, the attacker uses the perturbed hypergraph $(H + \Delta H^*, X + \Delta X^*)$ to re-train the target model. Since the attack occurred before the target model training, this is a *poisoning* attack scenario. This is in contrast to *evasion* scenario where the attack happens after the model is already trained.

Optimising such a bilevel problem is highly challenging due to the discrete nature of the incidence matrix $H$. Thus, the search space is $\mathcal{O}\left(\binom{mn}{\delta}\right)$, which is infeasible to enumerate. Furthermore, many hypergraph neural networks, such as HyperGCN, are not differentiable; hence, they are not suitable to be used as $f_{\theta}$. More importantly, our theoretical result (Theorem 2) shows that the hypergraph Laplacian

grows quadratically with structural perturbation budget, and inversely proportional to the minimum node degree $d_{\min}$ and hyperedge degree $\delta_{\min}$. To be precise, the attacks targeting low node-degree or hyperedge degree regions exploit the sensitivity of the hypergraph Laplacian to these two quantities to devise a strong attack. In HGNNS (especially those based on spectral methods), the Laplacian governs the entire message propagation dynamics. Due to these considerations, we define a new meta-objective function based on the Laplacian and use it as the attacker's loss function, $\mathcal{L}_{atk}$.

## C. Attack Effectiveness on Cora

We present the effectiveness analysis of our algorithms on Citeseer and Cora-CA in the main paper (Tables 2-3). In Table 6, we present the results on Cora. The findings resonate with what we observed on Citeseer and Cora-CA. `MeLA`-PGD performs better than other baseline attacks across both poisoning and evasion settings, with `MeLA`-FGSM performing slightly inferior yet superior to other baselines.

## D. Ablation Study for Attack

We study the effect of various design choices of our attack `MeLA`. We conduct ablation studies to compare results in both poisoning and evasion settings.

**Choice of hypergraph Laplacian.** We deploy other types of Laplacian (Schaub et al., 2020; Young et al., 2024), such as the Hodge Laplacian, to see if using them results in a stronger attack. The 0-th order Hodge Laplacian $L^h$ acts on node-level signals in an oriented hypergraph. Mathematically,

$$L^h = B_1 B_1^T$$

where $B_1$ is the signed boundary matrix that maps oriented hyperedges to the constituent nodes. The boundary matrix $B_1 \in \{-1, 1, 0\}$ is defined by assigning canonical orientation to each hyperedge:

$$(B_1)_{i,e} = \begin{cases} +1 & \text{if } v_i = v_0 = min(e) \\ -1 & \forall v_i \in e \setminus v_0 \end{cases}$$

We replace the $\mathcal{L}_{\text{smooth}}$ term with the following:

$$\mathcal{L}_{\text{smooth}} = \left\| L^h_{pert} Z_{pert} - L^h_{orig} Z_{orig} \right\|_2$$

If we consider embeddings $Z$ as potentials in nodes, $L^h Z$ captures the net disagreement in potential across hyperedges. Hence, the attacker wants to disrupt flows across hyperedges. Table 8 shows the comparison between Zhou's Laplacian and Hodge Laplacian on the effectiveness of `MeLA`-PGD.

*Table 6.* Comparison of different adversarial attacks (poisoning and evasion) for Cora. Drop = median relative % accuracy drop.

| Attack | HyperMLP (75.45 ± 0.42) | | AllsetTrans. (77.84 ± 0.54) | | HGNN (78.46 ± 0.48) | |
|---|---|---|---|---|---|---|
| | Poison / Drop (P) | Evasion / Drop (E) | Poison / Drop (P) | Evasion / Drop (E) | Poison / Drop (P) | Evasion / Drop (E) |
| RandFeat | 72.17 ± 1.02 / 4.5 | 71.49 ± 0.88 / 5.7 | 76.04 ± 0.92 / 1.7 | 76.22 ± 0.52 / 1.9 | 76.37 ± 0.76 / 2.8 | 76.60 ± 1.05 / 2.3 |
| RandFlip | 73.59 ± 0.87 / 2.1 | 75.45 ± 0.42 / 0.0 | 72.64 ± 1.70 / 6.7 | 71.88 ± 0.79 / 7.8 | 68.04 ± 1.06 / 12.8 | 68.98 ± 1.43 / 12.1 |
| GradArgMax | 75.33 ± 0.72 / 0.2 | 75.45 ± 0.42 / 0.0 | 75.98 ± 0.60 / 1.9 | 74.86 ± 0.82 / 4.0 | 75.30 ± 1.00 / 4.3 | 74.36 ± 0.67 / 5.1 |
| DICE | 74.03 ± 0.57 / 2.0 | 75.45 ± 0.42 / 0.0 | 75.04 ± 1.47 / 3.8 | 74.39 ± 1.38 / 3.8 | 70.40 ± 1.13 / 9.9 | 71.94 ± 0.68 / 8.5 |
| HyperAttack | 75.48 ± 0.43 / 0.0 | 75.45 ± 0.42 / 0.0 | 74.71 ± 0.40 / 4.2 | 74.65 ± 0.17 / 3.8 | 66.82 ± 1.03 / 14.5 | 72.88 ± 1.19 / 7.0 |
| MeLA-FGSM | 67.21 ± 0.38 / 11.0 | 42.81 ± 0.34 / 43.5 | 55.57 ± 4.03 / 26.8 | 41.06 ± 0.50 / 47.2 | 67.65 ± 0.44 / 13.7 | 41.42 ± 0.44 / 47.1 |
| MeLA-PGD | 65.29 ± 1.27 / **14.0** | 39.23 ± 0.85 / **48.1** | 49.78 ± 7.31 / **32.9** | 39.97 ± 0.81 / **49.0** | 55.42 ± 0.57 / **29.5** | 39.88 ± 0.31 / **49.0** |

*Table 7.* Choice of loss components of MeLA-PGD (HGNN model).

| Loss components | Citeseer (74.13 ± 0.10) | Cora (78.46 ± 0.48) |
|---|---|---|
| $\mathcal{L}_{\text{train}}$ | 57.44 ± 2.75 / 21.3 | 58.64 ± 0.67 / 25.4 |
| $\mathcal{L}_{\text{stealth}}, \mathcal{L}_{\text{train}}$ | 57.44 ± 2.75 / 21.3 | 58.64 ± 0.67 / 25.4 |
| $\mathcal{L}_{\text{smooth}}, \mathcal{L}_{\text{stealth}}, \mathcal{L}_{\text{train}}$ | 53.79 ± 1.83 / **27.1** | 56.45 ± 1.26 / **27.7** |

*Table 8.* Choice of Laplacian on MeLA-PGD (HyperMLP model)

| Laplacians | Citeseer (73.77 ± 0.29) | | Cora (73.26 ± 1.12) | |
|---|---|---|---|---|
| | Poison/Drop (p) | Evasion/Drop (e) | Poison/Drop (p) | Evasion/Drop (e) |
| Zhou | 61.18 ± 1.66 / **17.4** | 40.03 ± 1.03 / **44.9** | 61.18 ± 1.66 / **17.4** | 40.03 ± 1.03 / **44.9** |
| Hodge | 62.68 ± 2.00 / 13.8 | 47.40 ± 3.33 / 37.1 | 63.89 ± 1.54 / 16.9 | 49.32 ± 2.68 / 35.3 |

*Table 9.* Impact of joint structural and feature perturbations on MeLA-PGD (HyperMLP model).

| Ptb type | Citeseer (73.89 ± 0.42) | | Cora (72.23 ± 1.45) | |
|---|---|---|---|---|
| | Poison/Drop (p) | Evasion/Drop (e) | Poison/Drop (p) | Evasion/Drop (e) |
| $\delta_X = 0.05, \delta_H = 0\%$ | 71.81 ± 0.54 / 2.6 | 47.75 ± 1.09 / 35.8 | 72.58 ± 1.01 / 0.0 | 39.56 ± 0.32 / 45.1 |
| $\delta_X = 0.05, \delta_H = 20\%$ | 45.77 ± 1.71 / 36.7 | 39.57 ± 1.60 / 47.0 | 55.04 ± 2.54 / 23.2 | 29.63 ± 1.36 / 58.6 |

We observe that Zhou's Laplacian is more effective across various attack scenarios in both datasets.

**Significance of loss components.** Table 7 shows the significance of using the loss terms, in particular $\mathcal{L}_{\text{stealth}}$ and $\mathcal{L}_{\text{smooth}}$, on Citeseer and Cora under poisoning settings. We observe that the accuracy does not drop more when $\mathcal{L}_{\text{stealth}}$ is added to $\mathcal{L}_{\text{train}}$. This is because $\mathcal{L}_{\text{stealth}}$ regularizes node-degree changes; by itself, it is not designed to increase attack severity. However, the presence of $\mathcal{L}_{\text{smooth}}$ on top of both $\mathcal{L}_{\text{stealth}}$ and $\mathcal{L}_{\text{train}}$ improves the attack strength, as we observe an increased drop in accuracy across these datasets. For instance, on Cora, the median accuracy drop increases from 25.4% to 27.7%.

**Significance of structural and feature perturbations.** We evaluate the importance of considering structure and features in MeLA-PGD and present the results in Table 9. We observe that, in general, considering both structure and feature perturbation (as indicated by positive $\delta_X$ and positive $\delta_H$) yields a more effective MeLA-PGD attack than considering only feature perturbation.

**Significance of surrogate model.** In gray-box settings, a key success factor is how well the adversarial examples

*Table 10.* Comparison of adversarial test accuracy (± std) and median drop (%) under poisoning attacks.

| Attack (Surrogate) | Citeseer | Cora | Cora-CA |
|---|---|---|---|
| MeLA-PGD (MLP) | 66.71 ± 0.98 / 10.3 | 70.46 ± 1.72 / 11.1 | 77.93 ± 0.69 / 6.2 |
| MeLA-PGD (LinHGNN) | 53.79 ± 1.83 / **27.1** | 56.45 ± 1.26 / **27.7** | 73.21 ± 0.74 / **11.8** |
| MeLA-FGSM (MLP) | 39.08 ± 5.53 / **48.9** | 60.95 ± 1.52 / **22.4** | 76.19 ± 0.45 / 7.9 |
| MeLA-FGSM (LinHGNN) | 68.41 ± 0.82 / 7.2 | 67.65 ± 0.44 / 13.7 | 73.44 ± 0.62 / **11.4** |

generated using the surrogate model transfer to the unknown target model. In the main paper, we evaluated attack performance (of various target models) by using a linearized 2-layer HGNN (Feng et al., 2019) (referred to as LinHGNN) as a surrogate. Here we consider MLP, an additional surrogate to evaluate MeLA-D and MeLA-FGSM's effectiveness under poisoning setting. The goal is to analyze how the expressiveness of different surrogates affects attack strength. From Table 10 we observe that, MeLA-PGD with LinHGNN surrogate consistently outperforms MLP across all datasets, producing higher median accuracy drops and lower adversarial test accuracy. We also find that, MeLA-FGSM with MLP surrogate can also yield stronger attacks than MeLA-FGSM with LinHGNN, particularly on Cora and Citeseer datasets. This suggests a sharper gradient directions is obtained from the unstructured surrogate (MLP) on these dataset. This observation highlights that while structure-aware surrogates are crucial for multi-step optimization-based attacks like MeLA-PGD, simple models like MLP can still yield transferable perturbations in single-step attacks like FGSM, depending on the dataset.

# E. Defense Effectiveness.

### E.1. Performance of MeLA-D on other target models.

In the main paper, we presented the utility of adversarial training MeLA-D in defending a target model (HyperMLP) against adversarial attack. Here, we present another target model, HGNN, to demonstrate the effectiveness of adversarial training across different datasets and attacks in the poisoning setting.

Table 11 shows that the robustified MeLA-D+HGNN offers a certain degree of robustness gain against adversarially perturbed datasets obtained by various attacks.

*Table 11.* Adversarial-training results for `MeLA`-D+HGNN under poisoning attacks. Gain = avg. accuracy gain over runs.

| Attack | CiteSeer | | Cora-CA | | Cora | |
|---|---|---|---|---|---|---|
| | HGNN | MeLA-D+HGNN / Gain | HGNN | MeLA-D+HGNN / Gain | HGNN | MeLA-D+HGNN / Gain |
| Rand-Feat | $66.30 \pm 1.44$ | $72.92 \pm 0.37$ / 6.62 | $79.00 \pm 0.72$ | $79.70 \pm 0.82$ / 0.71 | $72.29 \pm 0.74$ | $75.81 \pm 1.16$ / 3.52 |
| Rand-Flip | $62.56 \pm 1.16$ | $66.35 \pm 1.22$ / 3.79 | $75.10 \pm 0.85$ | $76.87 \pm 1.05$ / 1.77 | $66.20 \pm 1.32$ | $69.39 \pm 1.33$ / 3.19 |
| GradArgMax | $66.84 \pm 0.53$ | $71.62 \pm 0.68$ / 4.78 | $77.81 \pm 0.71$ | $78.26 \pm 0.54$ / 0.44 | $70.87 \pm 1.09$ | $71.99 \pm 1.20$ / 1.12 |
| DICE | $62.97 \pm 0.62$ | $67.58 \pm 0.79$ / 4.61 | $73.18 \pm 0.64$ | $74.15 \pm 0.60$ / 0.97 | $66.56 \pm 0.60$ | $69.81 \pm 1.46$ / 3.25 |
| HyperAttack | $68.77 \pm 0.25$ | $73.45 \pm 0.47$ / 4.69 | $79.79 \pm 0.32$ | $80.24 \pm 0.63$ / 0.44 | $73.71 \pm 0.59$ | $76.16 \pm 1.32$ / 2.45 |
| MeLA-FGSM | $45.75 \pm 0.44$ | $73.45 \pm 0.78$ / 27.71 | $23.90 \pm 0.38$ | $77.40 \pm 1.01$ / 53.50 | $39.79 \pm 0.71$ | $75.24 \pm 1.76$ / 35.45 |
| MeLA-PGD | $35.02 \pm 2.94$ | $66.86 \pm 2.74$ / **31.84** | $19.41 \pm 0.44$ | $75.69 \pm 1.60$ / **56.28** | $37.34 \pm 0.37$ | $73.23 \pm 1.75$ / **35.89** |

*Table 12.* Comparison of `MeLA`-D+HGNN with baseline defenses: HGNN-Shield and ELasso-HGNN under `MeLA`-PGD poisoning. Base Acc. (Poisoned) is the vanilla HGNN poisoned accuracy used for the corresponding row; Gain = Defense Acc. (Poisoned) - Base Acc. (Poisoned) averaged over runs.

| Defense | Dataset | Base Acc. (Poisoned) | Defense Acc. (Clean) | Defense Acc. (Poisoned) / Gain |
|---|---|---|---|---|
| MeLA-D+HGNN | Cora | $37.34 \pm 0.37$ | $76.13 \pm 1.32$ | **$73.23 \pm 1.75$ / 35.89** |
| HGNN-Shield | Cora | $39.85 \pm 0.41$ | $47.68 \pm 4.16$ | $55.78 \pm 1.25$ / 15.92 |
| ELasso-HGNN | Cora | $39.85 \pm 0.41$ | $46.79 \pm 4.94$ | $55.16 \pm 1.40$ / 15.30 |
| MeLA-D+HGNN | CiteSeer | $35.02 \pm 2.94$ | $73.45 \pm 0.47$ | **$66.86 \pm 2.74$ / 31.84** |
| HGNN-Shield | CiteSeer | $39.78 \pm 3.42$ | $40.36 \pm 19.85$ | $40.97 \pm 11.14$ / 1.18 |
| ELasso-HGNN | CiteSeer | $39.78 \pm 3.42$ | $45.29 \pm 15.38$ | $45.65 \pm 8.34$ / 5.87 |
| MeLA-D+HGNN | Cora-CA | $19.41 \pm 0.44$ | $80.24 \pm 0.63$ | **$75.69 \pm 1.60$ / 56.28** |
| HGNN-Shield | Cora-CA | $19.85 \pm 0.55$ | $56.81 \pm 2.56$ | $71.49 \pm 0.49$ / 51.64 |
| ELasso-HGNN | Cora-CA | $19.85 \pm 0.55$ | $55.57 \pm 3.06$ | $71.49 \pm 0.79$ / 51.64 |

### E.2. Comparison with Existing Defenses.

We conducted comparison with two recent defenses: HGNN-Shield (Feng et al., 2025), a purification-based defense for hypergraphs, and Elasso-HGNN, which is an adaptation of the robust structure refinement module of ElassoGNN (Jiang et al., 2023). Table 12 shows that `MeLA`-D+HGNN yields more robustness gain than these baselines.

## F. Analyzing the Effectiveness of **MeLA**

### F.1. Characterizing attacks using Degree-Drift.

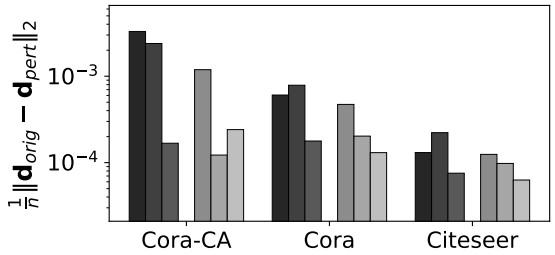

*Figure 3.* Node-degree drift under poisoning attacks on HyperMLP. As optimized `MeLA` perturbations may be concentrated on fewer nodes or hyperedges, we observe larger local degree drift than random baselines.

Degree drift provides a local view of how perturbations re-

distribute node-hyperedge incidences. This complements the operator-level and feature-level analysis in Figure 2: `MeLA`-PGD preserves a smaller raw Laplacian drift while using more targeted incidence updates than uniformly random baselines.

### F.2. Node Embedding Visualization

In order to understand how much the attack shifted the embedding geometry, we visualize the original and adversarial embeddings using t-SNE (Van der Maaten & Hinton, 2008) in $\mathbb{R}^2$. Figure 4 compares the t-SNE embeddings obtained from two datasets. We observe that in most regions, the red (adversarial) points remain near their corresponding black (original) points, but not exactly overlapping. This indicates a subtle yet structured embedding shift, indicating that `MeLA`-D perturbs the decision boundaries without drastically changing the global structure. At certain cluster boundaries, adversarial points intrude into other clusters, suggesting increased inter-class confusion. This further aligns with the adversary's objective to degrade classification performance by blurring decision boundaries.

### F.3. Node Embeddings Drift

The drift in node embedding $\|\boldsymbol{Z}_{orig} - \boldsymbol{Z}_{pert}\|_F$ encodes how much the learned node representations have changed

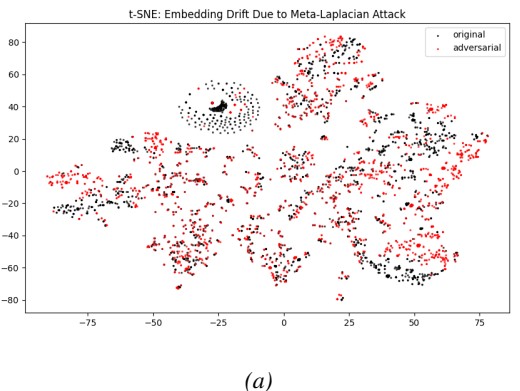
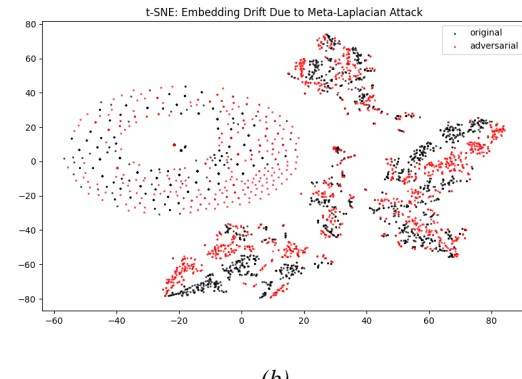

(a)

(b)

*Figure 4.* Visualization of original node embeddings $Z_{\text{orig}}$ from clean model and node embeddings from poisoned model $Z_{\text{pert}}$: (a) Cora-CA and (b) Citeseer

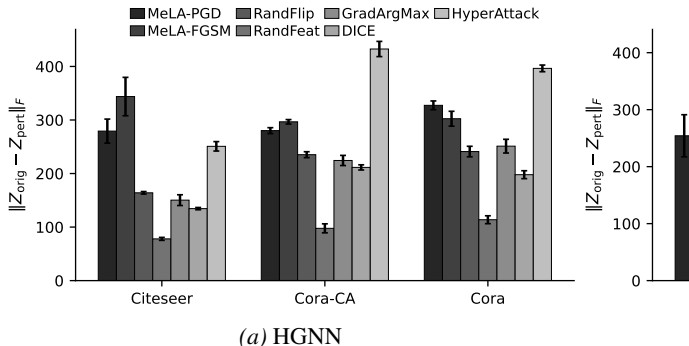
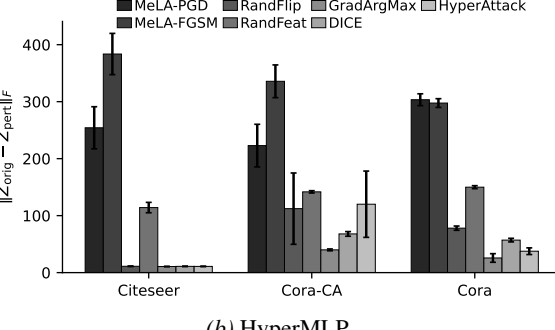

(a) HGNN

(b) HyperMLP

*Figure 5.* Node embedding shift under different attacks. Larger values indicate stronger movement of the learned representation.

due to an attack. Since embeddings capture both local semantics from node features and structural context from neighbors in the hypergraph structure, a large embedding drift usually indicates that the attack has disrupted either the structural information flow or the semantic alignment of similar nodes.

Figure 5 shows that different attacks perturb the learned representation in different ways. HyperAttack often yields the largest raw embedding displacement for HGNN, consistent with its aggressive rewiring of high-order neighborhoods. By contrast, `MeLA`-PGD and `MeLA`-FGSM produce decision-directed drift: their Laplacian-aware gradients jointly perturb structure and features, and the resulting representation movement aligns with the large accuracy drops in Tables 2–3. Thus, the strongest attack is not necessarily the one that maximizes raw embedding displacement, but the one that moves representations in directions that most damage the classifier.

### F.4. `MeLA`-PGD on a Toy Example

We generate a toy hypergraph with 12 nodes and 6 hyperedges to visualize how `MeLA`-PGD changes the decision

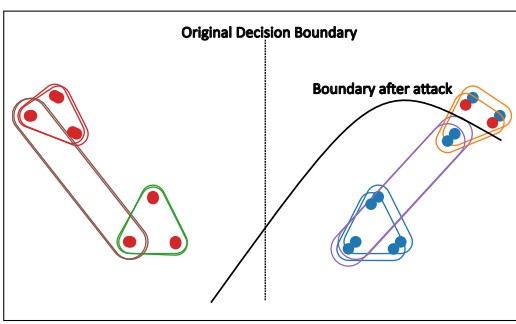

*Figure 6.* Effect on HGNN's decision boundary on a toy hypergraph due to attack `MeLA`-PGD.

boundary of HGNN (Feng et al., 2019). The node features are coordinates in $\mathbb{R}^2$. The nodes with $+x$ coordinates belong to the blue class while the nodes with $-x$ coordinates belong to the red class. The dotted line indicates $x = 0$ line that separates nodes from the two classes is the initial decision boundary of HGNN.

Figure 6 shows that two blue classes have been misclassified after the attack. This shift suggests that the decision boundary has been affected by attack `MeLA`-PGD.

*Table 13.* Statistical ranking of attacks using a non-parametric Friedman test (Citeseer dataset).

| Attack | HyperMLP | | AllSetTransformer | | HGNN | |
|---|---|---|---|---|---|---|
| | Rank (Poison) | Rank (Evasion) | Rank (Poison) | Rank (Evasion) | Rank (Poison) | Rank (Evasion) |
| | Test stat: 27.9 | Test stat: 30.0 | Test stat: 27.2 | Test stat: 29.2 | Test stat: 29.1 | Test stat: 29.7 |
| | p-value: $1.0{\times}10^{-4}$ | p-value: $3.9{\times}10^{-5}$ | p-value: $1.3{\times}10^{-4}$ | p-value: $5.6{\times}10^{-5}$ | p-value: $5.7{\times}10^{-5}$ | p-value: $4.6{\times}10^{-5}$ |
| RandFeat | 3.0 | 3.0 | 6.2 | 7.0 | 7.0 | 7.0 |
| RandFlip | 5.0 | 5.5 | 3.0 | 3.0 | 2.7 | 3.0 |
| GradArgMax | 6.0 | 5.5 | 5.8 | 5.7 | 6.0 | 5.8 |
| DICE | 5.3 | 5.5 | 4.2 | 4.2 | 4.4 | 4.0 |
| HyperAttack | 5.7 | 5.5 | 5.8 | 5.1 | 2.3 | 5.2 |
| MeLA-FGSM | 2.0 | 2.0 | 2.0 | 2.0 | 4.6 | 2.0 |
| MeLA-PGD | **1.0** | **1.0** | **1.0** | **1.0** | **1.0** | **1.0** |

*Table 14.* Statistical ranking of attacks using a non-parametric Friedman test (Cora dataset).

| Attack | HyperMLP | | AllSetTransformer | | HGNN | |
|---|---|---|---|---|---|---|
| | Rank (Poison) | Rank (Evasion) | Rank (Poison) | Rank (Evasion) | Rank (Poison) | Rank (Evasion) |
| | Test stat: 28.9 | Test stat: 30.0 | Test stat: 25.2 | Test stat: 27.9 | Test stat: 27.8 | Test stat: 29.0 |
| | p-value: $6.4{\times}10^{-5}$ | p-value: $3.9{\times}10^{-5}$ | p-value: $3.1{\times}10^{-4}$ | p-value: $9.6{\times}10^{-5}$ | p-value: $1.0{\times}10^{-4}$ | p-value: $6.2{\times}10^{-5}$ |
| RandFeat | 3.0 | 3.0 | 6.4 | 7.0 | 7.0 | 7.0 |
| RandFlip | 4.4 | 5.5 | 3.4 | 3.0 | 3.3 | 3.0 |
| GradArgMax | 6.2 | 5.5 | 5.8 | 5.0 | 6.0 | 5.8 |
| DICE | 4.7 | 5.5 | 5.2 | 4.8 | 4.9 | 4.4 |
| HyperAttack | 6.7 | 5.5 | 4.2 | 5.2 | 2.6 | 4.8 |
| MeLA-FGSM | 2.0 | 2.0 | 1.8 | 2.0 | 3.2 | 2.0 |
| MeLA-PGD | **1.0** | **1.0** | **1.2** | **1.0** | **1.0** | **1.0** |

*Table 15.* Statistical ranking of attacks using a non-parametric Friedman test (Cora-CA dataset).

| Attack | HyperMLP | | AllSetTransformer | | HGNN | |
|---|---|---|---|---|---|---|
| | Rank (Poison) | Rank (Evasion) | Rank (Poison) | Rank (Evasion) | Rank (Poison) | Rank (Evasion) |
| | Test stat: 25.5 | Test stat: 29.6 | Test stat: 29.5 | Test stat: 26.3 | Test stat: 27.5 | Test stat: 29.1 |
| | p-value: $2.7{\times}10^{-4}$ | p-value: $4.7{\times}10^{-5}$ | p-value: $4.8{\times}10^{-5}$ | p-value: $1.9{\times}10^{-4}$ | p-value: $1.2{\times}10^{-4}$ | p-value: $5.8{\times}10^{-5}$ |
| RandFeat | 4.3 | 3.0 | 7.0 | 6.8 | 7.0 | 7.0 |
| RandFlip | 4.2 | 5.5 | 4.8 | 4.0 | 4.0 | 3.2 |
| GradArgMax | 7.0 | 5.5 | 6.0 | 6.0 | 6.0 | 5.9 |
| DICE | 5.2 | 5.5 | 4.1 | 4.2 | 4.5 | 4.0 |
| HyperAttack | 4.3 | 5.5 | 3.1 | 4.0 | 3.3 | 4.9 |
| MeLA-FGSM | 2.0 | 1.8 | 2.0 | 1.8 | 2.0 | 2.0 |
| MeLA-PGD | **1.0** | **1.2** | **1.0** | **1.2** | **1.2** | **1.0** |

# G. Statistical Significance of Results

We now turn to testing whether there is a statistically significant difference in performance, measured by accuracy drop, across multiple attack types evaluated on the same set of seeds and datasets. Our null hypothesis is "all attack types have equivalent performance", whereas the alternative hypothesis is "at least one attack performs differently." We employ a non-parametric Friedman test (Demšar, 2006), which is a preferred procedure in formal statistical inference due to its minimal assumptions about the underlying data. Ranks are computed from the regenerated seven-attack grid using the same nonnegative seed-level accuracy drops reported in Tables 2, 6, and 3; lower rank indicates a stronger attack.

In Tables 13-15, the p-values are consistently below 0.05, so we reject the null hypothesis across target models and poisoning/evasion settings. The proposed attacks, especially MeLA-PGD, rank at the top across the datasets, confirming statistically significant differences among attacks.

# H. Choice of Hyperparameters

Tables 16–18 report the hyperparameters used in the attack evaluation. HyperMLP target models are trained for 100 epochs, while HGNN and AllSetTransformer target models are trained for 1000 epochs with early stopping based on validation loss. All attacks use $\delta_X = 0.05$ and $\delta_H = 0.2$ unless stated otherwise. In the tables, **Lr** indicates learning rate, **Wd** indicates weight decay, **Drop.** indicates dropout rate, and $\tau_s/\tau_a$ denotes surrogate/attack training epochs. Tables 19–20 report the hyperparameters used in the adversarial-training evaluation; $\tau_d$ denotes the number of epochs used for perturbation generation during defense.

*Table 16.* Choice of attack hyperparameters for HyperMLP. All HyperMLP attacks use `LinHGNN` surrogate.

| Dataset/attack | Lr | Wd | Hidden dim. | #Layers | HyperMLP-$\alpha$ | Drop. | $\alpha,\beta,\gamma$ | T | $\tau_s/\tau_a$ | $\eta_H$ | $\eta_X$ |
|---|---|---|---|---|---|---|---|---|---|---|---|
| All datasets, RandFeat/RandFlip/GradArgMax/DICE | 0.001 | 0.0 | 512 | 2 | 0.005 | 0.1 | 0.005,1,4 | – | 80 | 0.01 | 0.01 |
| All datasets, HyperAttack | 0.001 | 0.0 | 512 | 2 | 0.005 | 0.1 | 0.005,1,4 | – | 80 | 0.01 | 0.01 |
| All datasets, MeLA-FGSM | 0.001 | 0.0 | 512 | 2 | 0.005 | 0.1 | 0.005,1,4 | – | 80 | 0.01 | 0.01 |
| All datasets, MeLA-PGD except Cora-CA MeLA-PGD | 0.001 | 0.0 | 512 | 2 | 0.005 | 0.1 | 0.005,1,4 | 30 | 80 | 0.01 | 0.01 |
| Cora-CA MeLA-PGD | 0.001 | 0.0 | 512 | 2 | 0.005 | 0.1 | 64,1,4 | 30 | 80 | 0.02 | 0.01 |

*Table 17.* Choice of attack hyperparameters for HGNN.

| Dataset/attack | Lr | Wd | Hidden dim. | #Layers | Drop. | $\alpha,\beta,\gamma$ | T | $\tau_s/\tau_a$ | $\eta_H$ | $\eta_X$ |
|---|---|---|---|---|---|---|---|---|---|---|
| All datasets, RandFeat/RandFlip/GradArgMax/DICE/MeLA-FGSM | 0.001 | 0.0 | 512 | 1 | 0.5 | 0.1,1,4 | – | 80 | 0.01 | 0.01 |
| Citeseer/Cora/Cora-CA MeLA-PGD | 0.001 | 0.0 | 512 | 1 | 0.5 | 0.1,1,4 | 50/30/20 | 80 | 0.01 | 0.01 |
| All datasets, HyperAttack | 0.001 | 0.0 | 512 | 1 | 0.5 | 0.1,1,4 | – | 80 | 0.01 | 0.01 |

*Table 18.* Choice of attack hyperparameters for AllSetTransformer.

| Dataset/attack | Lr | Wd | Hidden dim. | #MLP layers | #Heads | Drop. | $\alpha,\beta,\gamma$ | T | $\tau_s/\tau_a$ | $\eta_H$ | $\eta_X$ |
|---|---|---|---|---|---|---|---|---|---|---|---|
| All datasets, RandFeat/RandFlip/GradArgMax/DICE/MeLA-FGSM | 0.001 | 0.0 | 256 | 2 | 4 | 0.5 | 0.1,1,4 | – | 80 | 0.01 | 0.01 |
| All datasets, MeLA-PGD | 0.001 | 0.0 | 256 | 2 | 4 | 0.5 | 0.1,1,4 | 30 | 80 | 0.01 | 0.01 |
| All datasets, HyperAttack | 0.001 | 0.0 | 256 | 2 | 4 | 0.5 | 0.1,1,4 | – | 80 | 0.01 | 0.01 |

*Table 19.* Hyperparameters for `MeLA-D+HyperMLP`. All rows use $\alpha,\beta,\gamma = (0.005, 1, 1)$, $\eta_H = \eta_X = 0.01$, $\delta_X = 0.05$, $\delta_H = 0.2$.

| Dataset | Attack | Lr | Drop. | T | $\tau_d$ | Epochs |
|---|---|---|---|---|---|---|
| Cora-CA | RandFeat | 0.0005 | 0.1 | 15 | 200 | 100 |
| | RandFlip, DICE, HyperAttack | 0.0001 | 0.1 | 15 | 200 | 100 |
| | GradArgMax | 0.0001 | 0.1 | 12 | 200 | 100 |
| | MeLA-FGSM, MeLA-PGD | 0.001 | 0.1 | 15 | 400 | 400 |
| Cora | RandFeat | 0.001 | 0.1 | 15 | 400 | 400 |
| | RandFlip, DICE, HyperAttack | 0.0001 | 0.1 | 15 | 200 | 100 |
| | GradArgMax | 0.0001 | 0.1 | 12 | 200 | 100 |
| | MeLA-FGSM | 0.001 | 0.1 | 15 | 400 | 400 |
| | MeLA-PGD | 0.001 | 0.1 | 15 | 40 | 100 |
| Citeseer | RandFeat | 0.003 | 0.1 | 10 | 400 | 400 |
| | RandFlip, DICE, HyperAttack | 0.001 | 0.0 | 15 | 180 | 100 |
| | GradArgMax | 0.001 | 0.0 | 12 | 200 | 100 |
| | MeLA-FGSM, MeLA-PGD | 0.001 | 0.1 | 15 | 400 | 400 |

*Table 20.* Hyperparameters for `MeLA-D+HGNN`. All rows use $\alpha,\beta,\gamma = (1, 1, 4)$, $\eta_H = \eta_X = 0.01$, $\delta_X = 0.05$, $\delta_H = 0.2$, and final-checkpoint evaluation. The HGNN-Shield and ElassoHGNN defense baselines are trained on poisoned data with learning rate 0.01, weight decay $5 \times 10^{-4}$, hidden dimension 32, dropout 0.5, 400 epochs, and validation checkpoint selection. HGNN-Shield uses thresholds 0.05 and 0.015, while ElassoHGNN uses proximal regularization coefficient $\beta_{\text{elasso}} = 5 \times 10^{-5}$ with two inner optimization steps.

| Dataset | Attack | Lr | T | $\tau_d$ | Epochs |
|---|---|---|---|---|---|
| All datasets | All attacks | 0.001 | 20 | 300 | 200 |

