# Adversarial Attacks and Robust Training for Hypergraph Neural Networks

## Abstract

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

 not noticeable when (i) there is an upper bound on the frequency of structural and feature changes, and (ii) statistics such as degree distribution are preserved. `MeLA` measures stealth violation as the change in the degree distribution after perturbation: $\mathcal{L}_{\text{stealth}}(\theta^*, \Delta \boldsymbol{H}, \Delta \boldsymbol{X}) \triangleq \frac{1}{n}\|\boldsymbol{d}_{pert} - \boldsymbol{d}_{orig}\|_2^2$, where $\boldsymbol{d}_{orig}$ ($\boldsymbol{d}_{pert}$) indicates the vectors of node degrees in the clean graph (poisoned graph). Note that an attacker can consider other types of stealthiness measures, such as hyperedge cardinality preservation.

*(c)* $\mathcal{L}_{\text{smooth}}$. Hypergraph Laplacian smoothing encourages nodes connected through shared hyperedges to have similar embeddings (Agarwal et al., 2006; Feng et al., 2019). In `MeLA`, the attacker tries to maximize the discrepancy in smoothed embeddings before and after the attack. By doing so, the attack disrupts the propagation behavior induced by the original hypergraph structure. We define Laplacian smoothing loss as $\mathcal{L}_{\text{smooth}}(\theta^*, \Delta \boldsymbol{H}, \Delta \boldsymbol{X}) \triangleq \|\boldsymbol{L}_{pert}\boldsymbol{Z}_{pert} - \boldsymbol{L}_{orig}\boldsymbol{Z}_{orig}\|_2$. Here, $\boldsymbol{Z} = f_{\theta^*}(\boldsymbol{H}, \boldsymbol{X})$ and $\boldsymbol{Z}_{pert} = f_{\theta^*}(\boldsymbol{H} + \Delta \boldsymbol{H}, \boldsymbol{X} + \Delta \boldsymbol{X})$ are node embeddings obtained from the clean graph and the poisoned graph, respectively. We adopt Zhou's Laplacian (Section 2), though `MeLA` is compatible with other Laplacians. Putting these components together, the attacker optimizes the following meta-objective:

$$\mathcal{L}_{\text{meta}}(\theta^*, \Delta \boldsymbol{H}, \Delta \boldsymbol{X}) = \alpha \mathcal{L}_{\text{smooth}}(\theta^*, \Delta \boldsymbol{H}, \Delta \boldsymbol{X}) - \beta \mathcal{L}_{\text{stealth}}(\theta^*, \Delta \boldsymbol{H}, \Delta \boldsymbol{X}) + \gamma \mathcal{L}_{\text{train}}(\theta^*, \Delta \boldsymbol{H}, \Delta \boldsymbol{X}) \quad (3)$$

such that $\theta^* = \arg\min_\theta \mathcal{L}_{\text{train}}(f_\theta(\boldsymbol{H} + \Delta \boldsymbol{H}, \boldsymbol{X} + \Delta \boldsymbol{X}), y_L)$, and $\alpha, \beta, \gamma > 0$ are regularizing coefficients. The surrogate parameters $\theta^*$ are obtained by training on the clean data and are held fixed during the attack.

**Step 2: Deriving Perturbation Strategies.** Given perturbation budgets $\delta_H$ and $\delta_X$, `MeLA` employs gradient-based updates to approximately maximize the meta-objective. In particular, under $\ell_\infty$-bounded feature perturbations, a first-order approximation yields

$$\Delta \boldsymbol{X} = \delta_X \text{Sign}(\nabla_{\Delta X} \mathcal{L}_{\text{meta}}),$$

which is an analogue of the Fast Gradient Sign Method (FGSM) (Goodfellow et al., 2015). FGSM enjoys properties such as satisfying First-Order Stationary Condition (FOSC) for constrained optimization that ensures convergence of the attack (Wang et al., 2019), and being the best response of an attacker against any robust training algorithm (Pal & Vidal, 2020). For structural perturbations on binary incidence

---

**Algorithm 2** MeLA-D: Meta Laplacian Defense

---

**Require:** Hypergraph $(\boldsymbol{H}, \boldsymbol{X})$, Budgets $\delta_X, \delta_H$, learning rates $\eta_H, \eta_X$, iterations $T$, training epochs $\tau_d$
1: Compute $\boldsymbol{L}_{orig} \leftarrow \text{LAP}(\boldsymbol{H})$.
2: Initialize perturbations
$\quad \Delta \boldsymbol{H}^{(0)} \sim \mathcal{N}(0, \boldsymbol{I}), \Delta \boldsymbol{X}^{(0)} \sim \mathcal{N}(0, \boldsymbol{I})$
3: **for** $t = 1, \dots, T$ **do**
4: $\quad \boldsymbol{H}_{pert} \leftarrow \text{CLIP}(\boldsymbol{H} + \Delta \boldsymbol{H}^{(t-1)}, 0, 1), \quad \boldsymbol{X}_{pert} \leftarrow \boldsymbol{X} + \Delta \boldsymbol{X}^{(t-1)}, \boldsymbol{L}_{pert}^{(t)} \leftarrow \text{LAP}(\boldsymbol{H}_{pert})$
5: $\quad h_{\phi^*}^t \leftarrow$ Train $h_\phi^t$ for $\tau_d$ epochs on $(\boldsymbol{H}_{pert}, \boldsymbol{X}_{pert})$.
6: $\quad$ Compute clean logits $\boldsymbol{Z}_{orig}^{(t)} \leftarrow h_{\phi^*}^t(\boldsymbol{H}, \boldsymbol{X})$,
$\quad$ perturbed logits $\boldsymbol{Z}_{pert}^{(t)} \leftarrow h_{\phi^*}^t(\boldsymbol{H}_{pert}^{(t)}, \boldsymbol{X}_{pert}^{(t)})$.

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

 | 73.62 ± 0.55 / 2.3 | 75.45 ± 0.75 / 0.0 | 81.54 ± 0.68 / 2.6 | 77.43 ± 0.76 / 7.1 | 80.15 ± 0.81 / 3.2 | 77.70 ± 0.95 / 6.1 |
| MeLA-FGSM | 69.48 ± 0.29 / 7.9 | 28.39 ± 0.44 / 61.8 | 73.83 ± 0.76 / 11.5 | 26.91 ± 0.53 / 67.8 | 74.15 ± 0.68 / 11.1 | 23.90 ± 0.24 / 71.1 |
| MeLA-PGD | 68.54 ± 0.43 / **9.1**** | 27.09 ± 0.73 / **63.3**** | 71.76 ± 0.97 / **14.3**** | 21.68 ± 0.91 / **74.2**** | 72.64 ± 0.38 / **12.4**** | 19.59 ± 0.34 / **76.2**** |

*Table 3.* Effectiveness of robustly training HyperMLP across different evasion attacks. ** means statistically significant.

| Attack | Citeseer | | Cora-CA | | Cora | |
|---|---|---|---|---|---|---|
| | HyperMLP | MeLA-D+HyperMLP / Gain | HyperMLP | MeLA-D+HyperMLP / Gain | HyperMLP | MeLA-D+HyperMLP / Gain |
| *Clean* | 73.77 ± 0.29 | 72.44 ± 0.99/0.0 | 75.45 ± 0.75 | 73.83 ± 1.21/0.0 | 73.26 ± 1.12 | 74.03 ± 0.85/0.8 |
| RandFeat | 63.50 ± 1.48 | 71.52 ± 1.49/8.2 | 69.22 ± 0.80 | 73.29 ± 0.59/4.1 | 63.78 ± 0.61 | 73.56 ± 0.79/9.9 |
| RandFlip | 70.48 ± 0.39 | 72.80 ± 1.07/2.5 | 49.36 ± 0.70 | 73.56 ± 0.53/23.8 | 50.34 ± 1.06 | 73.50 ± 0.93/24.2 |
| GradArgMax | 70.48 ± 0.39 | 73.00 ± 1.09/3.0 | 49.36 ± 0.70 | 72.73 ± 0.95/23.9 | 50.34 ± 1.06 | 73.59 ± 1.17/23.9 |
| MeLA-FGSM | 48.67 ± 0.70 | 69.03 ± 0.66/20.7 | 28.12 ± 0.40 | 63.69 ± 1.90/35.3 | 39.79 ± 0.22 | 68.06 ± 0.69/28.2 |
| MeLA-PGD | 42.54 ± 0.37 | 68.96 ± 0.53/**26.3**** | 25.38 ± 1.10 | 65.49 ± 1.53/**39.6**** | 35.81 ± 0.60 | 67.65 ± 0.59/**31.8**** |

run on a system with 64 x 2 CPU cores, 500 GB RAM and a NVIDIA Tesla V100 GPU with 32 GB of HBM3e memory. We set $\delta_X = 0.05$, $\delta_H = 20\%$, and $\eta_H = \eta_X = 0.01$ in all experiments. Due to space constraint, we defer the sensitivity analysis for $\delta_X, \delta_H$ to Appendix. In addition, Appendix contains (a) ablation studies of our attacks and defense (e.g., choice of Laplacians, choice of surrogates, impact of the loss components, significance of structural and feature perturbations), (b) node embedding and decision boundary visualization due to attack, (c) Node embedding drifts due to attack, (d) experiments on a large-scale dataset, and (e) statistical significance tests.

**Datasets and Target Neural Networks.** Following existing works (Chien et al., 2021; Wang et al., 2023; Duta et al., 2023), our datasets include **Cora** and **Citeseer** cocitation networks from (Yadati et al., 2019), and the Cora coauthorship network (**Cora-CA**) (Chien et al., 2021) with 50%/25%/25% train/validation/test split.

| Datasets | $|V|$ | $|E|$ | avg. edge size | # features | # classes |
|---|---|---|---|---|---|
| **Cora-CA** | 2708 | 1072 | 4.2 ± 4.1 | 1433 | 7 |
| **Cora** | 2708 | 1579 | 3.0 ± 1.1 | 1433 | 7 |
| **Citeseer** | 3312 | 1079 | 3.2 ± 2.0 | 3703 | 6 |

As for target HGNNs, we have chosen HypergraphMLP (Tang et al., 2024), HGNN (Feng et al., 2019), and AllsetTransformer (Chien et al., 2021) as representative hypergraph neural networks. For defense, we train a HypergraphMLP. Experimental results with other SOTA HGNNs and datasets are deferred to the Appendix.

**Baseline Attacks.** **RandFlip** (Dai et al., 2018) randomly flips $\delta_H$ entries in the incidence matrix $\boldsymbol{H}$. **RandFeat** first generates a random matrix $\boldsymbol{R}$ where each entry $\boldsymbol{R}_{ij} \in \{-1, +1\}$ with equal probability. In other words, $\boldsymbol{R}_{ij} = 2B_{ij} - 1$ where $B_{ij} \sim \text{Bernoulli}(\frac{1}{2})$. Finally, it constructs perturbed features $X_{\text{pert}} = X_{\text{orig}} + \boldsymbol{R}_{ij}\delta_X$ so that $\|\Delta\boldsymbol{X}\|_\infty \leq \delta_X$. **GradArgMax** (Dai et al., 2018) selects top-$\delta_H$ pairs of indices from $\boldsymbol{H}$ as flipping them causes the largest change in the gradient of loss function $\mathcal{L}_{\text{train}}$. For a fair comparison, we use LinHGNN as a surrogate for computing $\left|\frac{\partial\mathcal{L}}{\partial\boldsymbol{H}}\right|$ across all the attacks.

**I. Attack Effectiveness.** We craft adversarial perturbations using various attacks and target models. Tables 1–2 show the comparison among the attacks in both poisoning and evasion settings (results for Cora is in Appendix). We measure attack effectiveness by median relative drop (%) in clean accuracy.

*Poisoning Setting.* Across all datasets, MeLA-PGD yields significantly higher degradation in test accuracy (up to $60\%$) compared to baseline methods such as RandFlip, RandFeat, and GradArgMax. In contrast, MeLA-FGSM, which relies on a single-step first-order approximation of the meta-gradient, exhibits limited effectiveness and performs comparably to the baselines. These results highlight the necessity of multi-step optimization, as single-step approximations are insufficient to exploit the nonlinear dependence of the hypergraph Laplacian on structural perturbations (Theorem 2).

*Evasion Setting.* (i) MeLA-PGD is the most effective, achieving up to $76\%$ accuracy drop (Cora-CA). (ii) In eva-

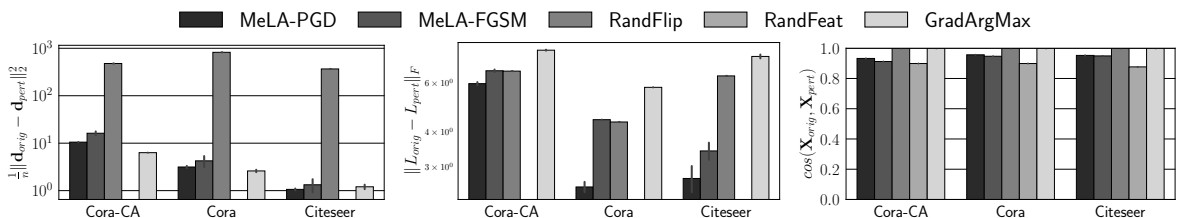

*Figure 1.* Stealthiness comparison: Node-degree drift (left), Laplacian drift (middle), and feature cosine similarity (right) are reported.

sion setting, MeLA-FGSM performs $3 - 12\times$ better than other baselines. Both MeLA-FGSM and MeLA-PGD optimize a meta-loss evaluated on a fixed surrogate model, which closely matches the evasion setting where the classifier parameters are frozen at test time. This objective alignment allows both attacks to outperform the baselines. We also adapt MeLA-PGD for attacking large-scale hypergraphs in mini-batches, e.g., DBLP-CA ($|V| = 41302, |E| = 22363$). MeLA-PGD reduces accuracy by 1.6% with a runtime of 140 seconds. We refer to Appendix for details.

**II. Attack stealthiness.** We analyze attack stealth along three complementary dimensions: structural visibility (via degree drift), operator-level impact (via Laplacian drift), and node-level semantic distortion (via feature drift).

**(a) Structural:** In Figure 1(left) we observe that RandFlip produces the largest degree drift while MeLA-PGD and GradArgMax exhibit relatively small one. As RandFeat only perturbs node features, there is no degree drift.

**(b) Operator-level:** As hypergraph Laplacian directly defines the linear operator used in hypergraph message passing, attacks that avoid inducing unnecessarily large Laplacian drift are less likely to be detected by spectral or operator-level defenses. Figure 1 (middle) shows that among structure-perturbing attacks, MeLA-PGD achieves the smallest Laplacian drift. Interestingly, despite being structurally stealthy (low degree-drift), GradArgMax is less stealthy in operator-level compared to MeLA-PGD and MeLA-FGSM. As the hypergraph Laplacian depends nonlinearly on inverse degree normalizations, a smaller degree drift does not necessarily imply smaller Laplacian drift. As shown in Theorem 2, indeed perturbing low-degree nodes induces large Laplacian drift while incurring only minor degree drift.

**(c) Semantic Distortion:** We measure the change in semantic features via mean cosine similarity between the original and perturbed node features. A high cosine similarity implies a limited distortion of the feature space, and hence, a more stealthy attack. *Figure 1 (right) shows that MeLA-PGD maintains high similarity due to its use of gradient-aligned feature updates with constrained step sizes.* RandFlip and GradArgMax perturb only the hypergraph structure, hence cosine similarity $\sim 1$. In contrast, RandFeat perturbs node features without gradient-based or structure-aware guidance, leading to reduced feature-level stealth.

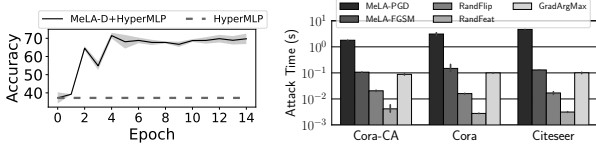

*(a)* MeLA-D: robustness gain     *(b)* Execution times

*Figure 2.* (a) Evolution of test accuracy of the robust and the baseline during adversarial training. (b) Attack execution times.

**III. Effectiveness of Defense.** We evaluate the robustness of HyperMLP and its adversarially trained variant MeLA-D+HyperMLP against all attacks and report the gain in robustness in Table 3. MeLA-D shows a moderate drop in accuracy on clean data, yet yields consistently higher accuracy against RandFeat, RandFlip, and GradArgMax attacks. Reinforcing its defensive ability, MeLA-D leads to a $20 - 40\%$ gain against stronger gradient-based attacks like MeLA-FGSM and MeLA-D itself.

In Figure 2, we demonstrate the accuracy of robust model MeLA-D+HyperMLP on MeLA-FGSM perturbed Cora across adversarial training epochs. MeLA-D+HyperMLP consistently maintains better accuracy than the baseline HyperMLP highlighting effectiveness of MeLA-D.

**IV. Computational Efficiency.** Figure 2b shows that Rand-Feat is the most efficient ($\sim 10$ ms), while MeLA-PGD is the least efficient among all attacks ($\sim 1 - 5$ sec). By trading off some attack strength, MeLA-FGSM provides a more efficient alternative ($\sim 100 - 200$ ms).

## 6. Conclusion and Future Directions

We proposed a novel generic meta-objective based framework MeLA for stealthy gray-box attacks on HGNNs designed for node classification tasks. We have introduced an adversarial training algorithm MeLA-D that constructs a robust model by harnessing adversarial perturbations generated from the Laplacian-based meta-objective. We proved the convergence of MeLA-D and showed that the new attacks MeLA-FGSM and MeLA-PGD are more effective than the baselines, while the defense yields significant robustness gains. Although MeLA-D is an efficient defense, it does not yield a robustness certification. We plan to investigate ways to close this gap in future work.

## Impact Statement

This paper presents work whose goal is to advance the field of Machine learning on hypergraphs and adversarial robustness of HGNNs. There are many potential societal consequences of our work, none of which we feel must be specifically highlighted here.

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

$$R \triangleq S\Delta A\Delta S + \Delta S\Delta AS + \Delta SA\Delta S + \Delta S\Delta A\Delta S. \tag{8}$$

**Step 2: Upper bounding the norms of the perturbations.**
First, using the triangle inequality and sub-multiplicative property of spectral norms, we get

$$\|L_{\mathcal{G}} - L_{\mathcal{G}'}\|_2$$
$$\leq \|S\Delta AS\|_2 + \|\Delta SAS\|_2 + \|SA\Delta S\|_2 + \|R\|_2$$
$$\leq \|S\|_2^2\|\Delta A\|_2 + 2\|\Delta S\|_2\|A\|_2\|S\|_2 + \|R\|_2. \tag{9}$$

Now, we derive the norm bounds on adjacency matrices and $S$'s for the original graph $\mathcal{G}$ and the Adversary-G, i.e.,

$$\|S\|_2 = d_{\min}^{-1/2}, \quad \|A\|_2 \leq d_{\max}, \quad \text{(Lemma 2)}$$
$$\|\Delta A\|_2 \leq \epsilon_{\text{row}}, \quad \text{(Lemma 3)}$$
$$\|\Delta S\|_2 \leq \frac{1}{2}d_{\min}^{-3/2}\epsilon_{\text{row}}. \quad \text{(Lemma 4)}$$

Thus, from Equation (9), we get

$$\|L_{\mathcal{G}} - L_{\mathcal{G}'}\|_2 \leq \frac{\epsilon_{\text{row}}}{d_{\min}} + \frac{d_{\max}\epsilon_{\text{row}}}{d_{\min}^2} + \|R\|_2$$
$$\triangleq \frac{\epsilon_{\text{row}}}{d_{\min}}(1 + \kappa(D)) + \|R\|_2. \tag{10}$$

Now, we bound $\|R\|_2$ (Equation (8)) term-by-term using again the triangle inequality and sub-multiplicative property of spectral norms.

$$\|R\|_2$$
$$\leq \|S\Delta A\Delta S\|_2 + \|\Delta S\Delta AS\|_2 + \|\Delta SA\Delta S\|_2$$
$$\quad + \|\Delta S\Delta A\Delta S\|_2$$
$$\leq \|S\|_2\|\Delta A\|_2\|\Delta S\|_2 + \|\Delta S\|_2\|\Delta A\|_2\|S\|_2$$
$$\quad + \|\Delta S\|_2^2\|A\|_2 + \|\Delta S\|_2^2\|\Delta A\|_2$$
$$= 2\|S\|_2\|\Delta A\|_2\|\Delta S\|_2 + \|A\|_2\|\Delta S\|_2^2 + \|\Delta A\|_2\|\Delta S\|_2^2$$
$$\leq 2(d_{\min}^{-1/2})(\epsilon_{\text{row}})\left(\frac{1}{2}d_{\min}^{-3/2}\epsilon_{\text{row}}\right) + d_{\max}\left(\frac{1}{2}d_{\min}^{-3/2}\epsilon_{\text{row}}\right)^2$$
$$+ (\epsilon_{\text{row}})\left(\frac{1}{2}d_{\min}^{-3/2}\epsilon_{\text{row}}\right)^2$$
$$= \left(\frac{\epsilon_{\text{row}}}{d_{\min}}\right)^2\left(1 + \frac{1}{4}\kappa(D) + \frac{1}{4}\frac{\epsilon_{\text{row}}}{d_{\min}}\right)$$
$$\leq \left(\frac{\epsilon_{\text{row}}}{d_{\min}}\right)^2\left(1 + \frac{1}{2}\kappa(D)\right). \tag{11}$$

The last inequality holds for any $\epsilon_{\text{row}} \leq d_{\max}$.

Now, combining Equation (10) and (11), we obtain

$$\|L_{\mathcal{G}} - L_{\mathcal{G}'}\|_2$$

$$\leq \frac{\epsilon_{\text{row}}}{d_{\min}}(1 + \kappa(D)) + \left(\frac{\epsilon_{\text{row}}}{d_{\min}}\right)^2\left(1 + \frac{1}{2}\kappa(D)\right)$$
$$\leq \left(\frac{\epsilon_{\text{row}}}{d_{\min}} + \left(\frac{\epsilon_{\text{row}}}{d_{\min}}\right)^2\right)(1 + \kappa(D)).$$

This concludes the proof of the upper bound.

**Part b: Proof of the lower bound.**

We derive an existential lower bound through an explicit construction of the graph and the perturbations.

**Step 1: Base graph construction.** Let $n = 4$ and label the vertices $\{1, 2, 3, 4\}$. Fix any $d > 0$ and define the base adjacency matrix $A$ by setting

$$A_{ij} = \begin{cases} d/3, & i \neq j, \\ 0, & i = j. \end{cases}$$

Then $A$ is symmetric, entrywise nonnegative, and has zero diagonal. $A$ can be written as

$$A = \frac{d}{3}(\mathbf{1}\mathbf{1}^\top - I)$$

For each node $i$ the degree is

$$d_i = \sum_{j=1}^4 A_{ij} = \sum_{\substack{j=1 \\ j\neq i}}^4 \frac{d}{3} = 3 \cdot \frac{d}{3} = d.$$

Hence

$$d_{\min} = \min_i d_i = d,$$
$$D = \text{diag}(d_1, d_2, d_3, d_4),$$

with condition number $\kappa(D) = \frac{d}{d} = 1$, and

$$S = D^{-1/2} = d^{-1/2}I.$$

**Step 2: Defining perturbation.** Let us fix

$$t \triangleq \frac{\epsilon_{\text{row}}}{2}.$$

Let us define $\Delta A$ to be supported on the 4-cycle $(1 - 2, 2 - 3, 3 - 4, 4 - 1)$ by $\Delta A_{12} = \Delta A_{21} = t$, $\Delta A_{23} = \Delta A_{32} = t$, $\Delta A_{34} = \Delta A_{43} = t$, $\Delta A_{41} = \Delta A_{14} = t$, and $\Delta A_{ij} = 0$ for all other pairs $(i, j)$, including $\Delta A_{ii} = 0$.

**(i) $\Delta A$ is $\ell_{1,\infty}$ budget satisfying.** For node 1, the only nonzero entries in row 1 are $\Delta A_{12} = t$ and $\Delta A_{14} = t$, so

$$\sum_{j=1}^4 |\Delta A_{1j}| = |t| + |t| = 2|t| = \epsilon_{\text{row}}.$$

Similarly, for node 2, the only nonzero entries are $\Delta A_{21} = t$ and $\Delta A_{23} = t$, so

$$\sum_{j=1}^{4} |\Delta A_{2j}| = |t| + |t| = 2|t| = \epsilon_{\text{row}}.$$

The same computation holds for vertices 3 and 4. Therefore,

$$\|\Delta A\|_{1,\infty} = \max_i \sum_j |\Delta A_{ij}| = \epsilon_{\text{row}}.$$

**(ii)** $\Delta D = \epsilon_{\text{row}} I$. For each $i$, $\Delta D_{ii}$ equals the row sum of $\Delta A$. Hence,

$$D_{ii} = \sum_{j=1}^{4} \Delta A_{ij} = 2t = \epsilon_{\text{row}}.$$

implying, $\Delta D = \epsilon_{\text{row}} I$

**Step 3: Expressing $L_{\mathcal{G}} - L_{\mathcal{G}'}$.** Let us define the perturbed adjacency $A' = A + \Delta A$. Then the perturbed $D'$ is

$$D' = D + \Delta D = (d_{\min} + \epsilon_{\text{row}})I.$$

Therefore

$$S' = (D')^{-1/2} = D^{-1/2} = (d_{\min} + \epsilon_{\text{row}})^{-1/2} I.$$

Hence the Laplacian difference

$$
\begin{aligned}
L_{\mathcal{G}} - L_{\mathcal{G}'} &= \big(I - SAS\big) - \big(I - S'A'S'\big) \\
&= S'A'S' - SAS \\
&= \frac{1}{(d_{\min} + \epsilon_{\text{row}})} A' - \frac{1}{d_{\min}} A \\
&= \left(\frac{1}{(d_{\min} + \epsilon_{\text{row}})} - \frac{1}{d_{\min}}\right) A + \frac{1}{(d_{\min} + \epsilon_{\text{row}})} \Delta A \\
&= \frac{\epsilon_{\text{row}}}{d_{\min}(d_{\min} + \epsilon_{\text{row}})} A + \frac{1}{(d_{\min} + \epsilon_{\text{row}})} \Delta A
\end{aligned}
$$

**Step 4: Lower-bound via quadratic form.** Since $L_G - L_G' = S\Delta A S$ is symmetric, the variational characterization gives

$$
\begin{aligned}
&\|L_{\mathcal{G}} - L_{\mathcal{G}'}\|_2 \\
&= \max_{\|x\|_2=1} \Big| x^\top \big(\frac{\epsilon_{\text{row}}}{d_{\min}(d_{\min} + \epsilon_{\text{row}})} A + \frac{1}{(d_{\min} + \epsilon_{\text{row}})} \Delta A\big) x \Big| \\
&\geq \Big| \frac{\epsilon_{\text{row}}}{d_{\min}(d_{\min} + \epsilon_{\text{row}})} x^\top A x + \frac{1}{(d_{\min} + \epsilon_{\text{row}})} x^\top \Delta A x \Big| \\
&= \Big| \alpha \, x^\top A x + \beta \, x^\top \Delta A x \Big|,
\end{aligned}
$$

where

$$\alpha \triangleq \frac{\epsilon_{\text{row}}}{d_{\min}(d_{\min} + \epsilon_{\text{row}})},$$

$$\beta \triangleq \frac{1}{d_{\min} + \epsilon_{\text{row}}}.$$

We now choose a witness vector

$$x \triangleq \frac{1}{2} \begin{bmatrix} 1 \\ 1 \\ 1 \\ 1 \end{bmatrix},$$

which is a unit vector, and evaluate the quadratic forms $x^\top A x$ and $x^\top \Delta A x$.

*Step 4a: Evaluating $x^\top A x$.*

$$
\begin{aligned}
x^\top A x &= \frac{d}{3}(x^\top \mathbf{1}\mathbf{1}^\top x - x^\top I x) \\
&= \frac{d}{3}((x^\top \mathbf{1})(\mathbf{1}^\top x) - x^\top x) \\

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

and

$$ab \triangleq \frac{\epsilon_{\mathrm{row}}}{2c} = \frac{\epsilon_{\mathrm{col}}}{2r}. \tag{27}$$

Then

$$\|\Delta\boldsymbol{H}\|_\infty = 2c\, ab = \epsilon_{\mathrm{row}}, \qquad \|\Delta\boldsymbol{H}\|_1 = 2r\, ab = \epsilon_{\mathrm{col}},$$

so the budgets hold with equality.

Now we compute $\frac{ab}{h}$ in terms of $d_{\min}, \delta_{\min}$. Recall $d_{\min} = ch$ and $\delta_{\min} = rh$, so

$$\sqrt{d_{\min}\delta_{\min}} = \sqrt{(ch)(rh)} = h\sqrt{cr}.$$

Using (27) and (26),

$$ab\sqrt{cr} = \left(\frac{\epsilon_{\mathrm{row}}}{2c}\right)\sqrt{cr} = \frac{\epsilon_{\mathrm{row}}}{2}\sqrt{\frac{r}{c}} = \frac{\epsilon_{\mathrm{row}}}{2}\sqrt{\frac{\epsilon_{\mathrm{col}}}{\epsilon_{\mathrm{row}}}}$$

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

$$\|\Delta\boldsymbol{A}\|_2 \leq \frac{2}{\delta_{\min}} \|\boldsymbol{H}\|_2 \sqrt{\epsilon_{\mathrm{col}}\epsilon_{\mathrm{row}}} + \frac{1}{\delta_{\min}^2} \|\boldsymbol{H}\|_2^2 \epsilon_{\mathrm{col}} + \frac{\epsilon_{\mathrm{col}}\epsilon_{\mathrm{row}}}{\delta_{\min}}.$$

*Proof.*

$$\Delta\boldsymbol{A} = (\boldsymbol{H} + \Delta\boldsymbol{H})(\boldsymbol{D}_e')^{-1}(\boldsymbol{H} + \Delta\boldsymbol{H})^\top - \boldsymbol{H}\boldsymbol{D}_e^{-1}\boldsymbol{H}^\top$$
$$= \boldsymbol{H}((\boldsymbol{D}_e')^{-1} - (\boldsymbol{D}_e)^{-1})\boldsymbol{H}^\top + \Delta\boldsymbol{H}(\boldsymbol{D}_e')^{-1}\boldsymbol{H}^\top +$$
$$\boldsymbol{H}(\boldsymbol{D}_e')^{-1}\Delta\boldsymbol{H}^\top + \Delta\boldsymbol{H}(\boldsymbol{D}_e')^{-1}\Delta\boldsymbol{H}^\top$$

$$\|\Delta\boldsymbol{A}\|_2 \leq \|\boldsymbol{H}((\boldsymbol{D}_e')^{-1} - (\boldsymbol{D}_e)^{-1})\boldsymbol{H}^\top\|_2 + \|\Delta\boldsymbol{H}(\boldsymbol{D}_e')^{-1}\boldsymbol{H}^\top\|_2 +$$
$$\|\boldsymbol{H}(\boldsymbol{D}_e')^{-1}\Delta\boldsymbol{H}^\top\|_2 + \|\Delta\boldsymbol{H}(\boldsymbol{D}_e')^{-1}\Delta\boldsymbol{H}^\top\|_2 \qquad (30)$$

**(1) The first term** yields

$$\|\boldsymbol{H}((\boldsymbol{D}_e')^{-1} - (\boldsymbol{D}_e)^{-1})\boldsymbol{H}^\top\|_2 \leq \|\boldsymbol{H}\|_2^2 \|(\boldsymbol{D}_e')^{-1} - (\boldsymbol{D}_e)^{-1}\|_2 \qquad (31)$$

Using the identity

$$(\boldsymbol{D}_e')^{-1} - \boldsymbol{D}_e^{-1} = \boldsymbol{D}_e^{-1}(\boldsymbol{D}_e - \boldsymbol{D}_e')(\boldsymbol{D}_e')^{-1}$$
$$= -\boldsymbol{D}_e^{-1}(\Delta\boldsymbol{D}_e)(\boldsymbol{D}_e')^{-1},$$

we obtain

$$\|(\boldsymbol{D}_e')^{-1} - \boldsymbol{D}_e^{-1}\|_2 \leq \|\boldsymbol{D}_e^{-1}\|_2 \|\Delta\boldsymbol{D}_e\|_2 \|(\boldsymbol{D}_e')^{-1}\|_2.$$

Since $\boldsymbol{D}_e^{-1}$ is diagonal and $\delta_{\min} = \min_e \delta_e > 0$, we have

$$\|\boldsymbol{D}_e^{-1}\|_2 = \max_e \delta_e^{-1} = \delta_{\min}^{-1}.$$

Under $\delta_{\min}' \geq \delta_{\min}$, we have

$$\|(\boldsymbol{D}_e')^{-1}\|_2 = \max_e (\delta_e')^{-1} = (\delta_{\min}')^{-1} \leq \frac{1}{\delta_{\min}}.$$

Combining gives

$$\|(\boldsymbol{D}_e')^{-1} - \boldsymbol{D}_e^{-1}\|_2 \leq \delta_{\min}^{-1} \|\Delta\boldsymbol{D}_e\|_2 \frac{1}{\delta_{\min}}$$
$$= \frac{1}{\delta_{\min}^2} \|\Delta\boldsymbol{D}_e\|_2.$$

**2. The summation of the second and third term**

$$\left\|\Delta\boldsymbol{H}(\boldsymbol{D}_e')^{-1}\boldsymbol{H}^\top\right\|_2 + \left\|\boldsymbol{H}(\boldsymbol{D}_e')^{-1}(\Delta\boldsymbol{H})^\top\right\|_2$$

$$\leq 2\|\boldsymbol{D}_e'\|_2^{-1} \|\boldsymbol{H}\|_2 \|\Delta\boldsymbol{H}\|_2$$
$$\leq \frac{2}{\delta_{\min}} \|\boldsymbol{H}\|_2 \|\Delta\boldsymbol{H}\|_2, \qquad (32)$$

**3. The last term** reduces to

$$\left\|\Delta\boldsymbol{H}(\boldsymbol{D}_e')^{-1}\Delta\boldsymbol{H}^\top\right\|_2 \leq \|\Delta\boldsymbol{H}\|_2^2 \|(\boldsymbol{D}_e')^{-1}\|_2 = \frac{1}{\delta_{\min}} \|\Delta\boldsymbol{H}\|_2^2 \qquad (33)$$

Thus equation (30) reduces to

$$\|\Delta\boldsymbol{A}\|_2 \leq \frac{1}{\delta_{\min}^2} \|\boldsymbol{H}\|_2^2 \|\Delta\boldsymbol{D}_e\|_2 + \frac{2}{\delta_{\min}} \|\boldsymbol{H}\|_2 \|\Delta\boldsymbol{H}\|_2 +$$
$$\frac{1}{\delta_{\min}} \|\Delta\boldsymbol{H}\|_2^2 \qquad (34)$$

**Bound for $\|\Delta\boldsymbol{D}_e\|_2$.** By definition,

$$\delta_e = \sum_{v=1}^{n} H_{ve},$$

$$\delta_e' = \sum_{v=1}^{n} H_{ve}' = \sum_{v=1}^{n}(H_{ve} + \Delta H_{ve}) = \delta_e + \sum_{v=1}^{n}\Delta H_{ve}.$$

Thus

$$(\Delta\boldsymbol{D}_e)_{ee} = \delta_e' - \delta_e = \sum_{v=1}^{n}\Delta H_{ve}.$$

Since $\Delta\boldsymbol{D}_e$ is diagonal,

$$\|\Delta\boldsymbol{D}_e\|_2 = \max_e |(\Delta\boldsymbol{D}_e)_{ee}| = \max_e \left|\sum_{v=1}^{n}\Delta H_{ve}\right|.$$

By the triangle inequality,

$$\left|\sum_{v=1}^{n}\Delta H_{ve}\right| \leq \sum_{v=1}^{n}|\Delta H_{ve}|.$$

Taking the maximum over $e$ yields

$$\|\Delta\boldsymbol{D}_e\|_2 \leq \max_e \sum_{v=1}^{n}|\Delta H_{ve}| = \|\Delta\boldsymbol{H}\|_1 \leq \epsilon_{\mathrm{col}}.$$

**Bound for $\|\Delta\boldsymbol{H}\|_2$.** Using the standard inequality valid for any real matrix,

$$\|\boldsymbol{M}\|_2 \leq \sqrt{\|\boldsymbol{M}\|_1 \|\boldsymbol{M}\|_\infty},$$

we obtain

$$\|\Delta\boldsymbol{H}\|_2 \leq \sqrt{\|\Delta\boldsymbol{H}\|_1 \|\Delta\boldsymbol{H}\|_\infty} \leq \sqrt{\epsilon_{\mathrm{col}}\epsilon_{\mathrm{row}}}.$$

Using the bound for $\|\Delta\boldsymbol{H}\|_2$ and $\|\Delta\boldsymbol{D}_e\|_2$ equation (34) reduces to

$$\|\Delta\boldsymbol{A}\|_2 \leq \frac{1}{\delta_{\min}^2} \|\boldsymbol{H}\|_2^2 \|\Delta\boldsymbol{D}_e\|_2 + \frac{2}{\delta_{\min}} \|\boldsymbol{H}\|_2 \|\Delta\boldsymbol{H}\|_2 +$$
$$\frac{1}{\delta_{\min}} \|\Delta\boldsymbol{H}\|_2^2$$
$$\leq \frac{\epsilon_{\mathrm{col}}}{\delta_{\min}^2} \|\boldsymbol{H}\|_2^2 + \frac{2\sqrt{\epsilon_{\mathrm{col}}\epsilon_{\mathrm{row}}}}{\delta_{\min}} \|\boldsymbol{H}\|_2 + \frac{\epsilon_{\mathrm{col}}\epsilon_{\mathrm{row}}}{\delta_{\min}}$$

which completes the proof. $\qquad\square$

## A.4. Convergence Analysis of `MeLA-D`

**Assumption 1** (Lipschitzness of Loss)**.** *Let us denote* $\mathcal{H} \triangleq (\boldsymbol{H}, \boldsymbol{X})$. *The gradients of the loss function* $\mathcal{L}_{\text{train}}(\phi, \mathcal{H}) \triangleq \mathcal{L}_{\text{train}}(g_\phi(\boldsymbol{H}, \boldsymbol{X}), \boldsymbol{y}_L)$ *satisfies three Lipschitzness conditions w.r.t.* $\ell_2$-*norm:*

1. $\nabla_\phi \mathcal{L}_{\text{train}}(\phi, \mathcal{H})$ *is* $L_{\phi\theta}$-*Lipschitz w.r.t.* $\phi$ *and* $L_{\phi\mathcal{H}}$-*Lipschitz w.r.t.* $\mathcal{H}$,

2. $\nabla_\mathcal{H} \mathcal{L}_{\text{train}}(\phi, \mathcal{H}) \qquad\qquad \triangleq$
   $(\nabla_{\boldsymbol{H}} \mathcal{L}_{\text{train}}(\phi, \mathcal{H}), \nabla_{\boldsymbol{X}} \mathcal{L}_{\text{train}}(\phi, \mathcal{H}))$ *is* $L_{\mathcal{H}\phi}$-*Lipschitz w.r.t.* $\mathcal{H}$.

Assumption 1 states the Lipschitz continuity of the loss gradient with respect to both the target model parameters $\phi$ and the hypergraph structure $\mathcal{H} = (\boldsymbol{H}, \boldsymbol{X})$. For a 1-layer HGNN (Feng et al., 2019) $g_\phi$, such as the formulation based on Zhou's normalized hypergraph Laplacian, the output can be written as

$$g_\phi(\boldsymbol{H}, \boldsymbol{X}) = \sigma\left(\boldsymbol{D}_v^{-1/2} \boldsymbol{H} \boldsymbol{D}_e^{-1} \boldsymbol{H}^\top \boldsymbol{D}_v^{-1/2} \boldsymbol{X} \phi\right),$$

where $\sigma$ is a Lipschitz activation function (e.g., ReLU), and $\phi$ is a learnable parameter. Each operation in this formulation–matrix multiplications, degree normalization, and activation is Lipschitz continuous under standard norms. Thus, the gradient $\nabla_\phi \mathcal{L}_{\text{train}}(\phi, \mathcal{H})$ is Lipschitz with respect to $\phi$. In addition, under smoothness assumptions on the loss (e.g., cross-entropy), it is also Lipschitz in $\boldsymbol{H}$. While the incidence matrix $\boldsymbol{H}$ is typically binary and discrete, we have employed its continuous relaxations during `MeLA-D`– this allows for smooth differentiability and thereby satisfies the Lipschitz condition. Therefore, the gradient components in Assumption 1 can be expected to be Lipschitz under such conditions, making the assumption reasonable for a 1-layer hypergraph convolutional network.

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

**Attacker's constraints.** To ensure the attack remains realistic and stealthy, it is generally advisable that attacks satisfy the unnoticeability constraints. Based on the traditional wisdom from the GNN attack literature (Zügner et al., 2018; Zügner & Günnemann, 2019), perturbations are not noticeable when (i) there is an upper bound on the frequency of structural and feature changes, and (ii) statistics such as degree distribution are preserved as much as possible. Finally, due to the nature of hypergraphs, it is possible for an attacker to consider other types of stealthiness measures, such as hyperedge cardinality preservation. Although we did not consider it in our paper, MeLA can accommodate such measures by design. In mathematical terms, there are three constraints on the attacker:

**I. Structural perturbation budget:** The number of modifications to the hypergraph structure is limited by

$$\|\Delta\boldsymbol{H}\|_0 \leq \delta_H,$$

where $\delta_H$ is a predefined budget.

**II. Feature perturbation bound:** The perturbation to node features is bounded in $l_\infty$-norm:

$$\|\Delta\boldsymbol{X}\|_\infty \leq \delta_X,$$

where $\delta_X$ controls the maximum allowable change per feature.

**III. Stealthiness constraints:** To maintain the overall structure and distribution of the hypergraph, the degree distribution of nodes should not deviate significantly.

## B.2. Attacker's Goal

The attacker's goal is not to misclassify any pre-defined target node(s), but rather to cause an overall drop in a target model's performance on the node classification task. To be precise, it wants to decrease the accuracy of a node classification algorithm after training on the perturbed hypergraph. In other words, it wants to have the test nodes classified as a class different from the true class. Specifically, the attacker seeks to find perturbations $\Delta\boldsymbol{H}$ and $\Delta\boldsymbol{X}$ such that the trained hypergraph neural network $f_{\theta^*}$ misclassifies nodes in $V_U$. Formally, the attack objective is to maximise the classification loss on unlabeled nodes:

$$\max_{\Delta\boldsymbol{X},\Delta\boldsymbol{H}}\mathcal{L}_{atk}(f_{\theta^*}(\boldsymbol{H} + \Delta\boldsymbol{H}, \boldsymbol{X} + \Delta\boldsymbol{X}), y_U) \quad \textbf{s.t.}$$

$$\theta^* = \arg\min_{\theta} \mathcal{L}_{tr}(f_\theta(\boldsymbol{H} + \Delta\boldsymbol{H}, \boldsymbol{X} + \Delta\boldsymbol{X}), y_L)$$

and the unnoticeability constraints I-III. Here, $\mathcal{L}_{atk}$ is the loss function that the attacker wants to maximise. This formulation represents a bilevel optimisation problem, where the inner optimisation corresponds to training the target Hypergraph Neural Network on the perturbed data, and the outer optimisation aims to maximise the attacker's loss on unlabeled nodes.

In our case, the attacker wants to worsen the generalisation ability of $f_{\theta^*}$ on the unlabeled nodes. One way to achieve this is to use $\mathcal{L}_{atk} = \mathcal{L}_{tr} = \mathcal{L}_{CE}(\boldsymbol{y}_L, f_{\theta^*}(V_L))$ where CE indicates cross-entropy loss, and by $f_{\theta^*}(V_L)$ we mean the evaluation of trained model on labeled nodes $V_L$. The reason for using labeled node is not because it is realistic, but rather it serves as a worst-case baseline, since a model with a bad training accuracy would most likely have a worse test accuracy.

Let us suppose the attacker obtained its optimal perturbations $(\Delta\boldsymbol{H}^*, \Delta\boldsymbol{X}^*)$ and constructs the perturbed hypergraph structure $\boldsymbol{H} + \Delta\boldsymbol{H}^*$ and feature matrix $\boldsymbol{X} + \Delta\boldsymbol{

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

 4.* Impact of the `MeLA`-PGD on the node embedding shift. A higher shift suggests a more effective attack.

*Table 8.* Comparison of adversarial test accuracy (± std) and median drop (%) under poisoning attacks. Higher median drop indicates more effective attacks.

| Attack (Surrogate) | Citeseer | Cora | Cora-CA |
|---|---|---|---|
| `MeLA`-PGD (MLP) | 66.71 ± 0.98 / 10.28 | 70.46 ± 1.72 / 11.11 | 77.93 ± 0.69 / 6.21 |
| `MeLA`-PGD (LinHGNN) | 54.37 ± 1.17 / **26.5** | 55.42 ± 0.57 / **29.5** | 72.64 ± 0.38 / **12.4** |
| `MeLA`-FGSM (MLP) | 54.30 ± 0.83 / **27.36** | 61.54 ± 0.65 / **22.03** | 76.72 ± 0.39 / 7.49 |
| `MeLA`-FGSM (LinHGNN) | 69.37 ± 0.14 / 6.4 | 67.62 ± 0.49 / 14.3 | 74.15 ± 0.68 / **11.1** |

Figure 4 shows that `MeLA`-PGD and `MeLA`-FGSM produce the largest shift in node embeddings among all the attack methods, which is indicative of their effectiveness as an attack.

### F.3. `MeLA`-PGD on a Toy Example

We generate a toy hypergraph with 12 nodes and 6 hyperedges to visualize how `MeLA`-PGD changes the decision boundary of HGNN (Feng et al., 2019). The node features are coordinates in $\mathbb{R}^2$. The nodes with $+x$ coordinates belong to the blue class while the nodes with $-x$ coordinates

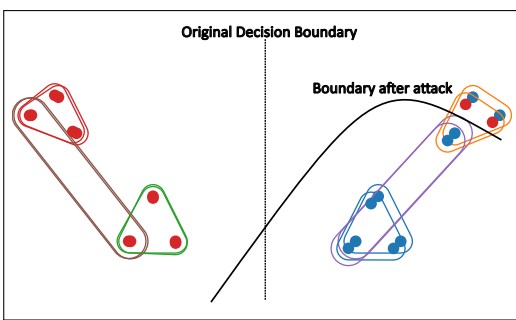

*Figure 5.* Effect on HGNN's decision boundary on a toy hypergraph due to attack `MeLA`-PGD.

belong to the red class. The dotted line indicates $x = 0$ line that separates nodes from the two classes is the initial decision boundary of HGNN.

Figure 5 shows that two blue classes have been misclassified after the attack. This shift suggests that the decision boundary has been affected by attack `MeLA`-PGD.

*Table 9.* Adversarial training of HGNN (evasion attack).

| Attack | Citeseer | | Cora-CA | | Cora | |
|---|---|---|---|---|---|---|
| | HGNN | MeLA-D+HGNN / Gain | HGNN | MeLA-D+HGNN / Gain | HGNN | MeLA-D+HGNN / Gain |
| *Clean* | $71.78 \pm 0.65$ | $71.78 \pm 0.65$ / 0 | $78.33 \pm 0.89$ | $78.33 \pm 0.89$ / 0 | $82.62 \pm 1.38$ | $82.62 \pm 1.38$ / 0 |
| RandFeat | $65.92 \pm 1.46$ | $71.81 \pm 0.61$ / 5.6 | $72.76 \pm 0.64$ | $77.73 \pm 0.89$ / 4.7 | $80.38 \pm 1.60$ | $82.66 \pm 1.40$ / 1.9 |
| RandFlip | $65.68 \pm 1.60$ | $69.23 \pm 0.58$ / 4.1 | $72.53 \pm 1.04$ | $74.59 \pm 0.92$ / 1.5 | $79.00 \pm 0.82$ | $80.71 \pm 0.81$ / 1.6 |
| GradArgMax | $69.30 \pm 1.53$ | $71.71 \pm 1.40$ / 2.9 | $71.49 \pm 2.43$ | $72.94 \pm 2.61$ / 1.5 | $79.56 \pm 1.50$ | $80.59 \pm 1.69$ / 1.2 |
| MeLA-PGD | $23.96 \pm 2.31$ | $63.94 \pm 1.03$ / **38.9** | $28.21 \pm 2.09$ | $72.32 \pm 1.51$ / **44.0** | $13.65 \pm 0.88$ | $75.04 \pm 1.55$ / **60.6** |
| MeLA-FGSM | $50.07 \pm 2.40$ | $76.01 \pm 1.32$ / 25.7 | $42.48 \pm 1.53$ | $79.97 \pm 1.51$ / 36.2 | $20.06 \pm 1.73$ | $78.29 \pm 1.95$ / 58.5 |

| Datasets | $|V|$ | $|E|$ | avg. edge size | # features | # classes |
|---|---|---|---|---|---|
| **DBLP-CA** | 41302 | 22363 | 4.45 | 1425 | 6 |

*Table 10.* Statistics of additional datasets.

*Table 11.* Attack effectiveness of mini-batch MeLA-PGD on HGNN for DBLP-CA dataset (Batch size=1024).

| Clean | Evasion / Drop (E) | Execution time (sec) |
|---|---|---|
| $90.56 \pm 0.23$ | $88.84 \pm 1.44$ / 1.6% | $140.59 \pm 5.12$ |

# G. Additional Experimental Analysis

In this section, we present and discuss some additional experimental results.

## G.1. Attack Effectiveness on Large-scale Datasets

In order to adapt our attack to large-scale hypergraphs such as DBLP-CA (Table 10), we have adapted our attack to process mini-batches.

**Mini-batch MeLA-PGD:** We divide the input hypergraph into random subhypergraphs induced by $b << |V|$ nodes sampled from $V$ where $b$ is the batch-size (chosen to be 1024 for DBLP-CA). Each mini-batch represents a block sub-matrix $(\boldsymbol{H}_{IJ}, \boldsymbol{X}_{IJ})$ of the input incidence matrix $\boldsymbol{H}$, and $\boldsymbol{X}$ respectively with $|I| = b$. Instead of computing $\Delta \boldsymbol{X}, \Delta \boldsymbol{H}$ for the full matrix $(\boldsymbol{X}, \boldsymbol{H})$, we compute local perturbations $\Delta \boldsymbol{X}_{IJ}, \Delta \boldsymbol{H}_{IJ}$ for the block matrices $\boldsymbol{H}_{IJ}, \boldsymbol{X}_{IJ}$. Essentially, lines 5-16 in Algorithm 1 run for each mini-batch, and after computing each mini-batch perturbations, the full matrix $\Delta \boldsymbol{H}, \Delta \boldsymbol{X}$ is updated using the mini-batches $\Delta \boldsymbol{X}_{IJ}, \Delta \boldsymbol{H}_{IJ}$.

Table 11 shows the accuracy and execution time of mini-batch MeLA-PGD on the DBLP-CA dataset in the evasion setting. We ran HGNN with 1 hidden layer with a dimension of 64 for 500 epochs with patience of 5. We set the following hyperparameters for MeLA-PGD on DBLP-CA: $T = 20, \tau_s = 20$. We find that the minibatch-MeLA-PGD reduces the accuracy by 1.6%. The attack algorithm took $\sim 140$ seconds.

## G.2. Statistical Significance of Results

We now turn to testing whether there is a (statistically) significant difference in performance (i.e., accuracy drop) across multiple attack types evaluated on the same set of seeds and datasets. Our Null hypothesis is "All attack types have equivalent performance", whereas the Alternative hypothesis is "At least one attack performs differently." We employ a non-parametric Friedman test (Demšar, 2006), which is a preferred procedure in formal statistical inference due to its minimal assumptions about the underlying data. Table 12 shows the test result.

In Tables 12-14, we notice that across various baseline HGNNs and in both poisoning/evasion settings, the p-value implies a statistically significant result ($p$-value $< 0.05$), so we reject the stated null hypothesis. Finally, we observe that the proposed MeLA-PGD and MeLA-FGSM consistently rank 1 among all the attacks. This, along with us rejecting the null hypothesis, suggests that our proposed attacks lead to a statistically significant performance degradation of the HGNNs.

# H. Hyperparameter sensitivity

## H.1. Sensitivity to Structural and Feature Budgets $\delta_H, \delta_X$ on MeLA

We analyse the impact of perturbation budget on attack effectiveness by employing MeLA-PGD to attack HGNN on all datasets.

First, we plot $\delta_H$ vs. Accuracy drop while keeping feature perturbation budget $\delta_X$ fixed at 0.05 in Figure 6 (left). We observe that as $\delta_H$ increases, so does the drop in classification accuracy.

We also plot $\delta_X$ vs. Accuracy drop while keeping structural perturbation budget $\delta_H$ fixed at 20% in Figure 6 (right). We observe that increasing $\delta_X$ from 0 to 0.05, then to 1.0, results in a consistent increase in the accuracy drop. This observation resembles the case for GNN attacks, where attack effectiveness also positively correlates with the attacker's budget.

*Table 12.* Statistical ranking of attacks using a non-parametric Friedman test (Citeseer dataset).

| Attack | MLP | | AllSetTransformer | | HGNN | |
|---|---|---|---|---|---|---|
| | Rank (Poison) | Rank (Evasion) | Rank (Poison) | Rank (Evasion) | Rank (Poison) | Rank (Evasion) |
| | Test stat: 19.4 | Test stat: 20.0 | Test stat: 19.0 | Test stat: 19.1 | Test stat: 19.2 | Test stat: 18.1 |
| | p-value: $6.7e^{-4}$ | p-value: $5.0e^{-4}$ | p-value: $7.7e^{-4}$ | p-value: $7.5e^{-4}$ | p-value: $7.2e^{-4}$ | p-value: $1.2e^{-3}$ |
| RandFeat | 4.0 | 3.0 | 4.6 | 3.1 | 5.0 | 4.2 |
| RandFlip | 2.2 | 4.5 | 3.0 | 4.1 | 2.5 | 3.2 |
| GradArgMax | 5.0 | 4.5 | 4.4 | 4.8 | 4.0 | 4.6 |
| MeLA-PGD | **1.0** | **1.0** | **1.0** | **1.0** | **1.0** | **1.0** |
| MeLA-FGSM | 2.8 | 2.0 | 2.0 | 2.0 | 2.5 | 2.0 |

*Table 13.* Statistical ranking of attacks using a non-parametric Friedman test (Cora dataset).

| Attack | MLP | | AllSetTransformer | | HGNN | |
|---|---|---|---|---|---|---|
| | Rank (Poison) | Rank (Evasion) | Rank (Poison) | Rank (Evasion) | Rank (Poison) | Rank (Evasion) |
| | Test stat: 19.4 | Test stat: 20.0 | Test stat: 16.5 | Test stat: 17.1 | Test stat: 20.0 | Test stat: 19.4 |
| | p-value: $6.7e^{-4}$ | p-value: $5.0e^{-4}$ | p-value: $2.4e^{-3}$ | p-value: $1.8e^{-3}$ | p-value: $5.0e^{-4}$ | p-value: $6.7e^{-4}$ |
| RandFeat | 4.2 | 3.0 | 4.4 | 4.4 | 5.0 | 5.0 |
| RandFlip | 3.0 | 4.5 | 3.4 | 3.4 | 3.0 | 3.2 |
| GradArgMax | 4.8 | 4.5 | 4.2 | 4.2 | 4.0 | 3.8 |
| MeLA-PGD | **1.0** | **1.0** | **1.2** | **1.0** | **1.0** | **1.0** |
| MeLA-FGSM | 2.0 | 2.0 | 1.8 | 2.0 | 2.0 | 2.0 |

*Table 14.* Statistical ranking of attacks using a non-parametric Friedman test (Cora-CA dataset).

| Attack | HyperMLP | | AllSetTrans. | | HGNN | |
|---|---|---|---|---|---|---|
| | Rank (Poison) | Rank (Evasion) | Rank (Poison) | Rank (Evasion) | Rank (Poison) | Rank (Evasion) |
| | Test stat: 17.7 | Test stat: 20.0 | Test stat: 18.5 | Test stat: 17.2 | Test stat: 19.4 | Test stat: 19.1 |
| | p-value: $1.4e^{-3}$ | p-value: $5.0e^{-4}$ | p-value: $1.0e^{-3}$ | p-value: $1.8e^{-3}$ | p-value: $6.7e^{-4}$ | p-value: $7.6e^{-4}$ |
| RandFeat | 4.1 | 3.0 | 4.7 | 4.0 | 5.0 | 4.9 |
| RandFlip | 1.6 | 4.5 | 4.1 | 4.5 | 3.3 | 3.8 |
| GradArgMax | 4.9 | 4.5 | 3.2 | 3.5 | 3.7 | 3.3 |
| MeLA-PGD | **1.6** | **1.0** | **1.0** | **1.0** | **1.0** | **1.0** |
| MeLA-FGSM | 2.8 | 2.0 | 2.0 | 2.0 | 2.0 | 2.0 |

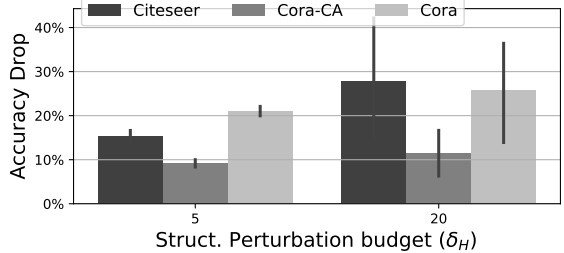
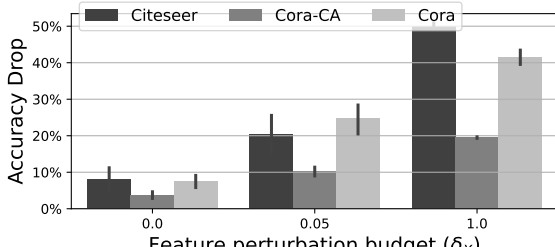

*Figure 6.* Change in accuracy with varying structural Perturbation budget (left) and feature perturbation budget (right).

## I. Choice of Hyperparameters

All HGNNs for attacks are trained for 1000 epochs with an early stopping value of 150 based on validation loss. All HGNNs use Adam optimizer with learning rates and weight decay prescribed in Table 15-17. All attacks use a budget of $\delta_X = 0.05$ and $\delta_H = 0.2$ (i.e., 20% of $|E|$). Tables 15 - 17 show the choices for various hyperparameters for reproducibility. In the tables, **Lr** indicates learning rate, **Wd** indicates weight decay, **Drop.** indicates drop-out rate. Our source codes are available at https://anonymous.4open.science/r/mela/.

Our adversarial training using MeLA-D uses the same parameters as attacks for various models, with a few exceptions we discuss in Tables 18-19.

*Table 15.* Choice of Hyperparameters for HyperMLP.

| | Lr | Wd | Hidden dim. | #Layers | HyperMLP-$\alpha$ | Drop. | $\alpha, \beta, \gamma$ | T | $\tau_s/\tau_a$ | $\eta_H$ | $\eta_X$ |
|---|---|---|---|---|---|---|---|---|---|---|---|
| **Citeseer,Cora-CA,Cora** | 0.001 | 0.0 | 512 | 2 | 0.005 | 0.1 | 1,1,4 | 80 | 80 | 0.01 | 0.01 |

*Table 16.* Choice of Hyperparameters for HGNN.

| | Lr | Wd | Hidden dim. | #Layers | Drop. | $\alpha, \beta, \gamma$ | T | $\tau_s/\tau_a$ | $\eta_H$ | $\eta_X$ |
|---|---|---|---|---|---|---|---|---|---|---|
| **Citeseer,Cora-CA,Cora** | 0.001 | 0.0 | 512 | 1 | 0.5 | 1,1,4 | 80 | 80 | 0.01 | 0.01 |

*Table 17.* Choice of Hyperparameters for AllSetTransformer.

| | Lr | Wd | Hidden dim. | #MLP layers | #Heads (AllSetTrans.) | Drop. | $\alpha, \beta, \gamma$ | T | $\tau_s/\tau_a$ | $\eta_H$ | $\eta_X$ |
|---|---|---|---|---|---|---|---|---|---|---|---|
| **Citeseer,Cora-CA,Cora** | 0.001 | 0.0 | 256 | 2 | 4 | 0.5 | 1,1,4 | 80 | 80 | 0.01 | 0.01 |

*Table 18.* Hyperparameters for `MeLA`-D+HyperMLP. The rest of the hyperparameters follow those for HyperMLP.

| Dataset | Attack | Lr | T | $\tau_d$ | Epochs |
|---|---|---|---|---|---|
| Cora-CA | RandFeat | 0.0005 | 15 | 200 | 100 |
| | RandFlip | 0.0001 | 15 | 200 | 100 |
| | GradArgMax | 0.0001 | 12 | 200 | 100 |
| | `MeLA`-D, `MeLA`-FGSM | 0.0001 | 15 | 400 | 400 |
| Cora | RandFeat | 0.001 | 15 | 400 | 400 |
| | RandFlip | 0.0001 | 15 | 200 | 100 |
| | GradArgMax | 0.0001 | 12 | 200 | 100 |
| | `MeLA`-D, `MeLA`-FGSM | 0.001 | 15 | 400 | 400 |
| Citeseer | RandFeat | 0.003 | 10 | 400 | 400 |
| | RandFlip | 0.001 | 15 | 180 | 100 |
| | GradArgMax | 0.001 | 12 | 200 | 100 |
| | `MeLA`-D, `MeLA`-FGSM | 0.001 | 15 | 400 | 400 |

*Table 19.* Hyperparameters for `MeLA`-D+HGNN. The rest of the hyperparameters follow those for HGNN.

| Dataset | Attack | Lr | T | $\tau_d$ | Epochs |
|---|---|---|---|---|---|
| All datsets | All attacks | 0.001 | 20 | 300 | 200 |