# OpenReview forum: "Adversarial Attacks and Robust Training for Hypergraph Neural Networks"
_ICML.cc/2026/Conference — ICML 2026 regular_

### Official Review · Reviewer_wjdw · 2026-02-19

**Soundness:** 3
**Presentation:** 3
**Significance:** 3
**Originality:** 3
**Overall Recommendation:** 4
**Confidence:** 5

**Summary:**

The manuscript explores adversarial robustness in hypergraph neural networks. The authors design a gray-box attack that modifies hypergraph connections and node features, guided by a meta-objective. They further develop an adversarial training approach to mitigate the damage. Experiments on benchmark datasets validate their findings.

**Compliance With Llm Reviewing Policy:**

Affirmed.

**Key Questions For Authors:**

1.	The manuscript adopts adversarial training as the main defense strategy. However, prior work [1] has pointed out a fundamental flaw in this approach, showing that it may cause the model to learn incorrect semantic–label associations. The issue does not stem from imperfect optimization, but from the fact that structural perturbations violate the core assumption of adversarial training, namely label-invariant semantics. The authors may consider incorporating the insights of [1] to improve their defense design.
2.	The proposed attack and defense frameworks appear rather complex, particularly the defense strategy. The manuscript would benefit from a detailed computational complexity analysis. This would help clarify the practical feasibility of the method, especially for larger hypergraphs.
3.	The proposed method should be evaluated on larger-scale datasets, such as Ogbn and Flickr, to better demonstrate its scalability and robustness. The current experiments are limited to relatively small benchmarks, which makes it difficult to assess performance in more realistic settings.
4.	The proposed defense should be compared with mainstream GNN defense methods from the past three years. Many of these methods are general enough to be extended to hypergraph settings. Including such comparisons would provide a clearer picture of the relative effectiveness of the proposed method.
5.	The convergence analysis is built upon standard assumptions such as local concavity, Lipschitz continuity, and bounded variance of stochastic gradients. However, the paper does not examine whether HGNN training satisfies these conditions. Since HGNN optimization is inherently nonconvex, the realism of these assumptions is questionable. Consequently, the practical significance of the convergence result remains uncertain.

[1] Boosting the Adversarial Robustness of Graph Neural Networks: An OOD Perspective

**Limitations:**

The manuscript currently lacks a sufficient discussion of limitations and broader impacts. The authors are encouraged to elaborate on potential weaknesses and possible negative consequences of the proposed approach.

**Strengths And Weaknesses:**

The manuscript is technically sound and competently presented, but its novelty and impact may be somewhat incremental. Further strengthening of experiments or positioning would enhance its overall contribution.

---

> ### Author Rebuttal · Authors · 2026-03-31
>
> Thank you for your valuable comments on adversarial training and experiments.
>
> > Q1. Adversarial training
>
> Thank you for pointing to this interesting work. A key difference between Good-AT [1] and MeLA-D is that Good-AT does adversarial defence by detecting OOD samples with the goal of inference with correct semantic-label association, whereas MeLA-D adversarially trains the target HGNN itself without purifying the input hypergraph. A direct adaptation of Good-AT for hypergraph would be to construct a detector of adversarial perturbations, and then to purify the hypergraph during inference yielding a different defense method than MeLA-D. During inference, one can recalibrate the robustified model trained via MeLA-D using the purified hypergraph yielded by Good-AT. We will discuss this enhancement as a future work.
>
> > Q2. Computational Complexity
>
> Let $n$ be the \# nodes, $m$ the \#edges, $d$ the feature dimension, and $E= ||H||\_0$ the number of nonzero entries in the incidence matrix. Let $\tau\_s$ be the number of surrogate training epochs, $T$ be the number of attack steps, and $\delta_H$ be the perturbation budget. Let $C_{\text{fwd}}$ and $C_{\text{bwd}}$ be the cost of one forward and backward pass of the surrogate respectively, and $C_{\text{LAP}}=\mathcal{O}(E)$ be the cost of computing the hypergraph Laplacian in the sparse setting.
>
> The overall time complexity of MeLA-PGD consists of one-time preprocessing cost $O\big(\tau_s (C_{\text{fwd}}+C_{\text{bwd}})\big)$ for training the surrogate model, followed by $T$ attack iterations.
> Each iteration requires one forward and backward pass through the HGNN, Laplacian construction, a top-$\delta_H$ selection over $E$ entries, and projection of perturbed node features, resulting in a per-step cost of $\mathcal{O}\big(C_{\text{fwd}}+C_{\text{bwd}} + E\log \delta_H + nd\big)$.
>
> Hence, the total time complexity is $\mathcal{O}(\tau_s (C_{\text{fwd}}+C_{\text{bwd}}) + T\big(C_{\text{fwd}}+C_{\text{bwd}} + E\log \delta_H + nd))$. The $nd$ term comes from per-step updates and projection of node features $X$. On large-scale dataset, the forward-backward passes term $\mathcal{O}((\tau_s + T)(C_{\text{fwd}}+C_{\text{bwd}}))$ dominates the cost.
>
> > Q3. Experiments on large-scale hypergraphs
>
> Please refer to our response to Q3 of Reviewer bdfx (Table 3) for experiments with OGBN-MAG and Flickr.
>
> > Q4. Comparison with recent defense baselines
>
> Thank you for raising this point. As the paper is not entirely about defense, and there were no HGNN defense paper at the time of preparation of this manuscript, we could not compare with them in the paper. To address this lacking, we conducted comparison with 2 recent defenses: (1) HGNN-Shield [3], a purification defense for hypergraphs, and (2) ElassoHGNN [3], adaptation of the robust structure refinement module of ElassoGNN [4].
>
> Tables 5-6 show the result on Cora and Citeseer. Due to space, we omit Cora-CA. Here, `Base HGNN Acc. (Poisoned)` denotes the test accuracy of the vanilla HGNN on poisoned hypergraph. `Defense Acc. (Clean)` and `Defense Acc. (Poisoned)` denote clean and poisoned test accuracy of each defense, respectively. `Gain` = `Defense Acc. (Poisoned)` - `Base HGNN Acc. (Poisoned)`.
>
> **Table 5: Comparison of MeLA-D+HGNN with existing defenses on Cora under MeLA-PGD (poisoning) attack.**
>
> | Defense | Defense Acc. (Clean) | Base HGNN Acc. (Poisoned) | Defense Acc. (Poisoned) / Gain |
> | --- | --- | --- | --- |
> | MeLA-D+HGNN | 78.43 ± 1.16 | 28.21 ± 2.09 | 72.32 ± 1.51 / **44.0** |
> | HGNN-Shield | 51.85 ± 0.36 | 37.89 ± 0.09 | 57.42 ± 0.82 / 19.5 |
> | ElassoHGNN | 48.82 ± 1.73 | 37.89 ± 0.09 | 56.09 ± 0.97 / 18.6 |
>
> **Table 6: Comparison of MeLA-D+HGNN with existing defenses on Citeseer under `MeLA-PGD` (poisoning) attack.**
> | Defense | Base HGNN Acc. (Poisoned) | Defense Acc. (Clean) | Defense Acc. (Poisoned) / Gain |
> | --- | --- | --- | --- |
> | MeLA-D+HGNN | 23.96 ± 2.31 | 71.84 ± 0.76 | 63.94 ± 1.03 / **38.9** |
> | HGNN-Shield | 35.90 ± 3.39 | 55.50 ± 4.44 | 49.94 ± 4.11 / 13.2 |
> | ElassoHGNN | 36.21 ± 3.02 | 51.28 ± 4.51 | 49.64 ± 3.76 / 12.9 |
>
> > Q5. Assumptions for convergence analysis
>
> We refer to our response to W2, Q3, and W3 of Reviewer i4AY for a detailed discussion.
>
> > W1. Limitations and impacts
>
> In the final version, we shall discuss limitations of our attack (still requiring higher training-time in minibatch setting) and defense, as you pointed out in Q1 regarding adversarial training facing incorrect semantic-label association.
>
> Hope our responses clarify your concerns.
>
> *References*
>
> [1] Boosting the Adversarial Robustness of Graph Neural Networks: An OOD Perspective, ICLR 2024.
>
> [2] Transferable Hypergraph Attack via Injecting Nodes into Pivotal Hyperedges, AAAI 2026.
>
> [3] HGNN Shield: Defending Hypergraph Neural Networks Against High-Order Structure Attack, TPAMI 2026.
>
> [4] Graph neural network meets sparse representation: Graph sparse neural networks via exclusive group lasso, TPAMI 2023.

---

> > ### Author Rebuttal · Reviewer_wjdw · 2026-04-02
> >
> > The authors have met my requirements, so my score remains unchanged.

---

> > > ### Author Response · Authors · 2026-04-07
> > >
> > > We sincerely appreciate your time in providing constructive feedback and reading our responses. We are pleased that our clarifications addressed your concerns, and we thank you for your positive assessment.

---

### Official Review · Reviewer_bdfx · 2026-03-08

**Soundness:** 2
**Presentation:** 3
**Significance:** 3
**Originality:** 3
**Overall Recommendation:** 4
**Confidence:** 4

**Summary:**

This paper fills the gap of adversarial learning in the context of hypergraphs by proposing a generic learning framework to conduct adversarial attacks and developing an adversarial training method for HGNNs. They further prove the convergence of adversarial training and provide extensive experiments to validate their proposed methods.

**Compliance With Llm Reviewing Policy:**

Affirmed.

**Final Justification:**

Given the authors' thorough experimental supplements and theoretical explanations in the rebuttal, I believe the paper covers a reasonably comprehensive range of content. Considering the remaining limitations in the current version, I suggest a weak acceptance.

**Key Questions For Authors:**

Q1. In Line 145, the perturbation budget for hypergraph structure attack is set as $\|\Delta H\|\_1\leq\delta_H$, but the perturbation budget in Theorem 2 changed to $\epsilon_{row}$ and $\epsilon_{col}$ based on $\|\cdot\|_{1,\infty}$. Is such a theoretical setting still consistent with the proposed algorithm?  Furthermore, the representation of the structure attack in Appendix B changed to $\|\Delta H\|_0$. I’m confused about the settings of the structure attacks.

Q2. Assumption 1 states that the activation and matrix operations in HGNN are Lipschitz continuous, but can it be directly concluded that the loss is smooth, especially since the formula involves the matrix inverse?

Q3. The adopted graph datasets are small-scale. It is supposed to be insufficient. In addition, it is recommonded to involve the typical hypergraph datasets to validate the effectiveness of the proposed method.

Q4.  What is the selection criterion for the baseline of the attack method? It seems to only consider the attack method proposed by Dai et al. (2018). In addition, can the comparisons with gray-box attack algorithms be considered?

**Limitations:**

Yes

**Strengths And Weaknesses:**

Strengths:

This work fills the research gap of adversarial learning in the context of hypergraphs and comprehensively investigates both white-box and gray-box settings, as well as covering both adversarial attacks and defenses.

The proposed method accounts for the global properties, stealthiness, and gray-box settings of the attack method, leading to a comprehensive and practical solution.Both the attack and defense algorithms are developed with theoretical foundations.

Weaknesses:

There is doubt about the consistency between the theoretical verification and the algorithm design (see Q1).

The experimental section fails to effectively verify the innovativeness and effectiveness of the proposed method in this paper (see Q3, Q4).

---

> ### Author Rebuttal · Authors · 2026-03-31
>
> Thank you for taking the time to review our manuscript and for offering many valuable comments that helped us improve the work.
>
> > Q1. Definition of budget
>
> Thank you for raising this subtle issue.
>
> - As the entries in $\Delta H$ are in \{0, -1, 1\}, the elementwise matrix norms $||\Delta H||_1 = ||\Delta H||_0$. Thus, Appendix B is consistent with Line 145. **The threat model and our algorithms use elementwise-norm $||\Delta H||\_0$ similar to graph attack literature, while induced norms $||.||\_{\infty}$ and $||.||\_1$ is used only for theoretical analysis of Adversary-H.** We will state this explicitely in the main (section 4.4) to avoid confusion.
>
> - Using induced-norm for theoretical analysis allows us to decouple the impact of rows and columns on the Laplacian. This isolation is not possible with elementwise $||.||\_0$-norm, since the attacker can trivially concentrate all his perturbation to one node (row of $H$) or one hyperedge (column of $H$) without violating the constraint. Thus we needed a "non-concentration assumption", which the induced $||.||\_{\infty}$-norm and induced $||.||\_{1}$-norm provides.
>
> - In this context, we would like to correct a typo in the statement of Theorem 2 in pg 6: The term  "$||\Delta H||\_{1,\infty} \triangleq \max_{e} \sum_{v} ||\Delta H_{ve}|| \leq \epsilon\_{col}$"
> should be revised as "induced norm $||\Delta H||\_{1} \triangleq \max_{e} \sum_{v} ||\Delta H_{ve}|| \leq \epsilon\_{col}$".
>
> We will clarify on this issue in the final version.
>
>
> > Q2. Assumption 1 in practice
>
> In practice, we have to consider a continuous relaxation of incidence matrix, which makes the loss smooth. We refer to our response to Reviewer i4AY, W2 for details.
>
> > Q3. Experiments on Large-scale hypergraphs.
>
> We evaluate MeLA-D on 5 additional large-scale hypergraphs: OGBN-MAG, Yelp, Trivago, Walmart and Flickr. Here, we report the runtime and drop in accuracy for each of the datasets.
>
> **Table 3: Effectiveness of mini-batch MeLA-PGD on large-scale hypergraphs.**
>
>
> | Dataset | \|V\| | \|E\| | Clean | Evasion / \%Drop | Exec. time (sec) |
> | --- | --- | --- | --- | --- | --- |
> | OGBN-MAG | 736,389 | 7,145,660 | 24.18 $\pm$ 0.23 | 22.11 $\pm$ 0.23 / 8.6 | 4614.94 |
> | Yelp | 50,758 | 4,523,594 | 31.98 $\pm$ 0.40 | 15.65 $\pm$ 1.11 / 51.1 | 2036.91 |
> | Trivago | 172,738 | 726,861 | 37.27 $\pm$ 1.96 | 24.01 $\pm$ 1.82 / 35.6 | 718.61 |
> | Walmart | 88,860 | 460,630 | 97.67 $\pm$ 0.01 | 95.67 $\pm$ 0.17 / 2.0 | 348.76 |
> | Flickr | 7,575 | 479,476 | 89.53 $\pm$ 0.78 | 35.75 $\pm$ 2.68 / 60.1 | 113.13 |
>
>
> > Q4. Comparison with other attacks
>
> Similar to [1], the baselines are selected to be representative of two mainstream structural attack types: Random (rand) and Gradient-based (GradArgmax), and random feature attack. Following [1], we include adaptations of DICE and, as per reviewer 7tip, Hyperattack as additional baselines for comparison. Table 4 shows that MeLA-PGD outperforms both adapted DICE and Hyperattack.
>
>
> (i) DICE: It is adapted from [2] that uses all train+test nodes to compute proxy edge labels: an proxy edge label is $c$ if majority of its nodes have label $c$. Deletion and Connection occurs with the same probability. Deletion: For each randomly selected existing node-edge pair, it removes them if node label == proxy edge label. Connection: For each randomly selected non-existent node-edge pair, it adds them if node label != proxy edge label.
>
> (ii) Hyperattack [3]: We modify it to use LinHGNN as surrogate to make it gray-box.
>
> **Table 4: Comparing MeLA-PGD with DICE and Hyperattack on Citeseer dataset.**
>
> | Attack       | HyperMLP\(P\) / Drop | HyperMLP\(E\) / Drop | AllSet\(P\) / Drop | AllSet\(E\) / Drop | HGNN\(P\) / Drop | HGNN\(E\) / Drop |
> |--------------|------------------|-------------------|----------------------|----------------------|---------------|----------------|
> | DICE         | 69.35 $\pm$ 0.99 / 6.2 | 72.95 $\pm$ 0.68 / 1.1 | 68.34 $\pm$ 0.69 / 6.3   | 66.69 $\pm$ 0.95 / 8.3   | 65.80 $\pm$ 0.81 / 11.9 | 62.80 $\pm$ 0.63 / 14.8 |
> | HyperAttack  | 73.19 $\pm$ 0.45 / 1.0 | 72.95 $\pm$ 0.68 / 1.1 | 73.31 $\pm$ 0.00 / 0.0   | 70.26 $\pm$ 0.49 / 3.8   | 73.70 $\pm$ 0.31 / 0.7  | 68.74 $\pm$ 0.58 / 7.0  |
> | MeLA-PGD     | 57.08 $\pm$ 1.78 / **21.7** | 43.60 $\pm$ 0.82 / **38.9** | 28.82 $\pm$ 0.75 / **60.0**  | 38.24 $\pm$ 2.26 / **49.1**  | 54.37 $\pm$ 1.17 / **26.5** | 35.51 $\pm$ 2.67 / **51.1** |
>
> We have obtained similar results on Cora-CA and Cora which we omit in the rebuttal for brevity.
>
> Hope our responses clarify your concerns. We would be available to discuss if you have further question/comment.
>
> *References:*
>
> [1] Transferable Hypergraph Attack via Injecting Nodes into Pivotal Hyperedges, AAAI 2026.
>
> [2] Waniek et al., Hiding individuals and communities in a social network, 2018.
>
> [3] Hu et al., Hyperattack: Multi-gradient-guided white-box adversarial structure attack of hypergraph neural networks, ArXiv 2023.

---

> > ### Author Rebuttal · Reviewer_bdfx · 2026-04-03
> >
> > I appreciate the authors' careful response, and most of my concerns have been addressed. I will adjust my score positively, and would kindly ask the authors to ensure that all necessary experiments discussed above are fully included in the final manuscript.

---

> > > ### Author Response · Authors · 2026-04-07
> > >
> > > Thank you for your careful consideration of our response and for the constructive feedback. We are pleased that our clarifications have addressed your concerns, and we appreciate your positive reassessment. We will make sure that all relevant experiments discussed are included in the final manuscript.

---

### Official Review · Reviewer_7tip · 2026-03-11

**Soundness:** 3
**Presentation:** 2
**Significance:** 2
**Originality:** 2
**Overall Recommendation:** 3
**Confidence:** 4

**Summary:**

This paper studies the adversarial robustness (poisoning and evasion) of hypergraph neural networks (similar as graph neural networks but operating on more general hypergraphs) in the gray-box setting (no access to the model under attack, but instead access to training data). The main contributions include (1) a meta-learning attack loss, (2) an empirical evaluation of the attacks and corresponding defense based on adversarial training, and (3) theoretical arguments that perturbations of hypergraphs differ from graph perturbations, as well as (4) a theoretical convergence analysis of their adversarial training approach.

**Compliance With Llm Reviewing Policy:**

Affirmed.

**Final Justification:**

I carefully considered the author's rebuttal, have read the discussion with other reviewers and generally agree that the empirical results of this paper are thorough. But overall, the technical contributions of this paper are limited extensions to the hypergraph domain and their impact (in particular, of the rather central hypergraph Laplacian smoothing) is not discussed enough. My recommendation is weak reject.

**Key Questions For Authors:**

1. Regarding the “median relative drop” evaluation, what do you compute exactly? According to the text this should be clean accuracy minus accuracy after the attack. However, for example for Table 1, computing clean accuracy of HyperMLP minus the mean (?) accuracy after MeLA-PGD is 73.77 - 57.08 = 16.69, which differs from the “median relative drop” metric 21.7. Does this stem from the difference between mean and median? Why do you report the median instead of the mean?
2. Does your attack and defense approaches generally scale to larger graphs? There is an additional experiment with a larger graph in Appendix G.1, but this experiment is missing a further discussion.
3. Line 206: Do you perform adversarial training only for feature perturbations or also for edges?

**Limitations:**

Authors only mention that this work proposes empirical attacks/defenses and not robustness certificates. While it is positive that they mention this limitation, it is also trivial given the title and contributions of the paper. What would help is if the authors would mention limitations of their particular adversarial attack (e.g. if this scales to larger graphs, if there are limitations for sparser/denser graphs, …) or what potential limitations of the empirical defense could be.

**Strengths And Weaknesses:**

**Strengths**

The discussion in the paper is easy to follow; context and problem are well-motivated, contributions are clear and well-described, and the theoretical approach is sound as well. Although the experiment section in the main text is rather short, the paper offers many additional ablation studies in the appendix. Experimental details are sufficiently described in the paper and appendix. The overall quality of the paper is good.

**Weaknesses**

W1) Limited technical contributions. The authors argue that existing attacks against GNNs cannot be easily transferred to Hypergraph-NNs and that fundamentally new, hypergraph-specific attacks are required. However, in the end, meta-learning attacks seem to transfer to hypergraphs seamlessly. While it is good to know that the extension to hypergraphs works in general, there does not appear to be a deeper technical challenge beyond the transfer of existing techniques to a slightly different discrete structure. Even if hypergraphs might require different attacks, the technical contributions in this direction remain rather limited in this paper. For example, it remains unclear from the empirical analysis if there is a particular contribution that was required to make meta-learning attacks work for hypergraphs specifically. The perhaps more interesting contribution tailored to hypergraphs might be the hypergraph Laplacian smoothing loss $\mathcal{L}_{smooth}$, however, the ablation study in Table 5 (Appendix D) indicates rather marginal improvements in attack strength, suggesting that this loss is not required to enable strong attacks. The authors do not discuss a broader impact either, and overall, the impact appears rather limited to a transfer of meta-learning attacks to this hypergraph domain.

W2) Missing baseline comparisons. The authors argue that “existing attacks only work in white-box setting[s]” and use this argument to avoid comparing to white-box baselines. However, the only difference in the proposed approach for gray-box settings is to train a surrogate model. It remains unclear why existing attacks cannot do this step as well. In particular, the paper does not compute white-box baselines such as e.g. hyperattack by replacing the model under attack with the same surrogate model already trained by the authors.

W3) Limited empirical discussion. The authors argue that “perturbations designed in the graph domain may not correspond to valid or meaningful perturbations in the hypergraph domain” and continue with a theoretical analysis about this difference. However, the authors miss to further develop this argument empirically. In particular, a naïve baseline that (1) converts the hypergraph into a normal graph, (2) computes an adversarial perturbation of the graph, and (3) converts this attack back to the hypergraph domain, would be required to support this statement (even if the mapping between graphs and hypergraphs is not unique). If computing this baseline is not possible, the paper would require a better discussion for why such a mapping is not possible.

W4) General writing quality. Discussing related work in a separate related work section would further improve clarity (currently this is mixed with the introduction and background). Furthermore, a Figure 1 that highlights the main contribution would improve the paper as well.

Overall, weaknesses W1 and W2 currently outweigh the strengths of this paper.

**Minor:**
* The graph notation in the introduction (Lines 046-047 right column) is not clear, making the example hard to follow.
* It would be helpful if the experiments in the main text could point to Appendix G.2. Without the explanation in the appendix, the “statistically significant” statement in the tables is not meaningful. The authors should at least mention the significance level used to decide if the experiments are statistically significant in the main text.
* Inconsistency: Lemma 1 in the Appendix (and the proof) differs slightly from Lemma 1 in the main paper.

---

> ### Author Rebuttal · Authors · 2026-03-31
>
> Thank you for the time and effort to review our paper.
>
> > Q1.  Explaining “Median relative drop”
>
> The relative drop% for each run is computed as 100 * (clean accuracy - accuracy after attack) divided by clean accuracy. We report median instead of mean to not let outlier performance (good or bad) affect our evaluation. The difference you highlighted stems from the measure being relative rather than an absolute difference.
>
> > Q2. Large-scale dataset
>
> We refer to Reviewer bdfx (Q3) for experiments with 5 large-scale datasets. We will discuss them in detail in Appendix G1.
>
> > Q3. Perturbations considered
>
> We perform adversarial training for both the feature and structure (edge) perturbations.
>
> > W1. Technical contributions
>
> First, we respectfully disagree with the premise that MeLA is an extension of meta-learning attacks to hypergraphs. We clarify that MeLA is **not a meta-learning attack** like Meta-attack for GNNs [2], as we do not adopt a "learning to learn" (like MAML) framework that enables adaptation to new tasks [1]. Meta-attack computes attack gradients through the surrogate training process, while MeLA computes attack gradients on a surrogate that was trained once and then kept fixed.
>
> Second, we study how Laplacian, a key component of HGNNs, behaves under perturbations, rather than making meta-learning work for hypergraph. Our study reveals deep insights about desirable characteristics of attacks and defenses on hypergraphs. This study is, by itself, a key technical contribution overlooked in prior HGNN works.
>
> Third, indeed, the Laplacian smoothing loss is an important contribution, as it improves the attack effectiveness but the stealth loss is also a key contribution. Holistically, they make MeLA not only effective but stealthy at both the structure-level and operator-level.
>
> Fourth, beyond the loss design, a unique characteristic of MeLA-PGD (absent in other HGNN attacks) is that it does projected gradient descent-like continuous updates to a continuous relaxation of the incidence matrix $H$, yet maintains an optimal top-$\delta_H$ flip mask $M$ to enable boolean updates to $H$ (flip $H_{ij}$ if $M_{ij}=1$). This yields a final perturbation $H^{T}$ that is boolean, but the intermediate iterates $H^{t}~(1\leq t \leq T)$ remain continuous.
>
> Lastly, as pointed out by reviewer i4AY, our attack setting is **unique**: we consider **both structural and feature perturbations simultaneously, whereas most papers (Graph & Hypergraph) have focused primarily on either structure or feature**. For instance, Meta-attack, a structure attack, is shown to work only for Boolean node features, whereas MeLA does not have such limitations.
>
> > W2. Missing Baselines
>
> We refer to our response to Q4 of Reviewer bdfx for results on two attack baselines including Hyperattack.
>
> > W3. Limited empirical discussion
>
> Thank you for your attention to this detail.
>
> The most problematic of the 3-step process you mentioned is *step\#3: converting the poisoned graph back to a valid hypergraph*. Suppose we apply clique-expansion to convert a hypergraph to a graph. Even if we do not conduct any attack, an exact inversion of that clique-graph back to a hypergraph is generally ill-posed because many different hypergraphs can yield the same clique-expanded graph. The attack only aggravates this issue by adding/removing edges, since **a valid hypergraph whose clique-expansion corresponds to a given attacked graph may not exist**.
>
> Nevertheless, we have adopted a greedy heuristic to approximately compute the inversion to enable Meta-attack [2] on the clique-expansion of the input hypergraph. At step 3, we project Meta-attack's attacked clique-graph back to a hypergraph by greedily perturbing the original incidence matrix so that its clique expansion matches the attacked graph as much as possible. At each iteration, until budget is exhausted, we evaluate candidate incidence flips and select the one that maximizes the current match between the hypergraph's clique-expanded graph and attacked graph.
>
> Results show that this approach does not yield any stronger attack than the ones that we already considered.
>
> **Table 2: Performance of Meta-attack on AllSetTransformer**
>
> | Dataset | Poison / Drop% | Evasion/Drop% |
> |---|---|---|
> | Citeseer | 73.10 $\pm$ 0.75/0.00 |  69.63 $\pm$ 0.97/4.79 |
> | Cora | 77.89 $\pm$ 0.74/0.38 | 74.32 $\pm$ 0.85/5.28 |
> | Cora-CA | 81.78 $\pm$ 0.98/1.60 | 79.12 $\pm$ 1.07/5.35 |
>
> > W4 + minor comments + Limitations
>
> Thank you for the suggestions. We address all of them in the final version. We shall discuss limitations of our attack and defense, as reviewer wjdw (Q1) pointed out regarding adversarial training facing incorrect semantic-label association.
>
> Hope our responses clarify your concerns. We are available if you have any further question/comment.
>
> *References:*
>
> [1] Vanschoren, J. (2019). Meta-Learning.
>
> [2] Zugner D., Gunnemann S., Adversarial attacks on graph neural networks via meta-learning, ICLR 2019.

---

> > ### Author Rebuttal · Reviewer_7tip · 2026-04-01
> >
> > Thank you for your rebuttal, and your response in particular to weakness 3.
> >
> > Regarding weakness 1, I understand that your "meta-objective-based learning framework" is not a meta-learning attack like Meta-attack and that the surrogate is fixed. Apart from that, does the fixed surrogate not actually limit the generality of your framework? Either way, it still remains unclear to me which technical contribution exactly is required for hypergraphs that is not already known for graphs. For example, the stealth loss is not specific to hypergraphs. In particular, the only part of the MeLA objective that is specific for the hypergraph domain is the Hypergraph Laplacian smoothing (which, as far as I understand, is not required for a strong attack). Also the continuous relaxation you mentioned is not new and has been studied in attacks against GNNs before, see [1] for an example. Overall, my concerns are only partially addressed so far.
> >
> > [1] Topology Attack and Defense for Graph Neural Networks: An Optimization Perspective

---

> > > ### Author Response · Authors · 2026-04-07
> > >
> > > Thank you for the follow-up discussion.
> > >
> > > > does the fixed surrogate ...
> > >
> > > 1. Our framework can accommodate both adaptive and non-adaptive training of the surrogate (ref. Section 4.2, Line 238-254). For grey-box attack (MeLA), we train a surrogate once and then use our meta-objective-based learning framework to construct feature and incidence matrix perturbations. For defense (MeLA-D), the robust model is adaptively trained at each iteration with the feature and incidence matrix perturbations generated in the previous iteration.
> > >
> > > > which technical contribution exactly is required for hypergraphs...
> > >
> > > 2. Thanks for pointing to [1]. We will cite it in the final version. [1] adapted PGD from continuous image space to discrete graph topology, and thus, as a novel contribution, [1] redefined the optimization domain over graph adjacency matrices. MeLA and MeLA-D extend gradient-based attacks and defenses to hypergraphs, where the optimization domain (incidence matrix) and operator (hypergraph Laplacian) fundamentally change the perturbation geometry, as corroborated by the empirical results below.
> > >
> > > 3. Since hypergraphs essentially generalize graphs, many structural and optimization primitives, such as continuous relaxation and stealth loss, used in hypergraph learning, including MeLA and MeLA-D, have antecedents in the graph learning literature. We clarify that our contribution is not a new optimizer for hypergraphs, rather **carefully constructing the gradient-based attacks and defenses for the hypergraph structure (e.g., incidence matrix, feature space) and incorporating the hypergraph Laplacian for smoothing. These induce hypergraph-specific higher-order, non-local perturbations with higher effectiveness that are not captured by graph-based formulations**.
> > >
> > > Now, we discuss the new empirical results in detail.
> > >
> > > - **(a) Non-locality of hypergraph-specific perturbations.**  We compare a single incidence flip with a single adjacency (clique-edge) flip projected back to a hypergraph. Under identical settings, **incidence flips affect substantially more nodes**, indicating less localized perturbation behavior. Here, a node is affected if its embedding changes by $>10^{-4}$. This shows that a single incidence perturbation affects about 2x more nodes on average and up to 5x more in the worst case, implying **incidence space-specific perturbations propagate more broadly through higher-order structure than adjacency-specific ones**.
> > >
> > > | Metric | Adjacency flip | Incidence flip |
> > > |---|---|---|
> > > | Affected nodes (mean) | 15.9 | **31.8** |
> > > | Affected fraction (mean) | 0.0048 | **0.0096** |
> > > | Max affected nodes | 41 | **198** |
> > >
> > > - **(b) Optimization trajectories with lower drifts but higher effectiveness.** We compare adjacency-space (via clique expansion) and direct incidence-space optimization for generating perturbations. We observe that with the same budget and steps, (1) **adjacency-space optimization yields no accuracy degradation and higher embedding/Laplacian drift, (2) incidence-space optimization induces a higher accuracy drop and lower embedding/Laplacian drift**. This indicates that **optimizing over incidence space rather than adjacency yields more effective HGNN attacks with smaller structural drift**.
> > >
> > > | Space | Metric / Steps =>      |             2 |               6 |             10 |
> > > | ------| -------------- | -------------| -------------| --------------|
> > > | Adjacency| emb. drift | 5.59 ± 0.57 | 10.40 ± 0.33 | 13.82 ± 0.56 |
> > > | Incidence | emb. drift | 4.38 ± 0.99 |  8.23 ± 0.29 | 10.71 ± 1.14 |
> > > | Adjacency| Laplacian drift | 0.57 ± 0.08 |  1.06 ± 0.12 |  1.39 ± 0.09 |
> > > | Incidence| Laplacian drift | 0.59 ± 0.03 | 0.99 ± 0.03 | 1.23 ± 0.02 |
> > > | Adjacency| acc. drop  | 0.00 ± 0.00 | 0.00 ± 0.00 |  0.000 ± 0.00 |
> > > | Incidence| acc. drop  | 0.03 ± 0.05 | 0.03 ± 0.05 | 0.03 ± 0.05 |
> > >
> > > - **\(c\) Laplacian-level (spectral) differences.** We further compare Meta-attack (applied on hypergraphs via clique expansion), MeLA-PGD, and MeLA-PGD ($\alpha=0$, i.e., without smoothing) through their effects on the hypergraph Laplacian's spectrum. Along with the Lalpacian drifts, we report the high- and low-frequency drifts, which are the absolute differences in the largest  and smallest 20 eigenvalues after attack, respectively.
> > > MeLA-PGD ($\alpha=0$) induces the largest high-frequency spectral distortion (0.0239), whereas MeLA-PGD produces a smaller high-frequency shift (0.0128), and Meta-attack produces none. Meta-attack still exhibits the largest overall Laplacian drift (3.35). Thus, **the attacks differ not only in the total amount of operator perturbation they induce but also in how that perturbation is distributed across the spectrum.**
> > >
> > > | Method | Low-freq drift | High-freq drift | Laplacian drift ($\Delta L$)|
> > > |---|---|---|---|
> > > | MeLA-PGD | 2.4e-7 | 0.0128 | 1.27 |
> > > | MeLA-PGD \($\alpha=0$\) | 4.6e-7 | 0.0239 | 1.41 |
> > > | Meta-attack | 0 | 0 | 3.35 |
> > >
> > > We will incorporate these results and discussions in the final draft.

---

### Official Review · Reviewer_i4AY · 2026-03-12

**Soundness:** 3
**Presentation:** 3
**Significance:** 3
**Originality:** 3
**Overall Recommendation:** 4
**Confidence:** 3

**Summary:**

The paper is interested in the general subject of adversarial robustness in the specific context of Hyper Graph Neural Networks. The authors propose some adversarial attacks, that can perturb both the hypergraph and the node features. They additionally propose an adversarial training procedure to enhance the robustness of HGNN.

**Compliance With Llm Reviewing Policy:**

Affirmed.

**Final Justification:**

I believe that the paper tackles an important and underexplored problem and presents a technically solid framework for both attacks and defenses in the gray-box setting.

In my initial review, I had some concerns regarding the theoretical framework. The provided rebuttal addressed most of my initial theoretical concerns, especially by clarifying the local nature of the convergence assumptions and the role of continuous relaxation.

At the same time, after reading the other reviews and discussion, I think the contribution is somewhat limited by concerns about novelty, empirical positioning, and practical impact. In particular, it remains only moderately convincing which parts of the method are truly specific to hypergraphs rather than adaptations of existing graph-based ideas. It seems also that scalability and complexity were  an issue that was clarified in the rebuttal, but still appear more as limitations than clear strengths.

Overall, while I find the paper relevant, reasonably complete, and technically sound, hence my "weak accept" assessment, there are a number of missing points making the manuscript rather moderate than with strong impact. For these reasons, I would still maintain a weak accept recommendation, but at the same time would question the paper's impact and therefore if it reaches the publishing bar of ICML, which I would suggest is out of my scope since I am not the expert in this field.

**Key Questions For Authors:**

- Is the proposed MeLA sensitive to the choice of the surrogate model? Have you tested this empirically ?
- How does the attack perform on larger hypergraphs? While the authors provide some empirical run time, I couldn’t find any time-complexity derivation to better understand how it could scale.
- Could you provide some empirical elements to validate the tightness of the theoretical perturbations bounds? (specifically in link to my previous weakness, so I can understand the validity of the assumptions).

**Limitations:**

- I believe that overall the paper is very interesting. Please note that my background is on adversarial robustness for GNNs, so I can't judge how innovative the method itself is.
- The theoretical study is a little bit limited giving some non-clear assumptions that should be clarified (hence why the weak accept).

**Strengths And Weaknesses:**

**Strengths:**
- The subject of adversarial robustness and the study of the vulnerabilities of the models is very important. Specifically, while previous work on GNNs is very intense, the work on hypergraph neural networks is on the other hand very under-explored. The paper therefore fills this gap by providing both attacks and a way to defend and enhance robustness through adversarial training.

- The combination of both structural and feature-based attacks is very interesting, which is very different from GNNs, where the majority of papers have rather focused on structural perturbations.
- The authors provided an extensive empirical results, showcasing the worth of both their attacks and proposed defense.

**Weakness:**
- On the theoretical analysis:
    - I have questions regarding Theorem 2, where the authors consider a block incidences matrices and some artificial sign vectors. While this indeed valid, it is a construction assumption of the graph, and therefore wondering if it applies to typical real-world hypergraphs.
    - Regarding assumption 1 on the training loss gradient being Lipschitz continuous: From my understanding you consider that the structure $H$ is discrete (connections/edges) and therefore the mapping from H to the laplacian is piecewise discontinuous, are questions about the gradient with respect to $H$ arises here. Could you please clarify this? It’s totally fine to consider that you consider a “normalized” adjacency and therefore one could consider the space to be continuous. And also in this latter case, I believe that if the node degree is too small, the derivate grows large (given the inverse relation), and therefore the mapping can’t be globally Lipschitz (you will need some assumptions there as well).
        - So in this specific direction, the authors should clarify that the result doesn’t not guarantee global optimality and depends clearly on some Lipschitz constants. (similar to the classical results of optimization where some assumptions are made).
    - Could you please clarify more the assumption 2, on the strong concavity of the train loss? Specifically, and again similar to my previous point, if you consider that the degrees are small, then the inverse of the degree matrix can create a large curvature, so even local curvature be unstable. I am a bit suprised here, because from my previous experience, I have only seen assumptions on smoothness for instance or weak concavity, but strong concavity is a much more stronger requirement and I have questions about its validity in empirical practical settings.

---

> ### Author Rebuttal · Authors · 2026-03-31
>
> Thank you for your valuable feedback and recognising the strengths of our work.
>
> > Q1. Sensitivity to surrogate
>
> In Appendix Table 8, we tested this empirically on two surrogates: MLP and Linearized HGNN. MeLA-PGD w/ LinHGNN consistently outperforms MeLA-PGD w/ MLP. But MeLA-FGSM w/ MLP  outperforms MeLA-FGSM w/ LinHGNN.
>
> > Q2. Time complexity
>
> Please see our response to reviewer wjdw, Q2.
>
> > W1. Theorem 2
>
> Thm 2 gives an **existential lower bound**- we are crafting a hypergraph instance and degree-preserving perturbation that yields this bound. The signed block incidence vectors $s_n$ and $s_m$ are devised to define an admissible perturbation $\Delta H$. Unlike $H$, $\Delta H$ may not necessarily define any real hypergraph.
> > W2,Q3. Assumption 1
>
> a. We clarify that, in practice, the gradients with respect to $\mathcal{H}$ are **not taken over the discrete incidence space**, but over a **continuous relaxation** via $\Delta H$:
> $H_{pert} = CLIP(H + \Delta H, 0, 1)$. Hence, the meta-loss is optimized w.r.t $\Delta H$, which parameterizes the **relaxed hypergraph**. In regions where clipping is inactive, this mapping is locally linear, and gradients w.r.t. $\Delta H$ coincide with gradients w.r.t. $H_{pert}$. Hence, the Lipschitz/smoothness assumption is naturally interpreted in this **relaxed continuous space**. Such relaxation is common to conduct differentiable optimization over discrete structures [1-3].
>
> b. We clarify that the **global Lipschitz continuity is not needed**. We interpret Assumption 1 as a **local smoothness condition, valid along optimization trajectories where the hypergraph remains well-conditioned**, meaning that node degrees and edge cardinalities stay bounded away from zero.
>
> *Numerical evidence:* MeLA-D generated perturbations remain well-conditioned without leveraging Lipschitzness assumption.
>
> **Table 1: Degree conditioning & gradient stability across MeLA-D steps**
> | Dataset        | Min Node Deg | Min Edge Card | Max Node Cond. num. | Max Edge Cond. num. | Mean Grad Ratio | Max Grad Ratio |
> |-------------|------------|--------------|--------------|--------------|--------------|---------------|
> | Cora-CA    | 1.0          | 1.0            | 146.2         | 143.8        | 2.8e-8         | 6.1e-8        |
> | Cora        | 1.0          | 1.0            | 120.5         | 118.6         | 1.8e-8         | 4.7e-8        |
> | Citeseer    | 1.0          | 1.0            | 98.3          | 102.1         | 2.1e-8         | 6.5e-8        |
>
> *Observations:*
> - **Min. node degree** and **min. edge cardinality** remain positive, as degenerate perturbations are explicitly prevented during optimization via projection.
>
> - There are no **zero-degree nodes** or **empty edges**.
>
> - The node and edge condition numbers are moderate.
>
> To further assess smoothness, at each MeLA-D step, we compute a **local gradient-sensitivity proxy** of $L_{\phi \mathcal{H}}$ around $H$ as $\frac{||\nabla_\phi L (\phi, H) - \nabla_\phi L (\phi, H')||_2}{ ||vec(H) - vec(H')||_2}$, where $H' \in \mathcal{B}(H,\epsilon)$. Table 1 shows that this ratio remains consistently low (~$10^{-8}$). This suggests that the loss behaves smoothly in the region explored by MeLA-D.
>
> c. As stated in the main paper, our result yields a **first-order stationarity guarantee**, and **not global optimality**. This is consistent with standard analyses of non-convex adversarial training [4].
>
> > W3. Assumption 2
>
> We **do not claim** that the HGNN training loss is **globally strongly concave** with respect to hypergraph perturbations.
> Assumption 2 follows the analytical framework of [4]: it is a **local condition** around the true $H$ for controlling the inner maximization and relate approximate ascent steps to bound error in the outer minimizer's gradient.  Let $H^{t+1}(\phi^{t+1})$ be the result of one gradient ascent step from $H^{t}(\phi^{t})$ on the inner maximization problem. Assumption 2, combined with Assumption 1, ensures that the change in the outer gradient remains stable:
> $||\nabla_\phi L(\phi^{t+1}, H^{t+1}(\phi^{t+1})) - \nabla_\phi L(\phi^{t}, H^{t}(\phi^{t})) ||_2 \le L||\phi^{t+1} - \phi^{t}||\_2,$
>
> where $L=\frac{1}{\mu} L_{\phi\mathcal{H}}L_{\mathcal{H}\phi} + L_{\phi\phi}$ and $H^{t+1}, H^{t} \in \mathbb{H}_{\delta_H, \delta_X}$. This is used to facilitate convergence analysis rather than describing the global loss landscape.
>
> *Numerical evidence:*  Although the loss landscape might have irregular curvature, the iterates enter a well-conditioned regime within 2-3 steps. We measure a local curvature proxy $\mu^t$ along consecutive inner ascent iterates ($H^{t}, H^{t+1}$). We find that the vast majority of inner steps (90-95%) have $\mu^t > 0$. For e.g., on cora: 193.1, -10, subsequently settling in the range of 2-5.
>
> Hope our responses clarify raised concerns.
>
> *References*
>
> [1] https://arxiv.org/pdf/1611.00712
>
> [2] https://arxiv.org/pdf/1903.11960
>
> [3] https://arxiv.org/pdf/2006.13009
>
> [4] https://arxiv.org/pdf/2112.08304

---

> > ### Author Rebuttal · Reviewer_i4AY · 2026-04-02
> >
> > I thank the authors for the additional clarifications. I believe that I grasped more the theoretical results now, and I maintain my already positive score while keep monitoring the other reviews to see if I have missed something.

---

> > > ### Author Response · Authors · 2026-04-07
> > >
> > > Thank you for the thoughtful reviews and for taking the time to read our clarifications. We greatly appreciate your positive assessment and your consideration of the broader discussions during rebuttal.

---

### Decision · Program_Chairs · 2026-04-30

**Decision:**

Accept (regular)

**Comment:**

This paper studies the adversarial robustness (poisoning and evasion) of hypergraph neural networks (similar as graph neural networks but operating on more general hypergraphs) in the gray-box setting (no access to the model under attack, but instead access to training data). Some reviewers have some concerns, which have been addressed by authors. I tend to accept it.